# Decay of driver mutations shapes the landscape of intestinal transformation

Filipe C. Lourenço[1], Iannish D. Sadien[1], Kim Wong[2], Sam Adler[1], Ashley Sawle[1], Leonor Schubert Santana[3], Lee Hazelwood[1], Giada Giavara[1], Anna M. Nicholson[1], Matthew D. Eldridge[1], Noori Maka[4,5], Gerard Lynch[3], Stephen T. McSorley[3,4], Joanne Edwards[3], Richard Kemp[1], David J. Adams[2] & Douglas J. Winton[1 ✉]

Colorectal cancer (CRC) has traditionally been thought to develop through stepwise mutation of the *APC* tumour suppressor and other driver genes, coupled with expansion of positively selected clones. However, recent publications show that many premalignant lesions comprise multiple clones expressing different mutant APC proteins[1–4]. Here, by mediating transformation on different mouse backgrounds containing mutations in *Kras* or other common CRC driver genes, we establish that the presence of diverse priming events in the normal mouse intestinal epithelium can change the transformation and clonal-selection landscape, permitting the fixation of strong driver mutations in *Apc* and *Ctnnb1* that are otherwise lost due to negative selection. These findings, combined with our demonstration of mutational patterns consistent with similar priming events in human CRC, suggest that the order in which driver mutations occur in intestinal epithelium can determine whether clones are positively or negatively selected and can shape subsequent tumour development.

Loss of the APC tumour suppressor is ubiquitous in CRC and is generally perceived as an early event and strong driver of the disease[5,6]. Truncating *Apc* mutations promote clonal growth during tumour initiation while also suppressing the growth of wild-type (WT) neighbours through WNT pathway inhibition[7–9]—a form of 'supercompetitor' behaviour. However, recent reports that many early intestinal lesions are polyclonal have challenged the accepted aetiology of CRC[1–4]. Clonal cooperation in polyclonal tumours can involve recruitment of clones expressing N-terminal APC truncations by those expressing more C-terminal truncations, suggesting an unequal transformation landscape for different *Apc* mutations[2]. This phenomenon may be related to the 'just right' model of CRC development, in which the optimal amount of WNT pathway disruption for transformation is achieved through combinations of APC mutants that retain some ability to regulate the pathway by binding to the β-catenin oncoprotein[10]. Our previous work established that polyclonality is context dependent: when initiated on an epithelial background containing activating mutations in *KRAS*, most tumours were monoclonal[2]. The objective of this study was to understand how different contexts created by priming normal mouse intestinal epithelium with *KRAS* and other driver mutations affect the selection and transformation landscape for different mutations in *Apc* and other CRC driver genes.

## Priming affects transformation rate

To model the effects of different cancer-driver mutations in healthy intestine, we created pro-oncogenic intestinal fields expressing *Kras*^*G12D*^ or deficient in *Fbxw7* (*Fbxw7*^*null*^) or *Trp53* (*Trp53*^*null*^) through tamoxifen treatment of inter-crossed *Villin-cre*^*ERT2*^ mouse lines (Methods and Supplementary Fig. 1a–d). These mice remain disease-free for many months in the absence of chemical mutagens[11–13] (Fig. 1d, Extended Data Fig. 2c and Supplementary Table 2). Tumorigenesis was then initiated through exposure to *N*-ethyl-*N*-nitrosourea (ENU), a direct-acting chemical mutagen that is rapidly cleared from the intestinal epithelium[14,15] (Extended Data Figs. 1 and 2a,b). WT mice treated with ENU survived for 100–480 days (median, 251 days; Fig. 1a–d). Mortality before 180 days was frequently associated with lymphoma, while mice with longer survival times primarily presented with intestinal tumour burden (multiplicity, 1–10; mean, 3.1; Fig. 1c and Supplementary Table 1), predominantly in the small intestine (SI) (Fig. 1b). By contrast, *Kras*^*G12D*^-primed mice treated with ENU had a much shorter median survival time of 37 days, and hundreds of intestinal lesions (mean (s.d.) multiplicity: 921 (184); Fig. 1d,e and Supplementary Table 1). Similarly, *Fbxw7*^*null*^-prime and *Trp53*^*null*^-primed mice treated with ENU had shorter survival times compared with ENU-treated WT mice (median, 64 and 120 days, respectively) and increased tumour multiplicity (mean (s.d.): 157 (41) and 40 (18), respectively; Fig. 1d,e and Supplementary Table 1).

These initial results indicated that different priming fields confer quantitatively different susceptibilities to ENU-mediated transformation and motivated the creation of additional pro-oncogenic fields: heterozygous loss of *Apc* or *Trp53*; homozygous loss of *Pten* or *Arid1a*; and gain-of-function mutations for *Pik3ca*^*H1047R*^ or *Notch1* intracellular domain (N1-ICD) (*Apc*^*het*^, *Trp53*^*het*^, *Pten*^*null*^, *Arid1a*^*null*^, *Pik3ca*^*H1047R*^ and *N1-ICD*^*het*^, respectively), as well as compounded allelic fields consisting of *Trp53*^*het*^*N1-ICD*^*het*^ and *Trp53*^*null*^*N1-ICD*^*het*^. Mice primed with all mutations except for *Trp53*^*het*^, *Pik3ca*^*H1047R*^ and *Arid1a*^*null*^ showed a

[1]Cancer Research UK Cambridge Institute, University of Cambridge, Cambridge, UK. [2]Wellcome Genome Campus, Wellcome Sanger Institute, Cambridge, UK. [3]Wolfson-Wohl Cancer Research Centre, School of Cancer Sciences, University of Glasgow, Bearsden, UK. [4]NHS Greater Glasgow & Clyde, Glasgow, UK. [5]Queen Elizabeth University Hospital, Glasgow, UK. ✉e-mail: doug.winton@cruk.cam.ac.uk

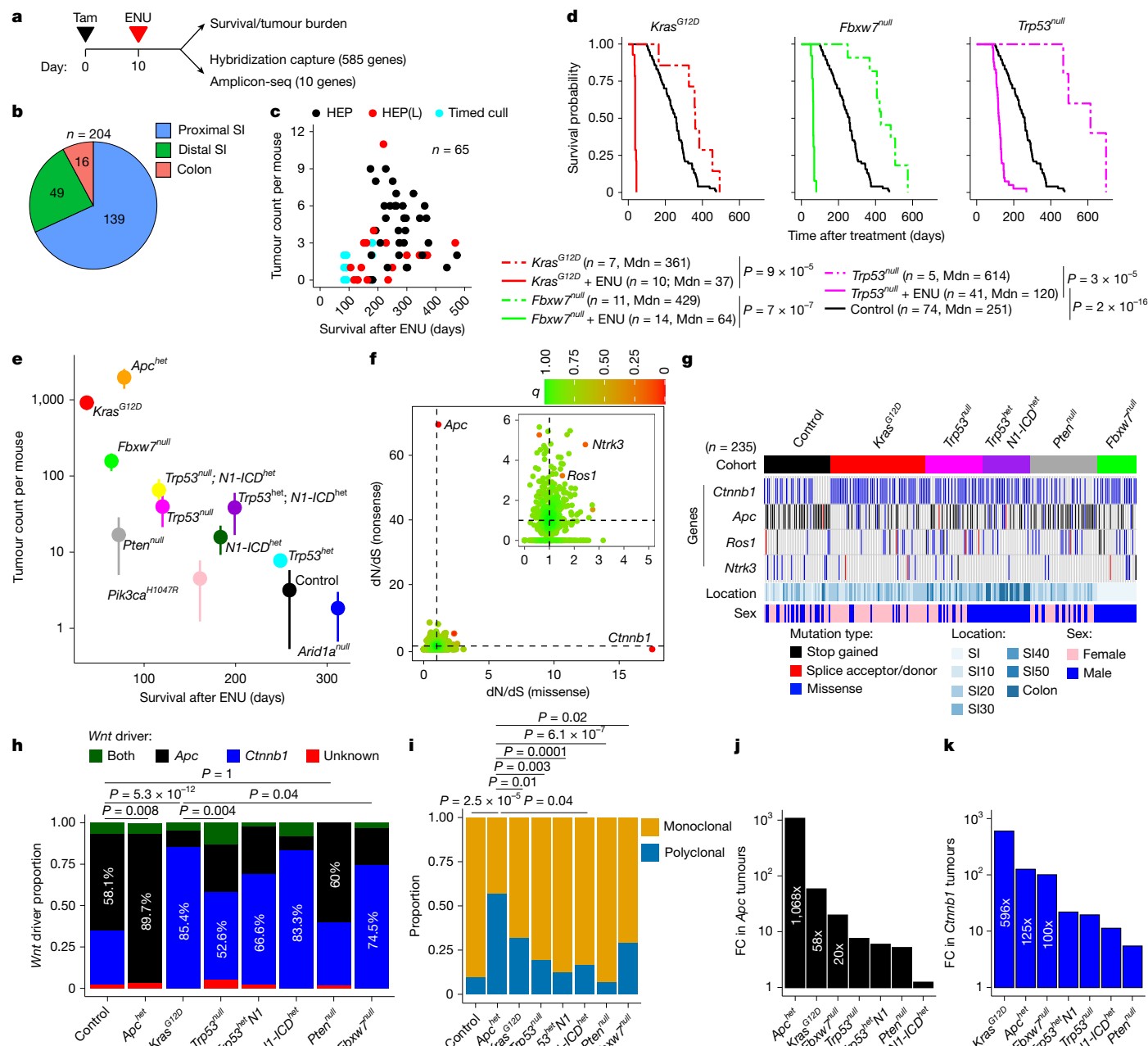

**Fig. 1 | Tissue priming determines tumour multiplicity in response to ENU mutagenesis. a**, The experimental protocol for priming (TE) experiments. Tam, tamoxifen. **b**, Cumulative tumour burden by intestinal region in control (ENU only) mice. Proximal and distal SI are defined as the first 20 cm from the pyloric sphincter and the remaining length, respectively. **c**, Intestinal tumour burden and survival after ENU treatment in control mice. Inset: cause of death. HEP, humane end point; HEP(L), presence of lymphoma/leukaemia at the humane end point. **d**, Kaplan–Meier plot of control and primed cohorts ($Kras^{G12D}$, $Trp53^{null}$, $Fbxw7^{null}$) with and without ENU treatment. Mdn, median. **e**, The mean ± s.d. tumour counts per mouse for each priming event in relation to the median survival after ENU treatment. **f**, Maximum-likelihood estimates from the dN/dS ratio analysis of tumour nonsense and missense mutations identified by hybridization capture. Genes under positive selection with a significant increase in intestinal tumour burden compared with the

$q$ value for all substitutions of <0.25 are labelled. **g**, Oncoprint for genes under positive selection in the control and primed groups. Synonymous mutations are excluded. SI10, SI20, SI30, SI40 and SI50 refer to tissue samples taken at 0–10, 10–20, 20–30, 30–40 and 40–50 cm distance from the pyloric sphincter. **h**, The proportions of tumours in each cohort containing detected $Apc$ or $Ctnnb1$ mutations. **i**, The proportions of monoclonal (presence of one or two $Apc$-truncating mutations or a single $Ctnnb1$ exon-3 mutation) and polyclonal tumours across cohorts for $Apc$- and $Ctnnb1$-driven tumours. **j,k**, The fold change in $Apc$-driven (**j**) or $Ctnnb1$-driven (**k**) tumours relative to the control. $P$ values were calculated using log-rank tests (**d**), a two-sample test of equality of $Apc$ proportions with continuity correction (d.f. = 1; two sided) (**h**) and two-sided Fisher's exact tests (**i**).

control mice. In mice primed with $Apc^{het}$, $Kras^{G12D}$, $Trp53^{het}N1\text{-}ICD^{het}$ and $Trp53^{null}N1\text{-}ICD^{het}$, tumours were observed in the colon, as well as in the SI (Extended Data Fig. 2e and Supplementary Table 1). Most SI and colonic tumours were adenomatous polyps, and local invasion was

a feature in 10%, 38% and 40%, respectively, of $Kras^{G12D}$, $Trp53^{null}$ and $Pten^{null}$ mice; metastases were less common (Extended Data Fig. 2d–f). These ENU-mutagenized lines showed different median survival times that were inversely correlated with the mean intestinal tumour multiplicities, spanning three orders of magnitude (Fig. 1e).

## Apc and Ctnnb1 mutations drive tumours

We next identified the driver events caused by ENU mutagenesis (Extended Data Fig. 1c). First, 239 excised tumours from ENU-treated WT (control), and $Kras^{G12D}$-, $Trp53^{null}$-, $Trp53^{het}N1$-$ICD^{het}$-, $Pten^{null}$- and $Fbxw7^{null}$-primed cohorts underwent next-generation sequencing using a hybridization capture panel that covered the exons of 585 cancer-relevant genes (Supplementary Table 3). Overall, 11,122 somatic mutations were identified (median read depth, 344; Extended Data Fig. 3a–e). Tumours from primed and control mice had similar mutation burdens (around 6 single-nucleotide variants (SNVs) per Mb), except for those arising on the $Trp53^{null}$ background that had had a significantly higher mutation burden (mean (s.d.): 9.9 (3.1) SNVs per Mb) and colonic, but not SI, tumours on the $Trp53^{het}N1$-$ICD^{het}$ background had a significantly lower mutation burden (mean (s.d.): 4.0 (1.1) SNVs per Mb) (Supplementary Fig. 2), probably reflecting differential survival of transforming events that would be eliminated through p53-mediated apoptosis in WT mice.

Assessment of the selection coefficients of ENU-induced mutations using the dNdScv method[16] confirmed that nonsense mutations in Apc and missense mutations in Ctnnb1 were positively selected (q values close to zero; Fig. 1f, Supplementary Table 4 and Supplementary Table 5), identifying these mutations as tumour drivers. There was a small but significant difference in the mutation burden (mean (s.d.): 7.2 (2.5) SNVs per Mb for Ctnnb1-mutant tumours versus 6.4 (2.6) SNVs per Mb for Apc-mutant tumours; Supplementary Fig. 2a–e). A further 205 tumours underwent targeted amplicon sequencing (median read depth, 6,010) and a subset of 81 tumours that was sequenced using the hybridization capture panel underwent additional shallow whole-genome sequencing (Extended Data Figs. 1 and 4 and Supplementary Fig. 3). These analyses confirmed that tumours arising from priming followed by ENU-induced mutagenesis were dominated by single-base mutations of either Ctnnb1 or Apc (Supplementary Fig. 3).

While each priming event promoted tumour development by both Apc and Ctnnb1 mutations, the ratios varied. For example, around 90% of the tumours arising in ENU-treated $Apc^{het}$-primed mice contained at least one Apc mutation, as did around 60% of tumours from control or $Pten^{null}$-primed mice (Fig. 1h and Extended Data Fig. 3g). However, in mice primed with other mutations, the proportion of tumours with Ctnnb1 driver mutations ranged from around 53% in $Trp53^{null}$-primed mice to about 85% in $Kras^{G12D}$-primed mice. Applying these driver proportions to overall intestinal tumour counts revealed that, for example, $Kras^{G12D}$ priming resulted in 596-fold and 58-fold increases in the number of Ctnnb1- and Apc-driven tumours, respectively, relative to the control cohort (Fig. 1j,k). Tumours in most groups were predominantly monoclonal, as indicated by the presence of one or two APC-truncating mutations or a single Ctnnb1 exon-3 mutation (Fig. 1i). By contrast, around 55% of tumours arising in $Apc^{het}$-primed mice were polyclonal, as described previously[2].

## Priming shapes Ctnnb1 mutations

Assessment of 348 Ctnnb1 driver mutations from 295 tumours revealed that almost all were missense mutations, clustering in seven codons within exon 3 (Fig. 2a). The corresponding section of protein contains docking and phosphorylation sites for GSK-3β and casein kinase-1 that promote ubiquitination by E3 ligase β-TrCP and degradation of β-catenin by the proteasome[17–21]. The p.G34E and p.T41I missense mutations accounted for around 41% of all Ctnnb1 mutations; tumours with p.G34E mutations also had a higher overall mutation burden compared with tumours with other exon-3 mutations (Supplementary Fig. 2f–h). However, while p.G34E predominated in tumours from $Kras^{G12D}$- and $Trp53^{null}$-primed mice, p.S37F and pD32G were more frequent in the $Fbxw7^{null}$-primed cohorts (Fig. 2a,b). This suggests that, even among priming events that favour transformation through

Ctnnb1 stabilization, distinct mutational outcomes are selectively favoured (Fig. 2a,b).

It is unclear whether high representation of any given driver mutation is a consequence of strong positive selection versus a high underlying event rate. To assess the latter possibility, the probability of a mutation being induced was derived for each trinucleotide context through analysis of all mutations called in all tumours sequenced, using hybridization capture for genes with no evidence for positive selection (Fig. 2c). This analysis revealed a broad mutational signature enriched for C to T, T to A and T to C changes. The two most favoured trinucleotide contexts (ACC to ATC, and TCC to TTC) were commonly mutated in Ctnnb1, generating the p.T41I and p.G34E mutations. Overall, mutation of eight trinucleotide contexts generated 14 different trinucleotide outcomes and accounted for all 21 observed exon-3 mutations across the 295 tumours analysed (Extended Data Fig. 5a,b). This mutational signature was then used to compare the observed Ctnnb1 exon-3 mutations to those that would be expected in the absence of selection (Fig. 2d,e and Extended Data Fig. 5a–c). This analysis confirmed that the high representation of p.T41I and p.G34E mutations arose from a high susceptibility to mutation, although the former was significantly under-represented compared to expectation (Fig. 2d). By contrast, the p.I35S (ATC to AGC) mutation that accounts for 8% of all Ctnnb1 mutations was fourfold over-represented (observed/expected (O/E) ratio of 4) compared with the expected mutation pattern. These results establish that the distribution of tumour drivers arising from exon-3 mutations of Ctnnb1 is shaped by a combination of underlying mutational preference and positive selection.

The O/E ratios were next used to characterize the selection landscape of Ctnnb1-driven transformation. Control and $Trp53^{het}N1$-$ICD^{het}$-primed tumours showed a preference for transformation by p.I35S, with O/E ratios of around 14 and 11, respectively, while several cohorts displayed a bias against p.D32N (Fig. 2e). Two mutations (p.S33P and p.S37P) that share a trinucleotide context and an amino acid substitution that blocks Gsk3β phosphorylation[18,22] had different mutational patterns: p.S33P was exclusive to the $Kras^{G12D}$ cohort (found in 10 out of 147 Ctnnb1-mutant tumours), while p.S37P had a broad distribution and positive O/E ratio (Fig. 2a,c–e). Tumours from $Kras^{G12D}$-primed mice showed the smallest range of O/E ratios; combined with the high tumour multiplicity observed in these mice, this finding indicates that expression of $Kras^{G12D}$ creates a broadly permissive transformation landscape for Ctnnb1-activating mutations. Overall, these results suggest that different exon-3 mutations have different transformation potentials that are dependent on the nature of the priming field.

## Priming shapes Apc mutations

Analysis of Apc driver mutations revealed that nonsense and, to a lesser extent, splice mutations were widely distributed over the N-terminal region (Fig. 3a). Excluding tumours primed by Apc heterozygosity, most tumours (59%) contained two Apc mutations, 33% had one, and 8% had three or four (Extended Data Fig. 3g); loss of heterozygosity (LOH) was also observed in some tumours with a single mutation (Extended Data Fig. 4d,e). We assigned each Apc mutation to one of eight spatial bins (A–H) to determine whether their distribution depended on the priming event (Fig. 3a). C-terminal bins F–H contained few mutations and were not considered further. For each priming group with at least 30 tumours, binned mutations were assessed for selection biases by summing the probabilities of individual truncating mutations by bin and calculating O/E ratios (Fig. 3b,c and Extended Data Figs. 6a–c and 7a). Mutations in bin A, the most N-terminal region, and bin E, which contains the 20-amino-acid repeats that bind to and target β-catenin for degradation, were under-represented, whereas mutations in bins B and D were over-represented (Fig. 3c). Notably, when analysed by priming group, $Pten^{null}$-primed and $Apc^{het}$-primed mice both showed a significant under-representation of mutations in bin A—that is,

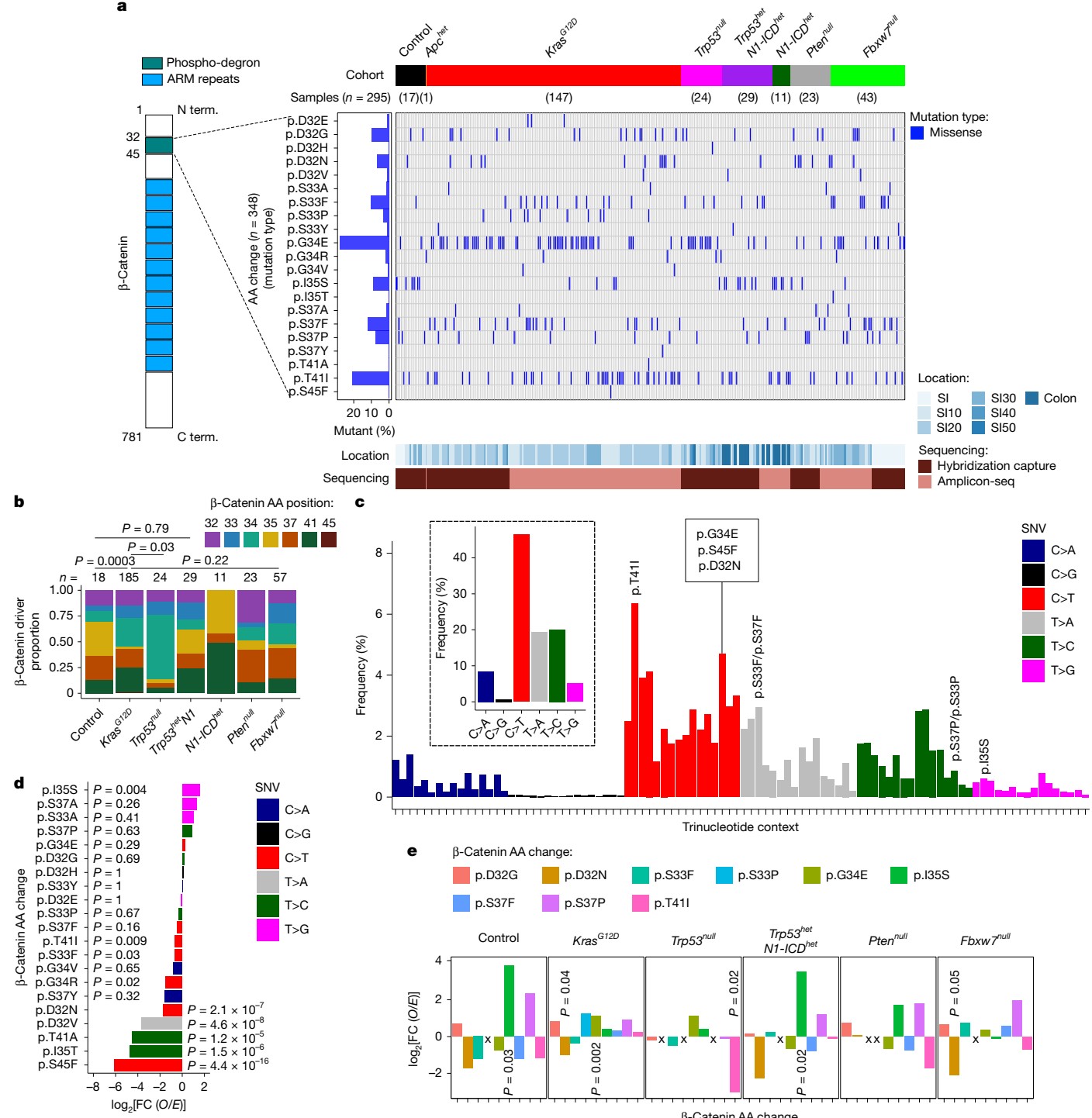

**Fig. 2 | Priming selects for different mutational patterns in *Ctnnb1*.**
**a**, Oncoprint of *Ctnnb1* missense mutations located in the phospho-degron motif (*n* = 348 out of 295 tumours screened). The percentage of each amino acid (AA) changed is shown. Term., terminus. **b**, The proportion of β-catenin amino acid position changes for each cohort. **c**, Mutation signatures depicted as the frequency of different SNVs by trinucleotide context (not shown) with the corresponding β-catenin amino acid change. Inset: mutation signatures as the overall frequency of different single-nucleotide changes. **d**, The *O/E* ratios

of *Ctnnb1* exon-3 mutations from primed cohorts in relation to amino acid changes. **e**, The *O/E* ratios of recurrent (*n* > 9) β-catenin driver amino acid changes for each selected cohort. Amino acid changes with no mutations are marked with an 'X'. *P* values were calculated using two-sided Fisher's exact tests (**b**) and $\chi^2$ tests (d.f. = 1) for each mutation (**d**,**e**). Only statistically significant *P* values are displayed in **e**; exact *P* values for all datapoints are provided as source data.

those resulting in the shortest proteins, which are truncated before the first Armadillo repeats that mediate the interactions of Apc with multiple proteins (Fig. 3d). *Apc*<sup>het</sup>-primed mice also exhibited enrichment of mutations within the Armadillo domain (bin B), while *Pten*<sup>null</sup>-primed

mice showed a marked depletion in bin E. By contrast, the control, *Trp53*<sup>null</sup> and *Kras*<sup>G12D</sup> cohorts displayed a closer alignment between *O/E* ratios, suggesting weaker selection pressures on the N-terminal half of *Apc* in these contexts (Fig. 3d).

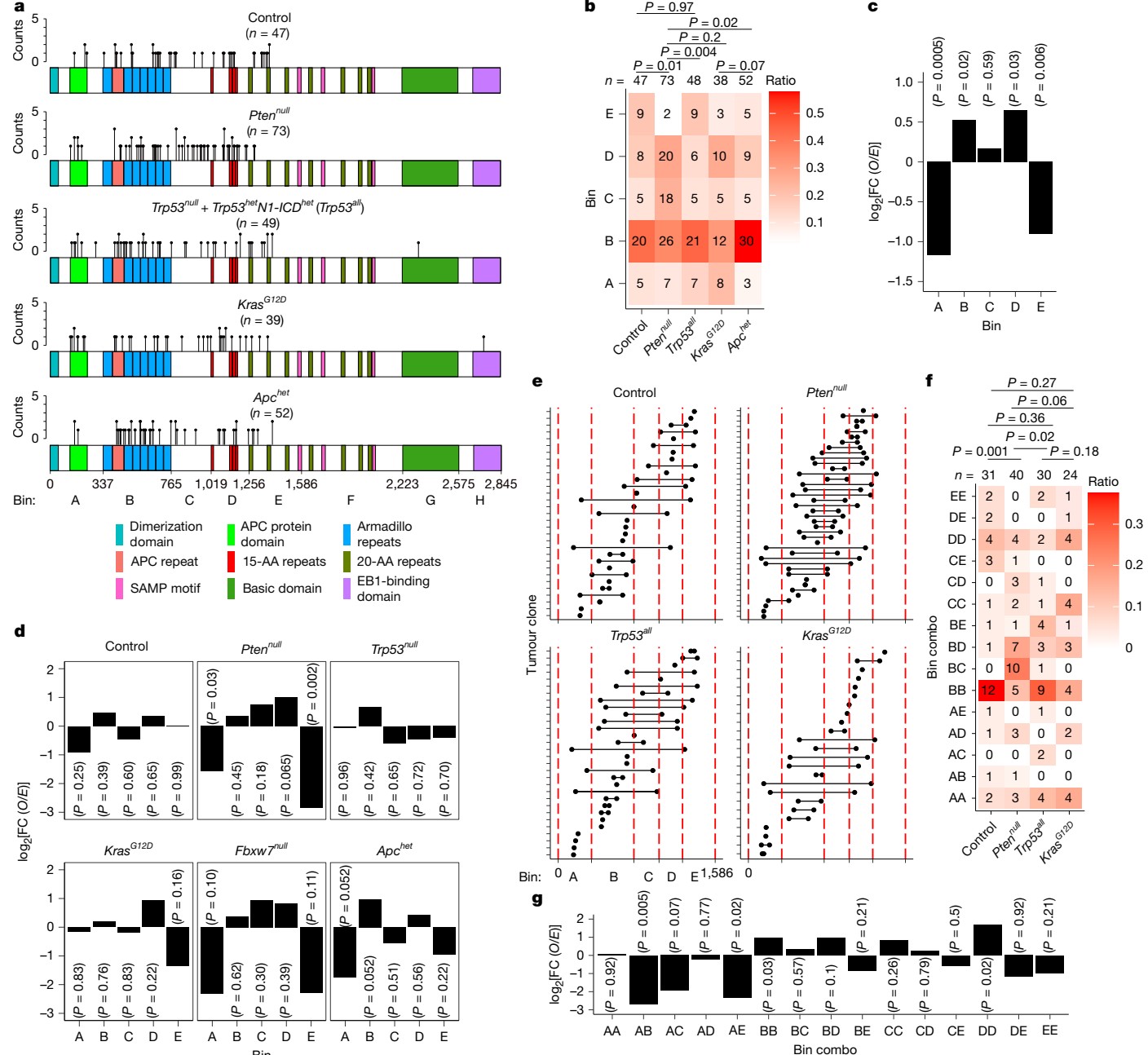

**Fig. 3 | Priming selects for different mutational patterns in *Apc*. a**, The positions of APC protein-truncating mutations in relation to the domain bin. **b**, The APC domain bin relative abundance from data in **a** for each cohort, with counts for each bin shown. **c,d**, The *O/E* ratios in relation to the APC domain bin for the primed cohorts combined (**c**) or individually (**d**). **e**, The truncation positions in APC for each tumour or clone, and linked N- and C-terminal truncation locations from cases in which different mutations are paired.

Single points are from tumours with one mutation of *Apc* with evidence of LOH. **f**, Heat-map representation of bin combination (combo) relative abundance from data in **e**, with counts for each bin combination shown. **g**, The *O/E* ratios in relation to the APC domain bin combinations for primed cohorts. *P* values were calculated using two-sided Fisher's exact tests (**b** and **f**) and $\chi^2$ tests (d.f. = 1) for each mutation bin or bin combination (**c**, **d** and **g**).

Mutation of both *Apc* alleles is required for transformation. However, there is evidence that the two hits are co-dependent, such that some residual functionality is required for transformation[23]. To investigate possible co-dependency in our model, co-occurring *Apc* mutations were assigned to the appropriate bin and considered co-ordinately (Fig. 3e,f). Both alleles from tumours with a single *Apc* mutation due to presumptive LOH were assigned a single common bin (Extended Data Fig. 4d,e). This analysis confirmed a significant trend to under-representation of the most N-terminal (bin A) truncations. By contrast, truncations occurring in bins B, C, and D remained

over-represented, except when co-occurring with mutations in bins A or E (Fig. 3g).

Within-group analysis revealed that control mice and the combined *Trp53^null^*- and *Trp53^het^N1-ICD^het^*-primed groups tended to enrichment of combinations of mutations that both fell within bin B (BB combinations; Fig. 3f). By contrast, *Pten^null^*-primed tumours exhibited a notable distal shift from BB combinations that resulted in retention of the entire Armadillo repeat domain while still generating loss of the 20 amino acids present in bin E. Armadillo repeat domains mediate APC's role in cell–cell adhesion[24–26]. Combined with previous reports demonstrating strong

synergy between *Apc* and *Pten* deficiencies[27,28], this finding suggests that optimal transformation in the *Pten*[null]-priming context uniquely requires an intact Armadillo domain.

## Decay of *Apc* and *Ctnnb1* mutations

In the absence of priming, conditional transforming events induced by ENU mutagenesis can either be retained or lost. Reasoning that retained events could be identified by rescuing their ability to promote tumour formation post hoc, we reversed the order of treatment such that the pro-oncogenic fields were created by exposing mice to tamoxifen 10–30 days after ENU mutagenesis (ENU–Tam). Tumours developing under rescue conditions were still predominantly driven by *Apc* and *Ctnnb1* mutations (Extended Data Fig. 8).

Tumour multiplicities from rescue experiments were compared to those observed in the corresponding priming (Tam–ENU) experiments (Fig. 4a,b and Supplementary Table 6). Mice rescued by *Kras*[G12D] 30 days after ENU had a mean (s.d.) small intestinal tumour multiplicity of 11 (7) compared with 388 (127) in the corresponding priming protocol. This 97% reduction suggests pronounced negative selection against ENU-induced driver mutations in unprimed tissue (Fig. 4b). Smaller reductions were also observed for other mice. To quantitatively assess selection bias, tumour decay dynamics were compared with the decay dynamics of normal stem-cell-derived clones as they compete for occupancy of the intestinal crypt epithelium, a process that is not subject to selective pressures[29–31] (Fig. 4c), having first confirmed that proliferative indices and clone sizes were not altered by previous ENU treatment (Supplementary Fig. 4). Significant negative bias was observed against most *Ctnnb1* mutations (except I35 mutations) in *Kras*[G12D]-rescued conditional transformants, with these mutations largely absent from tumours rescued 30 days after ENU treatment (Fig. 4d–f). However, many of the tumours rescued 10 days after ENU treatment still contained *Ctnnb1* mutations. Further analysis revealed that different *Ctnnb1* exon-3 mutations were lost with different efficiencies in *Kras*[G12D]-rescued mice (Fig. 4e,f; in order of descending probability of loss: p.S33, p.G34, p.D32, p.S37, p.T41 and p.I35).

Transforming mutations in *Apc* also decayed over time, with apparent negative selection bias against these mutations in *Kras*[G12D]-rescued mice (Fig. 4d). We assessed differential decay of *Apc* mutants affecting different parts of the protein (bins A–E, as above), normalized to the initial distribution derived with priming (Fig. 4g). No significant deviations from neutral decay dynamics were identified for *Apc* mutants in *Kras*[G12D]- and *Trp53*[null]-rescued mice in this stratified analysis, potentially due at least in part to data sparseness (Fig. 4g). However, when assessing the decay of binned *Apc* mutations in aggregate for mice rescued by *Kras*[G12D], *Trp53*[null] and *Fbxw7*[null] (KTF; Extended Data Fig. 9b), the data suggested that there was negative bias against more N-terminal mutations, with a significant decay for bin B mutations in mice rescued 10 days after ENU treatment and a trend for bin A negative decay at 30 days that did not reach significance (Fig. 4g). We next analysed co-occurring *Apc* mutations. The decay dynamics of mutations in each bin were compared with those of all of the other bins, excluding cases in which two mutations in the same bin co-occurred (Extended Data Fig. 9d–f). This further analysis revealed significant negative decay for N-terminal bins A and B, supporting selective bias against mutations in these N-terminal regions but broadly neutral decay dynamics for bin C–E mutations.

Rescue in *Apc*[het] mice occurs through Cre-mediated truncation of the fourth Armadillo repeat (bin B, position 580), providing a second hit in cells with ENU-induced single-allele mutations of *Apc*. When applying the second hit 30 days after ENU treatment, the mean (s.d.) tumour multiplicity was 109 (32) compared with 967 (267) in the primed condition—a 90% reduction. Strong negative selection was confirmed for mutations in bins B–D (Fig. 4b,h and Extended Data Fig. 9c), suggesting that these monoallelic truncations confer a gain of function

that acts dominantly to confer a negative bias that eliminates affected cells. By contrast, mutations in bins A and E exhibited neutral decay. Notably, the increased persistence of bin A mutations may explain their enrichment in polyclonal tumours due to a clonal recruitment process[2]. Overall, these differences suggest that the transforming potential of *Apc* mutations in different regions of the gene scales with increasing negative bias (Fig. 4i).

## Effects of mutational context and order

While the rescue experiments revealed evidence of negative selection against certain *Apc* and *Ctnnb1* mutations in *Kras*[G12D]-rescued mice, in *Fbxw7*[null]-rescued mice, the experiments suggested an epistatic effect. When collectively assessing all mutations in *Apc* or *Ctnnb1* in these mice, we observed a departure from the expected neutral decay dynamics for *Ctnnb1* but not *Apc* (Fig. 4d and Extended Data Fig. 8a–c). Although eventual loss of *Ctnnb1*-mutant transformants was observed in mice rescued at 30 days after ENU treatment, mice rescued at 10 days had an increased tumour burden compared with that observed with initial *Fbxw7* priming (mean 62 versus 53 tumours per mouse; Extended Data Fig. 8b). Incorporating consideration of the frequency distribution of the different affected codons in exon 3 revealed that this effect was largely due to p.I35 mutations being over-represented eightfold compared with the expected frequency at 10 days after ENU treatment (Fig. 4e,f). This finding indicates that the order in which these mutations occur dictates outcome, with a previous *Fbxw7* mutation protecting against transformation by *Ctnnb1* p.I35S mutations but subsequent loss of *Fbxw7* promoting tumour formation in cells with an existing *Ctnnb1* p.I35S mutation.

Similar effects were noted for a subset of *Apc* mutations in this background: transformants with mutations in bins B–E underwent neutral decay, but mutations in bin A were significantly over-represented when rescued at 10 days after ENU treatment (Fig. 4g and Extended Data Fig. 9a). As with the p.I35S mutations in *Ctnnb1*, this suggests that subsequent loss of *Fbxw7* promotes tumour formation by a subset of transformants with existing N-terminal truncations of APC. These observations are consistent with previous reports of epistatic relationships between *Fbxw7* and *Apc*, in which the loss of *Fbxw7* creates an altered epigenetic state that protects against subsequent *Apc*-mediated transformation[32].

## Evidence of priming in human CRCs

To determine whether there is evidence for similar priming events that may impact APC function in human CRC, we obtained genomic data from 17,000 CRCs from the AACR Project GENIE cancer registry and filtered the dataset to include only cancers with two APC-truncating mutations (Fig. 5a). To approximate the approach used in our mouse experiments, the distribution of *APC* mutations in these tumours was first compared with that expected from known mutational signatures of human epithelium[33]. Mutations from individual cancers and the relative frequencies of all expected truncations arising from the known signatures were mapped to their locations within the *APC* gene (Fig. 5b). The observed mutations showed a broad distribution and close correspondence to known recurrent CpG to TpG transitions associated with clock-like mutational signature SBS1 (refs. 34,35). Using the same binning boundaries as in the mouse model, assessing the $\log_2$-transformed fold changes ($\log_2$[FC]) in O/E ratios revealed strong mutation enrichment in bin E ($\log_2$[FC] = 2.1). This high number of mutations arose despite this mutation cluster region[36] of the human *APC* gene being a poor sequence context for CpG to TpG mutations (Fig. 5c). By contrast, mutations in the Armadillo domain (bin B) were strongly under-represented ($\log_2$[FC] = −1.7) despite this region being highly susceptible to CpG to TpG mutation (Fig. 5c). This latter result is consistent with the negative selection bias against mutations in the

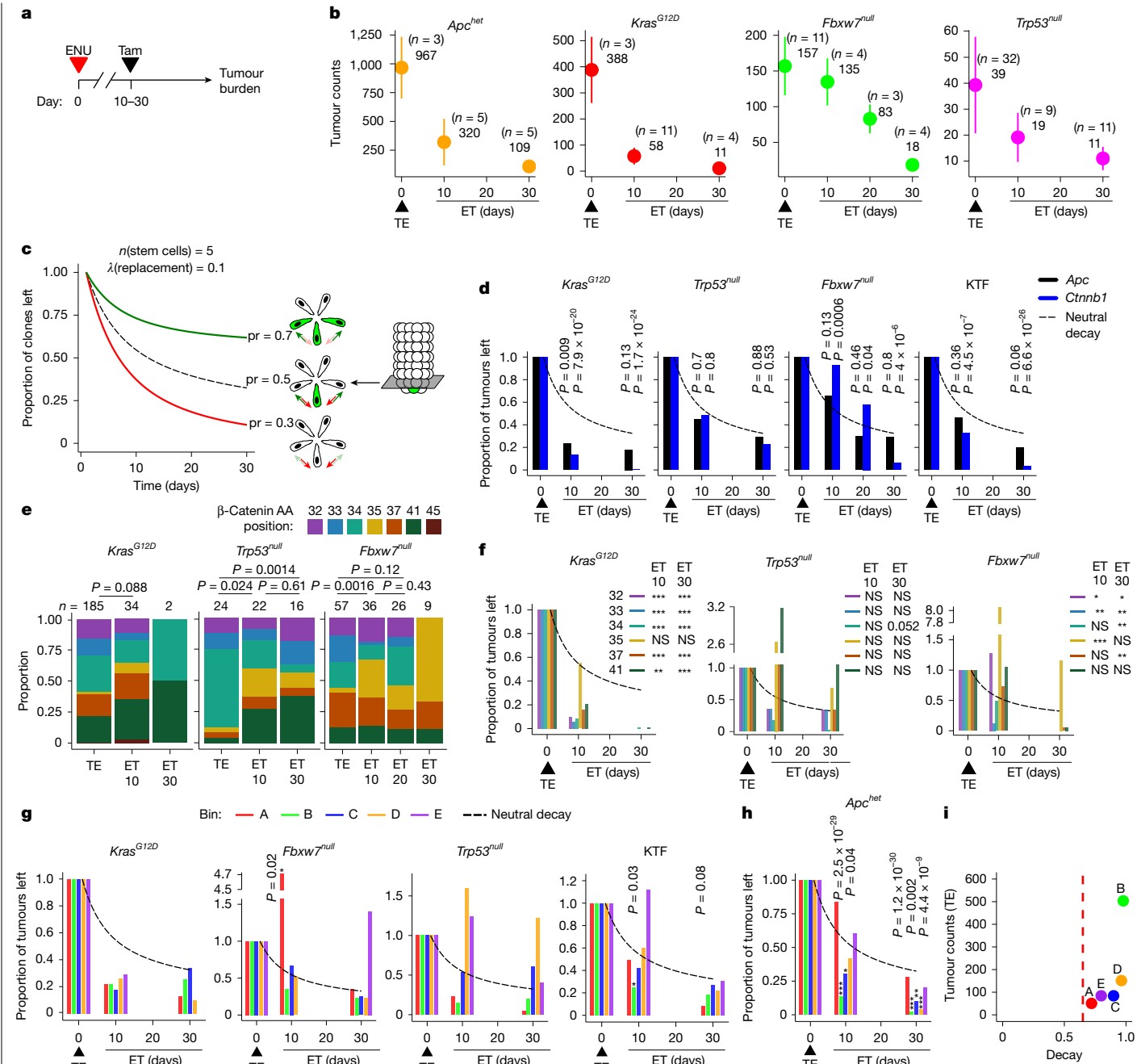

**Fig. 4 | Negative selection for conditional drivers confirmed by transformant rescue. a**, Experimental outline of the rescue (ET) protocol. **b**, The mean ± s.d. SI tumour burden for the priming (TE) and rescue (ET) protocols. Priming data are plotted at x = 0. **c**, Simulations of clonal decay curves for the proximal SI, complemented with a schematic, with fixed parameters for the number of stem cells (*n*) and replacement rate (*λ*), and varying the probability of replacement (pr), modelling neutral (pr = 0.5), positive (pr = 0.7) and negative (pr = 0.3) biases. **d**, The proportion of *Apc*- or *Ctnnb1*-driven tumours remaining in the rescue protocol relative to the priming protocol (set at y = 1 on day x = 0) for each cohort. The bars represent the observed tumour proportions, and the dashed lines indicate the expected neutral decay rate. **e**, β-Catenin driver amino acid position proportions from the *Kras^G12D^*, *Trp53^null^* and *Fbxw7^null^* cohorts. **f**, The proportion of tumours with driver β-catenin mutations

remaining in the rescue protocol relative to the priming protocol for each cohort. *P* values for each mutation, relative to the rescue timepoint, are displayed next to the plot. **g,h**, The proportion of tumours remaining in the rescue protocol per APC bin relative to the priming protocol for each cohort. **i**, The tumour counts per *Apc* bin in the *Apc^het^* (TE) protocol in relation to its decay at 30 days (ET30). Decay was calculated from **h** (1 − proportion of tumours left). The dashed line indicates neutral decay. *P* values were calculated using two-sided Fisher's exact tests (**e**) and χ² tests (d.f. = 1) for each rescue timepoint, for each β-catenin mutation or APC bin (**d** and **f**–**h**); *P < 0.05, **P < 0.01, ***P < 0.001. Only statistically significant or borderline *P* values are displayed in **g** and **h**; exact *P* values for all datapoints for **f**, **g** and **h** are provided as source data.

corresponding region of *Apc* that was observed in the mouse rescue experiments.

Mutation of *Kras* was the most efficient priming event in our mouse model, and *KRAS* is a major driver of human CRC[37,38]. Thus, we next

examined APC domain retention in CRCs with both *APC* and *KRAS* mutations to assess whether the latter might influence the selective landscape of APC truncations. Assessing the positions of both N- and C-terminal mutations in *APC* revealed that CRCs lacking KRAS-activating

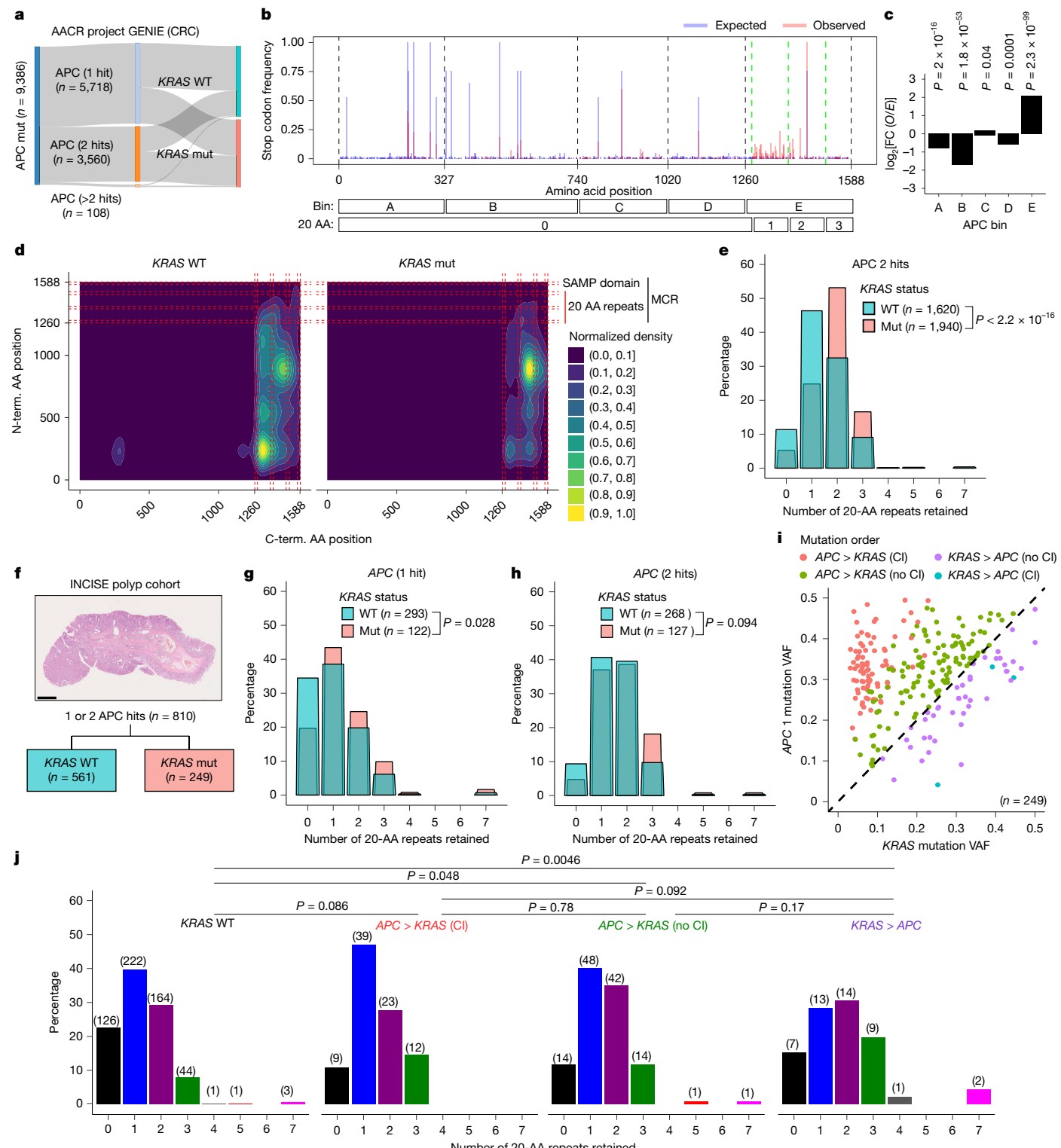

**Fig. 5 | Selection and priming shape APC mutation patterns in human CRC.**
**a**, Overview of GENIE CRC samples by *APC* and *KRAS* mutation status. **b**, The expected frequency for each potential *APC* stop, assuming an SNV, superimposed with observed APC mutations (two-hit tumours with truncations arising from SNVs; *n* = 952) from the GENIE cohort in the APC N-terminal half (amino acids 1–1,588). Frequencies were normalized to the highest expected (multiple sites) and observed (amino acid 1,450) sites. The black dashed lines demarcate domain bins. The green dashed lines separate the 20-amino-acid repeat domain (20 AA) retention boundaries. **c**, The *O/E* ratios in relation to the APC domain bin. **d**, The normalized density for APC N-terminal and C-terminal mutations in relation to *KRAS* status. **e**, The relationship between APC 20-amino-acid repeat retention and tumour sample percentage, stratified by *KRAS* status in samples with two *APC* hits. **f**, H&E-stained cross-section of an INCISE cohort polyp and an overview based on *APC* and *KRAS* mutation status. Scale bar, 2.5 mm. **g**,**h**, The relationship between APC 20-amino-acid repeat retention and tumour sample percentage, stratified by *KRAS* status for samples with one (**g**) or two (**h**) *APC* hits. **i**, The VAF of the highest-ranked *APC* mutation (APC 1) and *KRAS* mutation per sample. The colours indicate VAF-based hierarchy. CI, confidence interval. **j**, The relationship between APC 20-amino-acid-repeat retention and tumour sample percentage, stratified by *KRAS* status and VAF-based hierarchy for *KRAS*-mutant samples. *P* values were calculated using two-sided Fisher's exact tests (**e**, **g**, **h** and **j**) and $\chi^2$ tests (d.f. = 1) (**c**).

mutations commonly retained one 20-amino-acid repeat, while those with KRAS-activating mutations tended to retain two (Fig. 5d,e). This distribution reflected a major contribution from Arg1450 truncations, which were strongly associated with KRAS-activating mutations, occurring in 168 out of 509 (33%) KRAS-mutant tumours compared with in 51 out of 443 (11.5%) KRAS WT tumours. The trend of 20-amino-acid-repeat retention was also a feature of CRCs with only a single APC-truncating mutation and was not influenced by the specific KRAS mutation, or by other driver mutations (except for AMER1, which exclusively retained two or three 20-amino-acid repeats irrespective of KRAS status; Extended Data Fig. 10a–d). Notably, right-sided CRCs, which have a higher incidence of KRAS-activating mutations, have previously been associated with a similar tendency to retain more APC 20-amino-acid repeats[39]. To discount the possibility of the observed effects arising from a sidedness representation bias, the impact of location was considered, establishing that the observed effect is a feature of left-sided CRCs (Extended Data Fig. 10e,f).

We next identified a similar (albeit more modest) trend to increased retention of 20-amino-acid repeats as an early acquired feature of preneoplasia, as shown by analysis of 810 polyps with one or two APC mutations from the INCISE polyp cohort (Fig. 5f–h). Data were derived from a single transecting section across the polyp (Fig. 5f) that minimized subsampling effects. This factor motivated assessment of the variant allele frequency (VAF) values for KRAS and APC mutations in this cohort, with higher values probably representing earlier events. Ranking revealed a close correspondence in VAFs, with only 86 out of 249 polyps revealing a mutation chronology with 95% confidence: 83 of these polyps had VAF values consistent with an APC-to-KRAS mutation chronology, and three were consistent with a KRAS-to-APC mutation chronology (Fig. 5i). Despite this assignment limitation, among all 810 polyps, a trend of increased 20-amino-acid-repeat retention resembling that observed in mature CRCs was observed as the KRAS/APC VAF ratio increased, suggesting that polyps with higher relative KRAS VAFs (and therefore potentially earlier KRAS mutations) may be at a higher risk of progression (Fig. 5j), consistent with priming dictating mutational order and that primed polyps are at higher risk of progression.

## Discussion

Permissive fields promote transformation by Ctnnb1 and Apc mutations that are, respectively, poor or competent tumour initiators in a WT context. The generalized trend for priming to promote transformation by Apc mutations is associated with features reminiscent of the just-right model of human CRC development[10]. This hypothesis initially proposed that the optimal amount of WNT pathway dysregulation for survival and transformation is achieved by combinations of APC mutations that retain some residual ability to bind CTNNB1 and regulate the pathway[10]. More recently, this concept has been extended to encompass WNT-independent functions of APC, including the modulation of cell adhesion, directional migration and polarity (reviewed previously[40,41]). The current findings provide the additional insight that the transformation landscape encompasses strong negative biases against single-allele mutations that are potent drivers of tumour initiation but require a permissive environment to transform. There is substantial evidence that APC mutants in CRC have gain-of-function properties, ascribed to their ability to oligomerize with WT APC[40,41]. However, in contrast to previous reports suggesting that these mutations promote tumorigenesis through increased cell survival and induction of aneuploidy, the current findings indicate that their effects also include impaired survival of mutant cells.

In our mouse model, truncations within the Armadillo repeats (residues 337–765), an essential scaffold for protein–protein interactions involved in both WNT-dependent and WNT-independent processes[24,25,42,43], were the most frequent transforming event and showed the strongest negative bias. By contrast, N-terminal truncations before

residue 337 also showed negative bias. These latter mutations are rarely found as driver events but have recently been shown to contribute to polyclonal tumours due to recruitment by stronger driver events[2]. Tumour clones containing such mutations were characterized as WNT[high] and stem cell enriched. The current findings confirm that these WNT levels are too high to be 'just right'.

Recent studies have ascribed a supercompetitor behaviour to clones arising by biallelic inactivation of Apc, whereby growth advantage is conferred, in part, by inhibition of self-renewal in neighbouring WT cells[7,8]. Notably, such analyses focus on the growth properties of surviving clones and do not directly address their overall survival probability. Considered together, these observations suggest that mutations conferring the greatest potential for transformation have the highest bar to survival, with priming acting rheostatically to lower the latter.

In human CRC, it is notoriously difficult to deconvolute the order of early clonal driver events such as APC and KRAS mutations from sequencing data derived from bulk tumour samples. However, we found evidence that the distinct selection landscapes of APC-mutant CRCs with and without co-occurring KRAS-activating mutations result in distinct distributions of APC mutations. Specifically, the retention of more APC 20-amino-acid repeats in CRCs that contain both APC and KRAS mutations is a strong indication of priming. This shift in APC truncation sites in the context of KRAS mutations leads to the retention of a region spanning the second 20-amino-acid repeat and the adjacent β-catenin inhibitory domain, of which the retention markedly reduces WNT signalling[44]. This effect could also relate to mutual dependency in the turnover of both β-catenin and KRAS through a common mechanism of GSK3β-mediated phosphorylation and/or USP7-mediated deubiquitinition[45–49]. In CRC, APC mutations that co-occur with KRAS mutations may either be primed by previous KRAS activation or, conversely, favour the selection of subsequent activating KRAS mutation.

The known repertoire of positively selected events associated with oncogenesis in normal tissues is increasing[50–53]. For example, haploinsufficiency in ARID1A or PTEN is sufficient to induce positive biases in the human colon[54], and stable heritable epigenetic changes or those accompanying local inflammation may also create pro-oncogenic fields[55–59].

Overall, our study suggests that the ability of WNT-dysregulating mutations to confer a wide dynamic range of fitness outcomes and transforming potencies renders them uniquely well suited to create a sweet spot for transformation in diverse predisposing fields in the human colon. This paradigm may explain many aspects of the natural history of tumour evolution, including the ubiquity of APC loss in CRC—whether APC loss occurs as a primary event or a secondary event, it can still enable tumour growth as a single 'Big Bang'-type clonal expansion[60].

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

# Methods

## Mouse models

The intestinal epithelium-specific *Villin-cre^ERT2* (ref. 61) mouse line was intercrossed with various conditional tumour driver models: *Kras^(lsl)G12D/WT* (*Kras^G12D*)[62], *Pik3ca^(lsl)H1047R/WT* (*Pik3ca^H1047R*)[63], *Apc^Δ14/WT* (*Apc^het*)[64], *Trp53^Δ2-10/Δ2-10* (*Trp53^null*)[65], *Trp53^Δ2-10/WT* (*Trp53^het*), *Pten^Δ5/Δ5* (*Pten^null*)[66], *Fbxw7^Δ5/Δ5* (*Fbxw7^null*)[67], *R26-Notch1^ic/WT* (*N1-ICD^het*)[68], *Arid1a^Δ8/Δ8* (*Arid1a^null*)[69], and R26-CAG-Brainbow2.1/Confetti (*R26^(lsl)Confetti/WT*)[70,71]. Lgr5-DTR-eGFP[72] mice were used to study the effects of ENU mutagenesis on Lgr5-positive cells. Mice were from mixed backgrounds with high C57Bl/6J inheritance. Genotyping was done by Transnetyx.

## Animal husbandry

All experiments used male and female mice of at least 6 and 8 weeks of age, respectively. Mice were housed under controlled conditions (temperature $21 \pm 2$ °C, humidity $55 \pm 10$%, under a 12 h–12 h light–dark cycle) in individually ventilated cages in a specific-pathogen-free facility (tested according to the recommendations for health monitoring by the Federation of European Laboratory Animal Science Associations). Food and water were provided ad libitum. None of the mice were involved in any procedure before the study. For survival curve generation, mice were aged until they showed clinical signs of tumour burden (anaemia, hunching and loss of body condition). All of the animal experiments were performed in accordance with the guidelines of the UK Home Office under the authority of project licence PD5F099BE, approved by the Animal Welfare and Ethical Review Body at the CRUK Cambridge Institute, University of Cambridge.

## Field induction and ENU mutagenesis

Complete intestinal field induction was achieved by a single intraperitoneal injection of 4 mg (3 mg for mice <20 g in weight) of tamoxifen (Merck T5648) dissolved in ethanol and sunflower oil (Merck S5007) at a 1:9 ratio (20 mg ml⁻¹ stock solution). A lower tamoxifen dose (0.15 mg) was used for *Kras^G12D* mice in the sequencing cohort to reduce tumour multiplicity and therefore random tumour collisions. We resuspended ENU (Merck N3385) in 5 ml 95% ethanol/45 ml phosphate/citrate buffer at a final concentration of 20 mg ml⁻¹ (ref. 73) and injected it intraperitoneally at 200 mg per kg.

## Mouse dissection

Survival data and most tumour counts were generated from mice culled at the humane end point. Timed culls before this end point were used to assess tumour initiation kinetics in the control cohort and to allow tumour counts in high tumour density models (*Kras^G12D* cohort). The pancreas, mesenteric lymph nodes, liver and lung were dissected and placed in formalin solution overnight at room temperature then exchanged to 70% ethanol for histological assessment of metastasis. Other organs were also collected and processed if relevant pathology was observed during dissection. Intestines were dissected, longitudinally opened, whole-mounted and fixed in 4% PFA overnight at 4 °C in an orbital shaker. Tumours were counted and either excised and processed for genomic DNA (gDNA) extraction or Swiss-rolled for histopathological assessment.

## Intestinal tumour count and microscopy

Tumours in whole-mounted intestines were counted under a dissection microscope. Proximal SI was defined as the first 16–20 cm, comprising SI10 and SI20. Distal SI was defined by the remaining length, comprising SI30, SI40 and, sometimes, SI50. The colon location comprised the caecum and proximal and distal colon. *Apc^het* (SI + colon) and *Kras^G12D* (colon) whole-mounts were sliced into 2–3 cm pieces, optically cleared using the CUBIC1a protocol[74] for 7 days at 37 °C with solution change every other day, stained with DAPI (1:1,000; Thermo Fisher Scientific, D1306), and refractive-index-matched in Rapiclear 1.52 (Sunjin Lab,

152002) for 2 days before mounting in 1 mm iSpacers (Sunjin Lab). Representative tissue pieces for each region were imaged throughout the whole thickness in a Leica SP5 TCS confocal microscope (×10/0.4 NA objective, 8–11 µm z-step, ×1.5 optical zoom). Tumours were manually counted using Fiji image analysis software[75] and the total tumour count was extrapolated.

## Lineage tracing of ENU-treated mice

*Villin-cre^ERT2R26^(lsl)Confetti/WT* mice were treated with ENU (200 mg per kg) or vehicle on day 0 then induced with 1 mg tamoxifen on day 18. Mice were dissected 21 days after tamoxifen and representative 2 cm segments of the proximal SI were collected. Tissue samples were fixed overnight in 4% PFA, washed in PBS, optically cleared, mounted and imaged using a confocal microscope. The width of YFP-positive clones was quantified by calculating the fraction of the crypt circumference (1/8 to 8/8) occupied by YFP. The average clone width per treatment was calculated by multiplying the count of each clone size by the number of crypt fractions occupied (1–8), then dividing by the total number of clones counted.

## Tumour gDNA extraction

Whole PFA-fixed tumours were washed in PBS, excised, cut into smaller pieces and gDNA extracted using the QIAamp DNA FFPE Tissue Kit (Qiagen 56404). In brief, tissue was digested in lysis buffer + proteinase K solution overnight at 56 °C with shaking at 1,200 rpm in a thermocycler (Eppendorf) and then homogenized in QIAshredder columns (Qiagen 79656) before in-column washing and elution in ATE buffer according to the manufacturer's protocol (although the 90 °C incubation step after tissue digestion was omitted to avoid excessive DNA fragmentation). The concentration of the extracted DNA was assessed using a Qubit dsDNA HS assay kit (Thermo Fisher Scientific, Q32854).

## Antibodies

**Primary antibodies.** CTNB1 (0.25 µg ml⁻¹, mouse, 610154, BD Biosciences), O6-ethyl-2′deoxyguanosine (0.5 µg ml⁻¹, rat, SQX-SQM001, Squarix Biotechnology), GFP (10 µg ml⁻¹, chicken, ab13970, Abcam), PTEN (1:300, rabbit, 9559, Cell Signalling), BrdU (2.6 µg ml⁻¹, sheep, ab1893, Abcam), CD326 (EPCAM) AlexaFluor 647 antibody (1:2,000, 118210, BioLegend). Secondary antibodies were as follows: rabbit anti-rat (1:250, A110-322A, Bethyl Laboratories), rabbit anti-mouse (1:1,500, ab125913, Abcam), anti-chicken (1:500, 303-005-003, Jackson Labs) and anti-sheep (1:500, 313-005-003, Jackson Labs).

## Immunohistochemistry

Fixed tissue was paraffin embedded and cut into 3–4 µm sections by the CRUK CI Histopathology Core. Haematoxylin and eosin (H&E) staining was performed using an automated ST5020 Multi-stainer (Leica Biosystems). Staining was performed on Leica's automated Bond-III platform in conjunction with the Polymer Refine Detection System (Leica, DS9800). For CTNB1 and O6-ethyl-2′deoxyguanosine, antigen retrieval was performed in sodium citrate (Leica's Epitope Retrieval Solution 1, AR9961) for 20 min at 100 °C. For GFP and BrdU, antigen retrieval was performed using enzymatic digestion. In brief, enzyme digestion (proteinase K) was run at 37 °C using Leica's Bond enzyme concentrate (AR9551), which contains proteolytic enzyme (17 mg ml⁻¹) and stabilizer. For BrdU, slides were submerged in 2 M HCl for 15 min, following the dewax and rehydration step. Blocking was performed with Protein Block Buffer (X090930-2, Dako). After incubation with the primary antibody, the sections were incubated with the secondary antibody before development and mounting. For CTNB1 staining, an additional mouse-on-mouse blocking step was performed.

## RNA in situ hybridization

Detection of mouse *Trp53* was performed on FFPE sections using RNAscope 2.5 LS Reagent Kit-RED (322150, ACD) and RNAscope LS

2.5 Probe-Mm-*Trp53*-O1 (513008, ACD). In brief, 3-µm-thick sections were baked for 1 h at 60 °C before loading onto a Bond Rx instrument (Leica Biosystems). Slides were deparaffinized and rehydrated on-board before pretreatment using Epitope Retrieval Solution 2 (AR9640, Leica Biosystems) at 95 °C for 15 min and ACD enzyme from the LS Reagent kit at 40 °C for 15 min. Probe hybridization and signal amplification were performed according to the manufacturer's instructions, with the exception of Amp5 (incubation for 30 min). Fast red detection was performed on the Bond Rx using the Bond Polymer Refine Red Detection Kit (DS9390, Leica Biosystems) with an incubation time of 20 min. The slides were then removed from the Bond Rx, heated at 60 °C for 1 h, dipped in Xylene and mounted using EcoMount Mounting Medium (EM897L, Biocare Medical). The slides were imaged on the Aperio AT2 (Leica Biosystems) system to create whole-slide images. Images were captured at ×40 magnification with a resolution of 0.25 µm per pixel.

## Single-cell preparation for flow cytometry

The first 10 cm of proximal SI from tamoxifen-induced mice was flushed in cold PBS, longitudinally opened and cut into 0.5 cm segments. The segments were repeatedly passed through a 10 ml pipette using cold PBS until the solution ran clear, incubated in cold 5 mM EDTA/PBS for 30 min, washed and resuspended in cold PBS. Crypt-enriched fractions were released by five 5 s manual shaking cycles then pelleted and dissociated in 20 ml trypsin (1× 0.05% EDTA, 25300054, Thermo Fisher Scientific) at 37 °C for 7 min with vigorous shaking every minute. After multiple washes in PBS/2% FBS, dissociated cells were resuspended in PBS/2% FBS before incubation with anti-mouse CD326 (EPCAM) Alexa-aFluor 647 antibody and addition of DAPI (10 µg ml$^{-1}$) to distinguish between live and dead cells. Flow sorting was carried out on the BD FACS Aria SORP (BD Biosciences) system using appropriate single-stained and unstained controls (Supplementary Fig. 6).

## Genotyping PCR

Sorted EPCAM-positive and EPCAM-negative cells were pelleted and washed once in cold PBS. Genomic DNA extraction was performed using the ReliaPrep gDNA tissue miniprep system (A2051, Promega) and PCR was performed according to the Q5 Hot Start High-Fidelity DNA Polymerase protocol (M0493L, New England BioLabs) with enhancer, using the following primers (10 µM, 1:1:1 stoichiometry): *Kras*$^{G12D}$, 5′-GTCTTTCCCCAGCACAGTGC, 5′-CTCTTGCCTACGCCACCAGCTC, 5′-AGCTAGCCACCATGGCTTGAGTAAGTCTGCA; *Fbxw7*$^{null}$, 5′-CAGTGGAG TGAAGTACAACTC, 5′-GCATATTCTAGAGGAGGGTAT, 5′-GGCCAGCC TGGTCTGTATA. The resulting PCR products were analysed using the TapeStation 4200 System (Agilent).

## Hybridization capture bait design

A custom mouse cancer gene list was assembled, with a total of 589 genes. Capture baits were designed using the SureDesign application (Agilent). Species: *Mus musculus* (UCSC mm9, NCBI build 37, July 2007). Target parameters: coding exons + 10 bases each from the 5′ and 3′ ends. Target summary: 9,936 target IDs resolved to 9,936 targets comprising 9,936 regions for a total of 588 genes. Region size: 6.778 Mb. Probe summary: total probes, 155,861; total probe size, 6.757 Mb; coverage, 98.7%.

## Targeted amplicon panel design

Standard BioTools D3 Assay Design software was used to assemble a targeted panel of primers covering ten genes (*Apc*, *Ctnnb1*, *Kras*, *Nras*, *Hras*, *Braf*, *Pten*, *Fbxw7*, *Smad4* and *Trp53*). *Apc*, *Ctnnb1*, *Kras* and *Trp53* had 100% of their exomes covered by the panel, whereas coverage for the other genes was limited to previously identified hotspots sequenced using the hybridization capture panel (Supplementary Fig. 5a).

## Hybridization capture panel sequencing

Captured DNA was sequenced on the Illumina HiSeq 4000 platform, generating paired-end sequence reads from multiplexed libraries.

Bait coordinates were mapped relative to the GRCm38 reference using the liftOver tool[76]. Adaptor clipping and PCR duplicate marking were performed using biobambam2 (v.2.0.79)[77] and sequence reads were aligned to GRCm38 using BWA-MEM (v.0.7.17)[78]. Of the 588 genes originally targeted, 551 (94%) had at least 90% of their coding regions (Ensembl v96 gene build) within 100 bp of a bait mapped to GRCm38. Two genes (*Rit1* and *Dlg2*) had minimal or no coverage, and *Trerf1* was fully covered but not annotated as a gene in the Ensembl v96 gene build. A total of 585 genes (Supplementary Table 3) were analysed for variant calling. The median depth of sequencing coverage across all tumour and normal samples (excluding PCR duplicates) in the target regions was 308×.

## Hybridization capture sequencing variant calling and filtering

The Pisces (v.5.2.10.49) single-sample variant caller[79] was used to call SNVs in each tumour and normal adjacent sample. The stitcher tool within Pisces was used to preprocess the alignment files, followed by variant calling and variant quality recalibration. Normal samples were processed in both the somatic and germline modes within Pisces to call variants with high and low VAFs. The consequences of variants were annotated with the Ensembl Variant Effect Predictor (VEP)[80] using gene builds from Ensembl release 96. In genes with multiple transcripts, only the consequences associated with the canonical transcript, as defined by Ensembl, were considered. In tumour samples, variants that failed Pisces internal filters, variants annotated as single-nucleotide polymorphisms (SNPs) or indels in the Mouse Genomes Project (MGP release v6)[81], and variants overlapping at least one normal sample from germline calling mode were removed. Variants were not removed if they were in the same genomic position as a MGP variant or germline call but with a different variant allele. To remove artefacts from PCR errors, sequencing errors and read misalignments, several additional filtering steps were implemented. To select SNVs with high-quality mismatched bases, SNVs were required to have at least two reads with the alternative allele, where at least half of the alternative alleles had a minimum base alignment quality score of 30 (ref. 82). To remove artefactual variants found in regions where reads are frequently misaligned, SNVs were removed if at least four normal samples had at least four gapped reads within 10 bp of the variant position. To further remove likely artefacts with low VAFs, SNVs were removed if they were called in somatic mode in at least two normal samples. Moreover, a variant was removed if it fell inside a genomic region with a mappability score <1, where mappability was calculated from 75-mers across the reference genome using gem-mappability (build 1.315)[83]. After removing SNPs and artefacts as above, we identified SNVs with low VAFs that were recurrent in the tumours of one or more mice. We suspect that these SNVs were due to coincident lymphoid expansions (leukaemia/lymphoma); recurrent variants were therefore excluded from the tumour call set if they had a maximum VAF ≤ 0.05, or if the variant in a matched normal sample(s) had a VAF ≥ 0.01.

## Identification of driver genes

The dNdScv package (v.0.1.0)[16] was used to identify genes under positive or negative selection. Genes that were either absent from the RefCDS used (*Trerf1*) or not covered by pull-down baits (*Dlg2, Rit1*) were removed from the input gene list. The following dNdScv parameters were used: (refdb = "mm10_Ens96_dNdScv.rda", max_muts_per_gene_per_sample = Inf, max_coding_muts_per_sample = Inf).

## Targeted amplicon sequencing

The targeted amplicon library was prepared according to the Standard BioTools protocol using the 8.8.6 integrated fluidic chip and the Juno system. In brief, each integrated fluidic chip allowed the interrogation of 48 samples against eight primer pools, leading to the generation of 286 amplicons for each sample. The collected amplicons for each chip run were quantified using a Bioanalyzer 2100 (Agilent) and pooled

equimolarly before sequencing as paired-end 150 bp reads on the Illumina NovaSeq platform.

## Targeted amplicon sequencing variant calling and filtering
FASTQ files were aligned against the Genome Reference Consortium mouse genome 39 (GRCm39)[84] using bwamem (https://github.com/lh3/bwa). Samples with a median number of reads per amplicon of <10 were excluded from further analysis. Mutation calling was performed using the ampliconseq pipeline (https://github.com/crukci-bioinformatics/ampliconseq) with VarDict as variant caller and a minimum allele fraction threshold of 0.01. Variant annotation was performed using Ensembl VEP[80]. The list of called mutations was filtered to remove variants that did not pass internal noise filters. Indels were removed because of the tendency for ENU to cause SNVs[85]. Finally, variants were retained only if they were called in at least two overlapping amplicons per sample and supported by at least five mutant reads.

## ENU mutation signature inference
We inferred ENU signatures from SNVs called in hybridization capture-sequenced tumour samples from all cohorts and protocols ($n = 15,979$). A subset of these SNVs ($n = 1,573$) are from ENU-treated cohorts not analysed in the manuscript but were included to increase the total sample size for this analysis. *Apc* high-impact SNVs and all SNVs in *Ctnnb1*, *Ros1* and *Ntrk3* were excluded. A total of 15,362 SNV trinucleotide contexts were converted to the COSMIC convention and normalized to SNV abundance within the hybridization panel target regions to calculate frequency. In *O/E* ratio plots for *Ctnnb1* mutations, the expected frequencies for each mutation trinucleotide context (Fig. 2c) were rescaled by dividing the frequency by the sum of frequencies present in each protocol or cohort analysed (Extended Data Fig. 5a,d). For *Apc*, expected frequencies were calculated per domain bin (A, B, C, D, E) where all frequencies from trinucleotide contexts resulting in a stop codon were summed. Expected frequencies for each of the 15 *Apc* bin combinations were calculated by multiplying the probabilities of the two component bins.

## Decay curves
To generate decay curves illustrating the proportion of tumours remaining in the rescue protocol, we used *Apc* and *Ctnnb1* mutation ratios obtained from sequencing experiments (Figs. 1h and 4e and Extended Data Fig. 8e). These ratios were then multiplied by the average tumour count in the SI at each timepoint, considering both priming TE and the rescue ET10 and ET30 protocols (Fig. 4b). Subsequently, these values were divided by the TE protocol ($X = 0$) to generate the plots. The same logic was applied for individual *Apc* bins and *Ctnnb1* exon-3 mutation decay curves where the ratios obtained from sequencing experiments were multiplied by the average tumour count for *Apc* and *Ctnnb1* at each timepoint. $\chi^2$ statistical tests were used to compare observed and expected tumour counts at each rescue timepoint, with values of <0.05 considered to be statistically significant. Expected tumour counts were generated from neutral decay curve proportions at day 10 and day 30, multiplied by the observed tumour count in the TE priming protocol ($X = 0$). The neutral decay curve was inferred using CryptDriftR (https://github.com/MorrisseyLab/CryptDriftR) with the following parameters: $N_s$ (number of stem cells) = 5, $\lambda$ (replacement rate) = 0.1, $P_r$ (probability of replacement) = 0.5, tau = 1 in a time sequence (1,30, by = 0.5).

## Copy-number analysis
The tumour gDNA samples used for copy-number analysis comprised a subset of the samples used for hybridization capture sequencing (Extended Data Fig. 1). Libraries were prepared from 400 ng gDNA according to the manufacturer's guidelines (Illumina DNA PCR-Free Prep, Tagmentation, 20041795). Libraries were sequenced on the NovaSeq 6000 (Illumina) system. Shallow whole-genome sequence data were aligned to the mm10 reference genome using BWA (v.0.7.17)[86]

and processed with QDNAseq (v.1.30.0)[87] to generate copy-number data. Read counts within 100 kb bins, corrected for sequence mappability and GC content, were normalized relative to a matched normal adjacent sample, if available, before segmentation. For tumour samples for which no matched normal was used, 250,000 reads were sampled from each sample and pooled to create an aligned BAM file that was used as the control. Ploidy (defined as 2), cellularity estimation (2 × average VAF) and absolute copy-number fitting were carried out using Rascal (v.0.7.0)[88].

## GENIE CRC cohort analysis
This study used data from the AACR Project GENIE registry, an international public cancer registry that aggregates clinical-grade cancer genomic data with clinical outcome data[89]. The GENIE colorectal cohort data were accessed through the registry's public data releases (version 15), which are available to the research community for use in genomics projects. We analysed tumour samples (one per donor) containing APC-truncating mutations ($n = 10,329$). Samples were excluded if they had more than two driver mutations (excluding *APC* and *KRAS*) from the following genes: *ATM*, *ARID1A*, *AMER1*, *BRAF*, *FBXW7*, *PTEN*, *PIK3CA*, *SMAD4*, *SOX9* and *TCF7L2* ($n = 9,386$). Moreover, samples with more than two APC mutations were removed. The final dataset ($n = 9,278$ samples) was analysed on the basis of the presence of one or two *APC* mutations and *KRAS* driver mutation status. Each *APC* mutation (for single-hit cases) or the most C-terminal mutation (for double-hit cases) was assigned a 20-amino-acid repeat retention score of 0–7, representing the number of full-length 20-amino-acid repeats preserved (for example, a mutation retaining three repeats received a score of 3). The expected relative frequencies for each potential APC stop codon, assuming a SNV (Fig. 5b), were calculated on the basis of the 96 COSMIC trinucleotide context frequencies from normal human colon[33], recalculated for protein-coding regions.

## INCISE cohort
Ethical approval was obtained for the retrospective use of data (GSH/20/CO/002) and analysis of surplus diagnostic tissue without individual informed consent (22/WS/0020) following applications to Glasgow SafeHaven, NHS Greater Glasgow and Clyde and the West of Scotland Research Ethics Committee.

The cohort comprises individuals 50–74 years of age who underwent polypectomy at bowel screening colonoscopy in NHS Greater Glasgow and Clyde between 2009 and 2016 and had a surveillance colonoscopy between 6 months and 6 years after the index procedure. Data, including demography, clinical characteristics, endoscopy report results and histopathological data, were collected retrospectively. These variables included, but were not limited to, the number of polyps present at index and follow-up colonoscopies, polyp histology, morphology, location, size and grade of dysplasia. Data were collected from electronic health records (Clinical Portal, Orion Health and Trakcare, Intersystem), electronic endoscopy reporting software (Unisoft GI Reporting Software, v.2.5, Unisoft Medical Systems) and Telepath Laboratory Information Management System electronic pathology database and linked using the Community Health Index unique identifier. Individuals with CRC, history of CRC, diagnosed polyposis or CRC predisposition syndrome, and inflammatory bowel disease were not included.

## INCISE adenoma tissue processing and DNA sequencing
The most advanced adenoma removed at the index bowel screening colonoscopy was identified on the basis of lesion size and highest grade of dysplasia, and the corresponding FFPE samples retrieved from the clinical pathology archive. The identified synchronous and metachronous polyps were similarly retrieved. After DNA extraction (Maxwell CSC FFPE, Promega) and quality control (Qubit 4 Fluorometer, Thermo Fisher Scientific, minimum acceptable DNA concentration 1.0 ng μl⁻¹), sample sequencing and variant calling were performed by the Genomic

Innovation Alliance (GIA) using the Agilent SureSelect XT2 HS2 method (Agilent). Regions of interest were enriched with the Agilent SureSelect CancerPlus panel (design ID: S3225252) and the quality and quantity of libraries was determined using the TapeStation using a D1000 ScreenTape (5067–5582, Agilent).

Sequencing data were processed and SNV files were generated using the GIA HOLMES pipeline (v.1.3.1). Bcl2fastq conversion software (Illumina, v.2.19.1.403, v.2.20.0.422 on C++) was used to convert NovaSeq-6000-generated .cblc raw data files to FASTQ files, which were then aligned using Burrows–Wheeler Alignment[90] (v.0.7.15 as a C programme). deepSNV/Shearwater[91,92] (v.1.22/v.1.1.0, v.1.22.0.5/v.1.1.0 as an R package/R wrapper script) was used for calling SNVs, while Pindel[93] (v.0.2.5b8-ww1 as a standalone application) was used for large insertions/deletions. Once called, variants were annotated using CAVA[94] (v.1.2.2.ww1, v.1.2.2.ww5 on Python). We converted VCF files to MAF format using vcf2maf (v.1.6.22) for easier data handling, and VEP (v.112) was used for reannotation. Discordant grouping was used to identify structural variation breakpoints using BRASS[95] (v.5.3.3-ww10 on C++), and copy numbers were called using geneCN (v.2.1 on Perl and on R). A total of 653 index, 123 synchronous and 34 metachronous colonic adenomas were analysed on the basis of the presence of one or two APC driver (truncating) mutations, *KRAS* driver mutation status and location (right, left, rectum). VAFs above 0.5 for the highest-ranked *APC* driver mutations were halved, assuming LOH.

## Data analysis and visualization

Statistical analyses were performed in RStudio[96] mostly with R v.4.3.2. $\chi^2$ statistical tests were used to compare observed and expected counts extracted from signature data and decay curves. Two-sided Fisher's exact tests or two-sample tests of equality of proportions with continuity correction were used to compare mutation proportions. Two-sided *t*-tests were used to compare tumour mutation burden between two groups and ANOVA with Welch's correction for 3+ group comparison. log-rank tests were for mouse survival analysis. For data visualization, oncoprint (waterfall), copy-number frequency and copy-number spectral plots were designed using the GenVisR[97] package (v.1.34.0), while all other plots were designed using the ggplot2 (ref. 98) package (v.3.5.1). *Apc* lollipop plots were designed using MutationMapper[99,100] and further adapted. All figures were assembled using Inkscape software. Immunohistochemistry and RNAscope image analysis and quantification were performed using QuPath[101].

## Reporting summary

Further information on research design is available in the Nature Portfolio Reporting Summary linked to this article.

## Data availability

DNA-sequencing data from hybridization capture sequencing are archived at the European Nucleotide Archive (ENA) under accession PRJEB26559. DNA-sequencing data from amplicon sequencing and shallow whole-genome sequencing (tumour copy-number analysis) are archived in NCBI BioProject under accession PRJNA1327837. DNA-sequencing data from the INCISE polyp cohort are archived at EMBL-EBI ArrayExpress under accession E-MTAB-15619. The AACR GENIE colorectal cohort data were accessed through the registry's public data releases (https://genie.cbioportal.org/study/summary?id=genie_public; version 15). Source data are provided with this paper.

## Code availability

Analysis scripts, including hybridization capture sequencing variant calling, ENU mutation signature inference and GENIE CRC cohort analysis, are available at GitHub (https://github.com/fcLourenco/ENU_mutagenesis_decay_driver_mutations).

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

**Acknowledgements** We thank the members of all of the CRUK CI core facilities, in particular the Biological Resource Unit, Genomics, Light Microscopy, Histopathology and Bioinformatics/Statistics; the staff at the Wellcome Sanger Institute Genomics facility; and C. Ennis for editing assistance. D.J.W. was supported by CRUK Cambridge Institute core funding grant (A24456, Cancer Research UK); D.J.A. and K.W. by a Wellcome Trust grant (220540/Z/20/A); and J.E., G.L., S.T.M., L.S.S. and N.M. by Innovate UK (10054829, 105858), CRUK (CTRQQR-2021\100006) and Beatson Cancer Charity (24-25-045 BCC). We acknowledge the American Association for Cancer Research and its financial and material support in the development of the AACR Project GENIE registry, as well as the members of the consortium for their commitment to data sharing. Interpretations are the responsibility of study authors.

**Author contributions** F.C.L. and D.J.W. conceived the project and designed the research. F.C.L., I.D.S., R.K., D.J.A. and D.J.W. interpreted the results of all of the experiments. F.C.L. (with assistance described below) managed the colony and performed most of the mouse modelling experiments. I.D.S. provided data on the *Apc*^het^ mouse cohort. S.A. performed library preparation for amplicon-seq and CNV experiments. G.G. helped with mouse dissection. A.M.N. provided data on the *Arid1a*^null^ cohort. L.H. performed early bioinformatic analysis. K.W. processed the variant calling/filtering pipeline and analysis of hybridization capture sequencing. A.S. performed bioinformatic analysis on GENIE and INCISE data. J.E., G.L., S.T.M. and N.M. managed and contributed with data from the INCISE project. J.E. leads the INCISE project. L.S.S. performed bioinformatic analysis on INCISE data. M.D.E. processed the copy number data. R.K. processed the amplicon-seq data, ENU signature and helped in multiple bioinformatics tasks. D.J.A. provided advice and infrastructure for the hybridization capture sequencing. D.J.W. and F.C.L. wrote the manuscript.

**Competing interests** The authors declare no competing interests.

**Additional information**
**Correspondence and requests for materials** should be addressed to Douglas J. Winton.

**a**

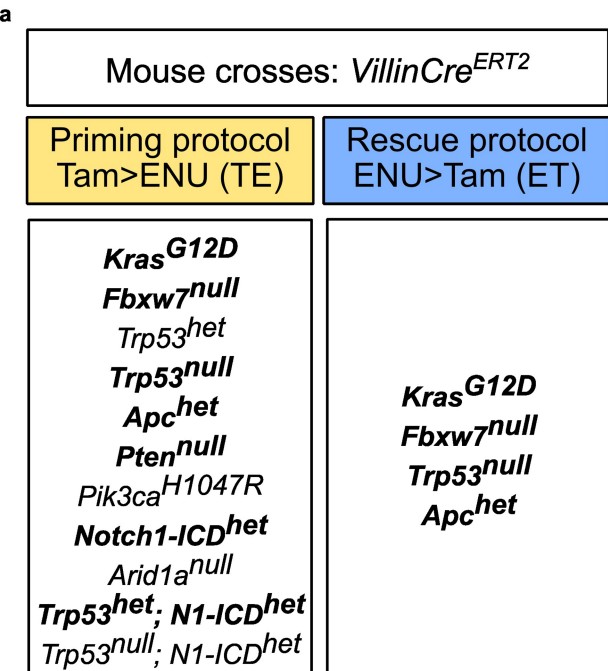

**b**

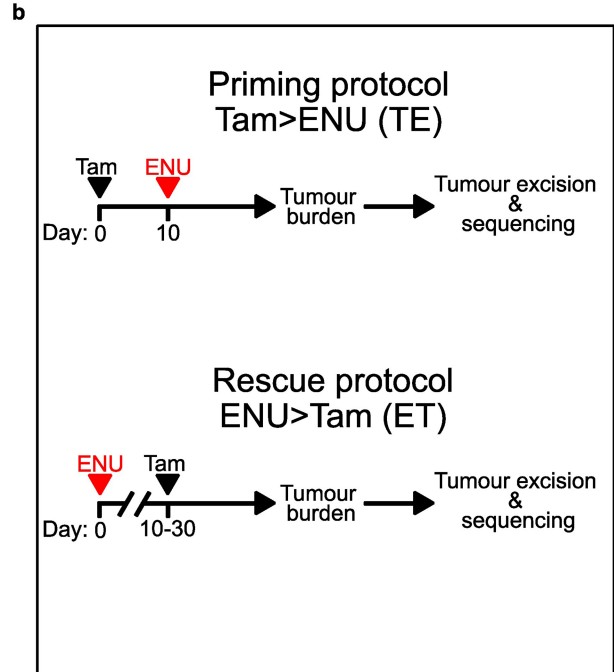

**c**

Tumour Sequencing

| Control ENU only (E) | Priming protocol Tam>ENU (TE) | Rescue protocol ENU>Tam (ET) |
|---|---|---|

| Hybridisation capture 585 genes 43 Tumours | Hybridisation capture 585 genes 196 Tumours | Multiplex Amplicon 10 genes 205 Tumours | Hybridisation capture 585 genes 72 Tumours | Multiplex Amplicon 10 genes 170 Tumours |

| sWGS (CNV) 12 Tumours | sWGS (CNV) 69 Tumours | | sWGS (CNV) 68 Tumours | |

**Extended Data Fig. 1 | Breakdown of experimental cohorts and study design. a**, *Villin*CreERT2 and various conditional (floxed) mouse lines were inter-crossed to generate the experimental cohorts shown, with sequenced cohorts highlighted in bold. **b**, Schematic of priming and rescue protocols. **c**, Sequencing modalities employed in analysis of tumours from priming and rescue protocols. sWGS (shallow whole-genome sequenced) tumours are a subset of tumours sequenced using hybridization capture to identify copy number variation (CNV).

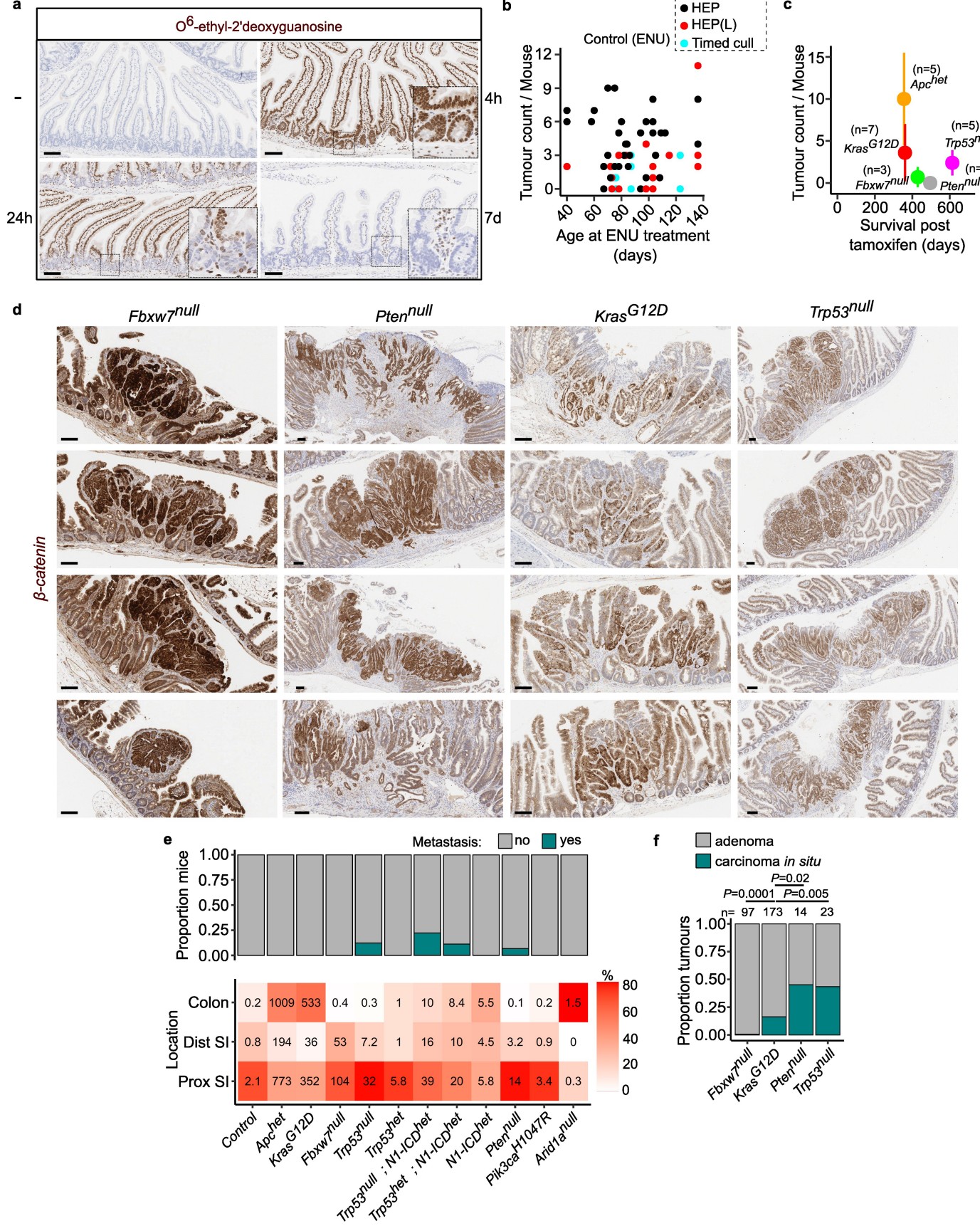

**Extended Data Fig. 2** | See next page for caption.

**Extended Data Fig. 2 | Profiling of intestinal epithelium following ENU treatment. a**, Immunohistochemistry (IHC) of ENU-induced O6-ethyl-2'deoxyguanosine adducts at different time-points post-ENU (no ENU control, 4 h, 24 h, and 7 days; n = 1 mouse per treatment/time-point) in proximal SI. **b**, Intestinal tumour count per mouse in relation to age at treatment in control (ENU) cohort (HEP, humane end point; HEP(L), presence of lymphoma/leukaemia at HEP). **c**, Mean (SD) intestinal tumour count per mouse in relation to median survival in tamoxifen-induced (Tam) cohorts. **d**, Representative β-catenin IHC in a panel of tumours from $Fbxw7^{null}$ (n = 97), $Pten^{null}$ (n = 14), $Kras^{G12D}$ (n = 173), and $Trp53^{null}$ (n = 23) cohorts. **e**, Lower panel: heatmap with relative tumour burden per location in relation to cohort. Mean tumour counts displayed in each square. Upper panel: proportion of mice with distant metastases per cohort. **f**, Proportion of tumours with local invasion (carcinoma in situ) in relation to cohort. *P*-values in (f) were derived using a two-sample test of equality of proportions with continuity correction (two-sided). Scale bar = 100 μm.

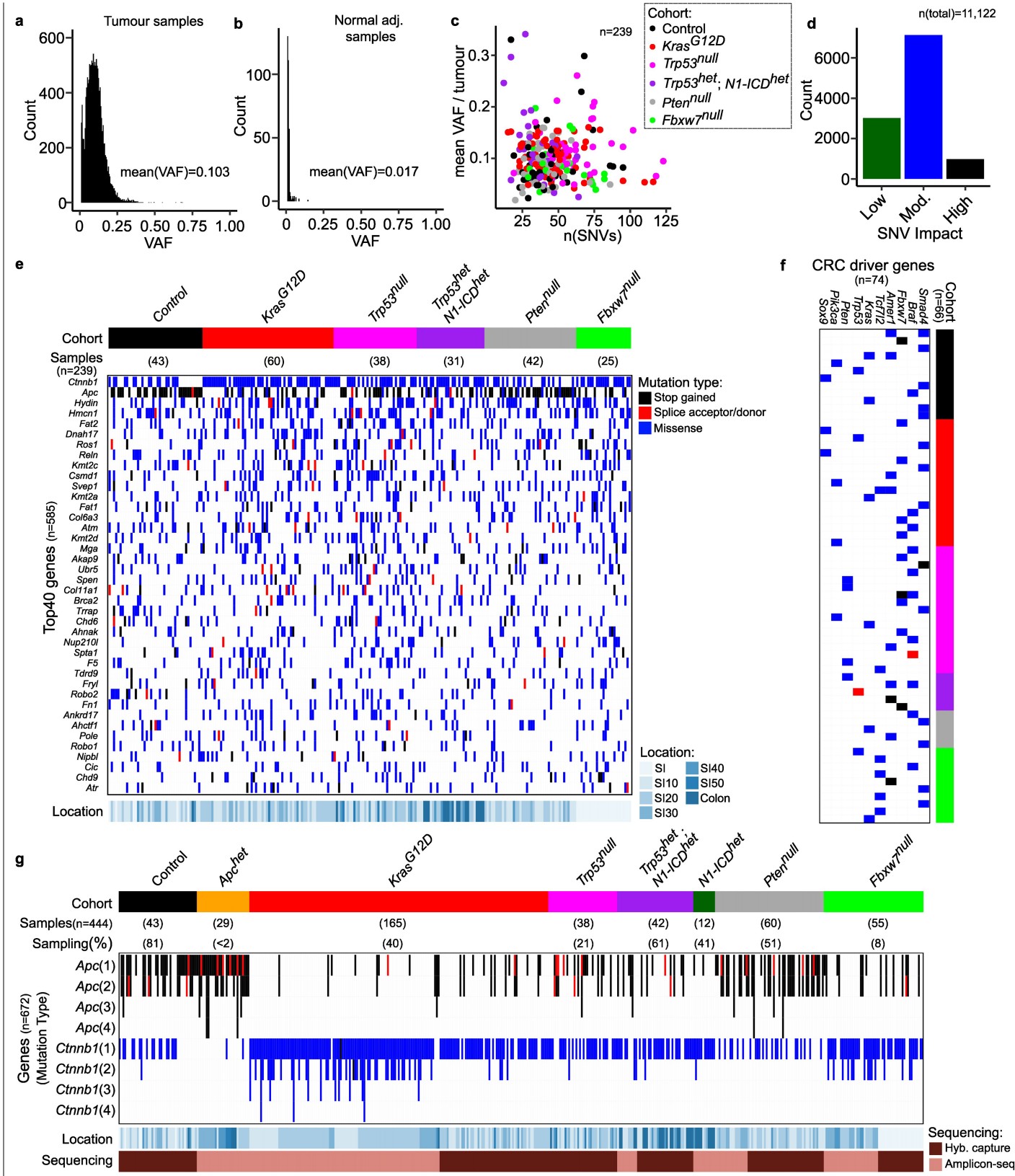

**Extended Data Fig. 3 | Tumour sequencing analysis. a**, Variant allele frequency (VAF) histogram for each SNV called in tumour samples. **b**, VAF histogram for each SNV called in adjacent normal samples. **c**, Mean VAF per tumour sample (colour indicates cohort) relative to number (n) of called SNVs. **d**, SNV count relative to Ensembl variant effect predictor (VEP) IMPACT (Low = synonymous; Moderate = missense; High = stop gained, splice site). $n$(LOW) = 3,019, $n$(MODERATE) = 7,131, $n$(HIGH) = 972, $n$(total) = 11,122. **e**, Oncoprint of the 40 most recurrently mutated genes, in descending order.

Tumour samples are arranged by cohort. Legend shows mutation type (synonymous SNVs not plotted). Location: SI, small intestine; SI10, SI20, SI30, SI40, and SI50, refer to tissue samples taken at 0–10, 10–20, 20–30, 30–40, and 40–50 cm distance from pyloric sphincter). **f**, Oncoprint of colorectal cancer (CRC) driver gene subset. **g**, Oncoprint of *Apc* and *Ctnnb1* SNVs. Number in brackets following gene names accommodates multiple SNVs per gene per tumour. Legend key for mutation type and location as in (e).

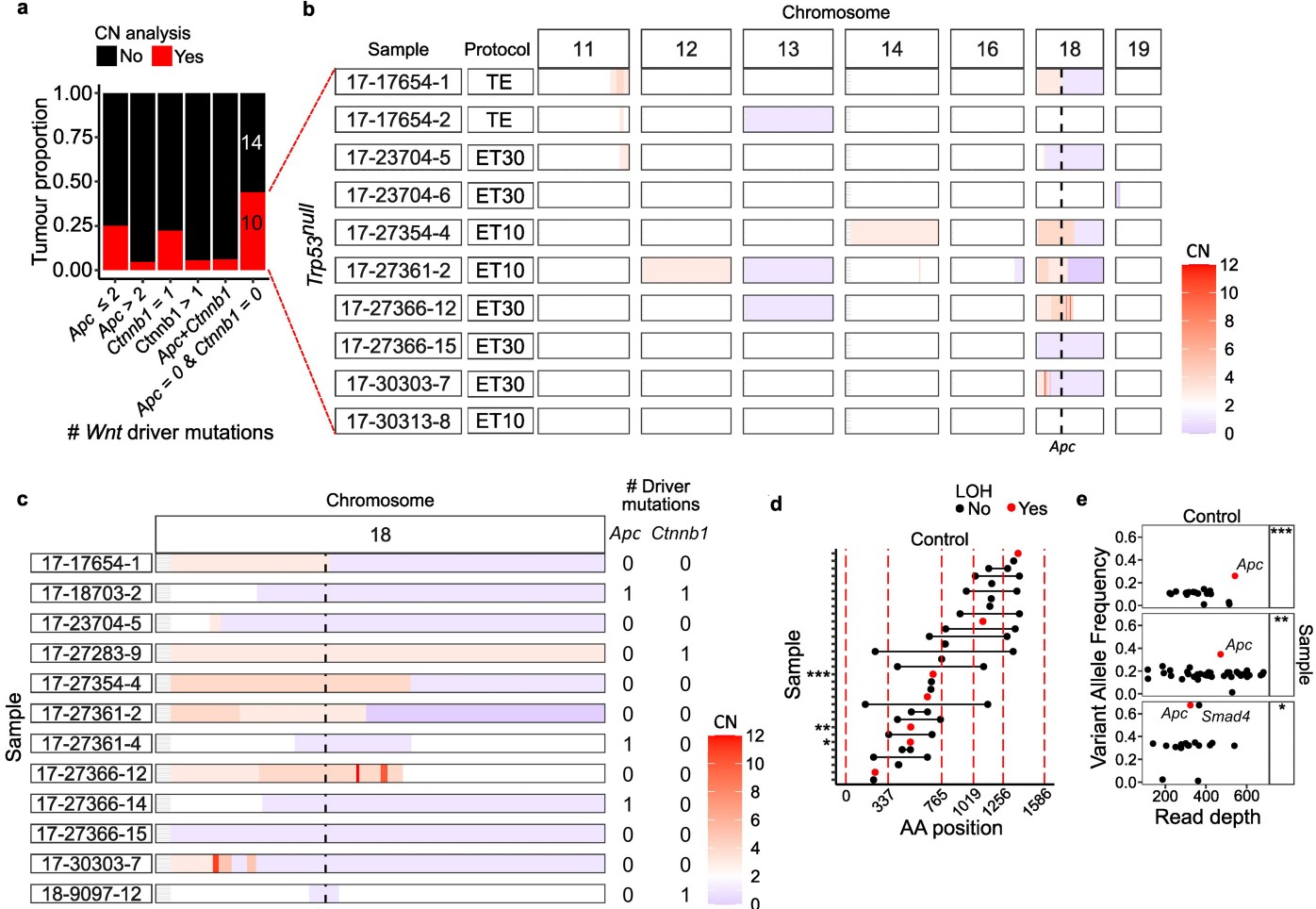

**Extended Data Fig. 4 | Copy number variation in relation to _Wnt_ driver mutation status. a**, Proportion of tumours with and without copy number (CN) analysis relative to the number of _Wnt_ driver mutations for all cohorts and protocols. **b**, Copy number spectrum plot of informative chromosomes for tumours with no _Apc_ and _Ctnnb1_ drivers. Vertical dashed line indicates _Apc_ locus in Chr18. **c**, Copy number spectrum plot for all samples with copy number variants in chromosome 18 (Chr18) where the _Apc_ locus is located (dashed line), combined with a table displaying number of driver mutations for the _Wnt_ drivers _Apc_ and _Ctnnb1_. Legend shows heat map with CN value. **d**, _Apc_ mutation combination per tumour sample in control cohort with LOH status. **e**, Examples of copy number neutral LOH from (d) where _Apc_ VAF is -2× of clonal VAF. Asterisks are used to match samples in (d).

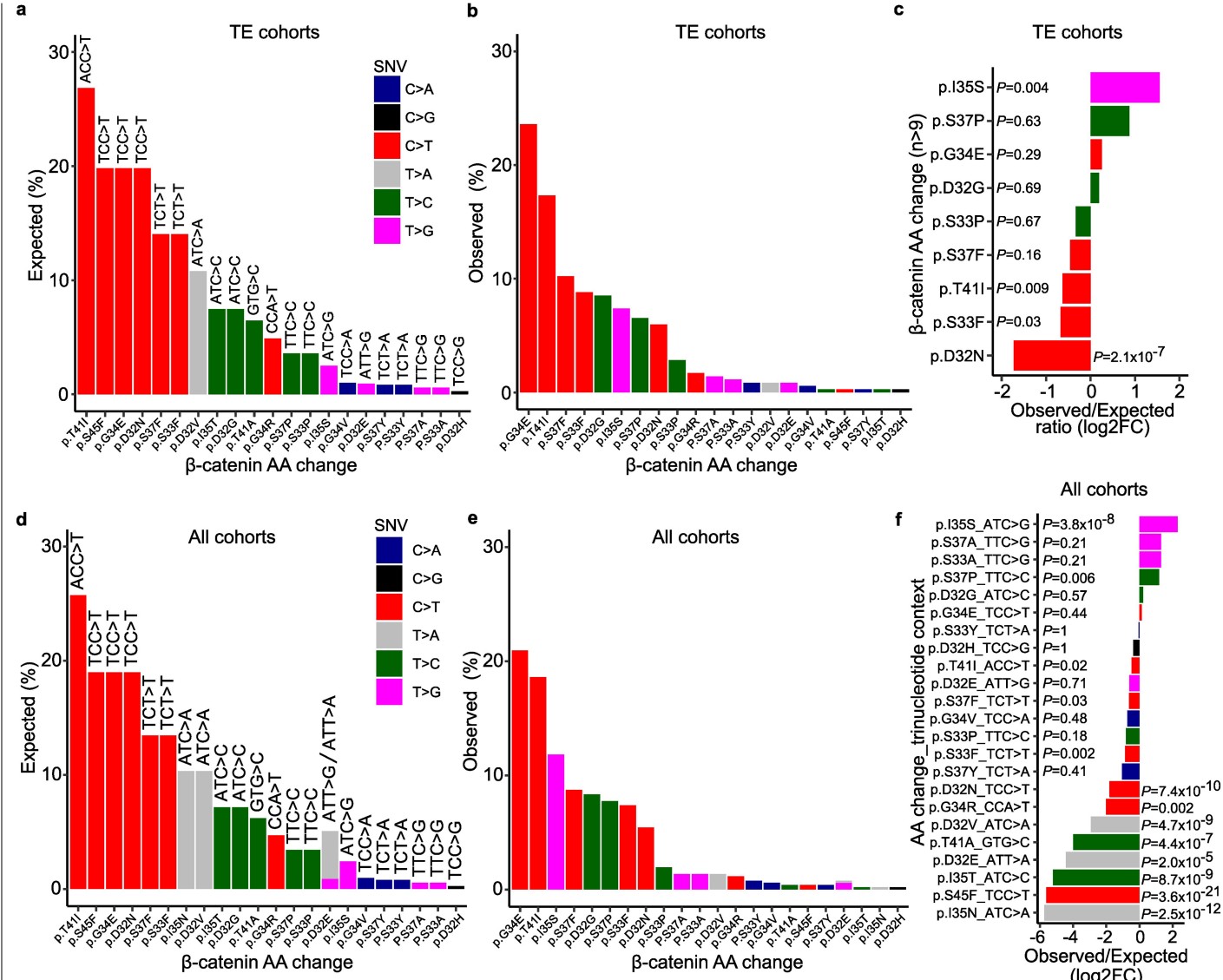

**Extended Data Fig. 5 | Impact of ENU mutational signature on *Ctnnb1*.**
**a**, Predicted frequency (expected) of *Ctnnb1* exon 3 mutations from mutational preference for primed (TE) cohorts. **b**, Observed frequency of *Ctnnb1* exon 3 mutations identified for TE cohorts. **c**, Observed/expected ratios for β-catenin driver aa change (*n* > 9). **d**, Predicted frequency (expected) of *Ctnnb1* exon 3

mutations from mutational preference for all cohorts. **e**, Observed frequency of *Ctnnb1* exon 3 mutations identified for all cohorts. **f**, Observed/expected for all β-catenin driver aa changes. *P*-values were derived using a chi-squared test for each mutation (df=1).

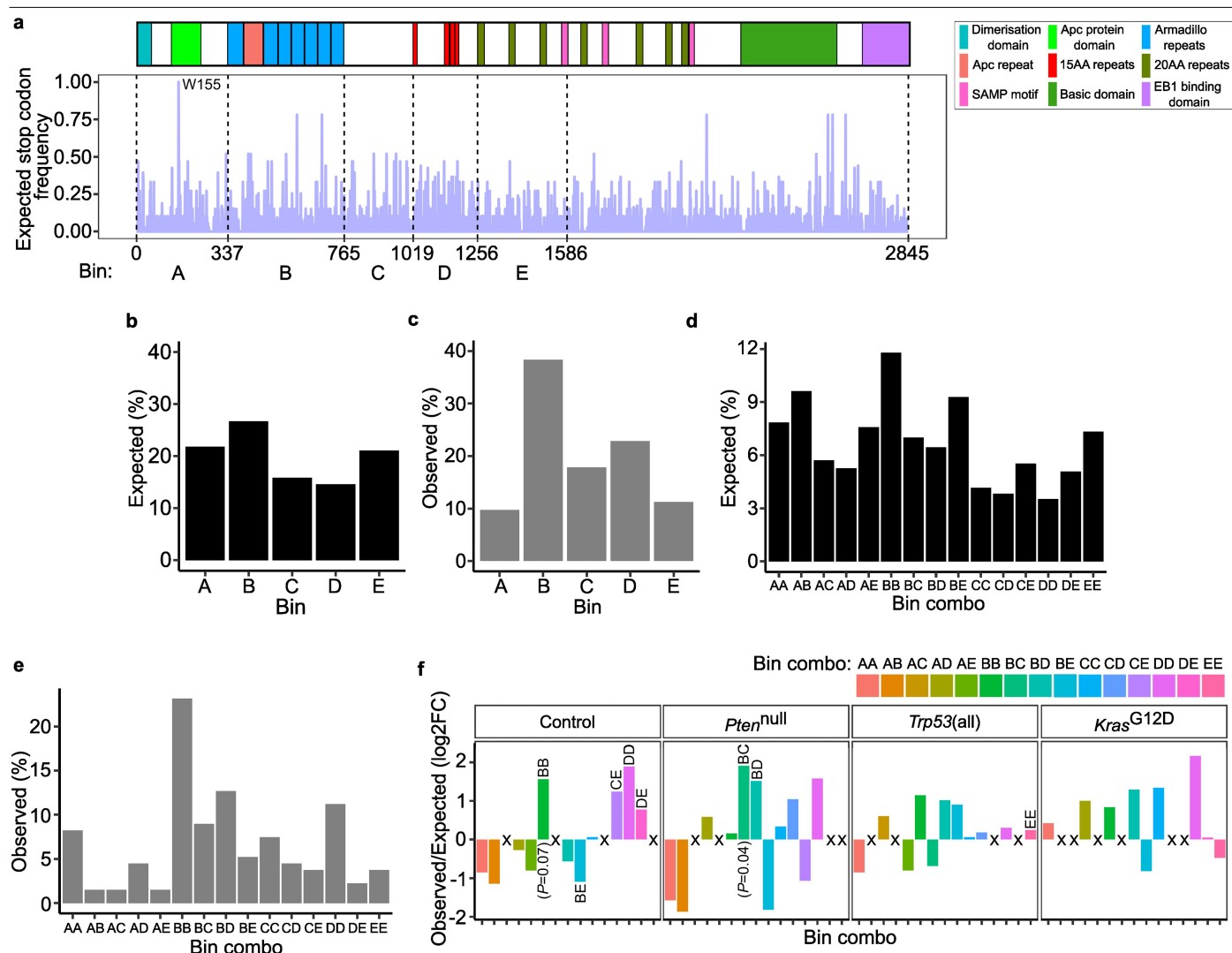

**Extended Data Fig. 6 | Impact of ENU mutational signature on *Apc*. a**, Plot showing expected relative frequency for each *Apc* codon to mutate into a stop assuming an SNV. Values were normalized against likeliest stop codon (encoding W155). **b**, **c** Expected (b) and observed (c) mutations in relation to APC domain bin for primed cohorts. **d**, **e** Expected (d) and observed (e) mutations in relation to APC domain bin combination for primed cohorts.

**f**, Observed/expected ratios in relation to APC bin combination for each cohort. Bins with no mutations in each cohort are marked with X. *P*-values in (f) were derived using a chi-squared test (df=1). Only statistically significant or borderline *P*-values are displayed. Exact *P*-values for (f) are provided in the source data.

**Extended Data Fig. 7 | *Apc* mutation overview for all cohorts. a**, Plot showing expected and observed relative frequencies for each *Apc* codon in N-terminal half (aa 1–1,586) for all cohorts. Frequencies were normalized against most expected and observed stop codon position (encoding W155 for both). **b**, Plot showing truncation position in APC for each tumour or clone and linked N- and C-terminal truncation locations from cases where different mutations are paired for all cohorts except *Apc^het^*. Dashed lines separate bin (black) and domain/repeats (blue) boundaries depicted in legend below. MCR: mutation cluster region in human CRC.

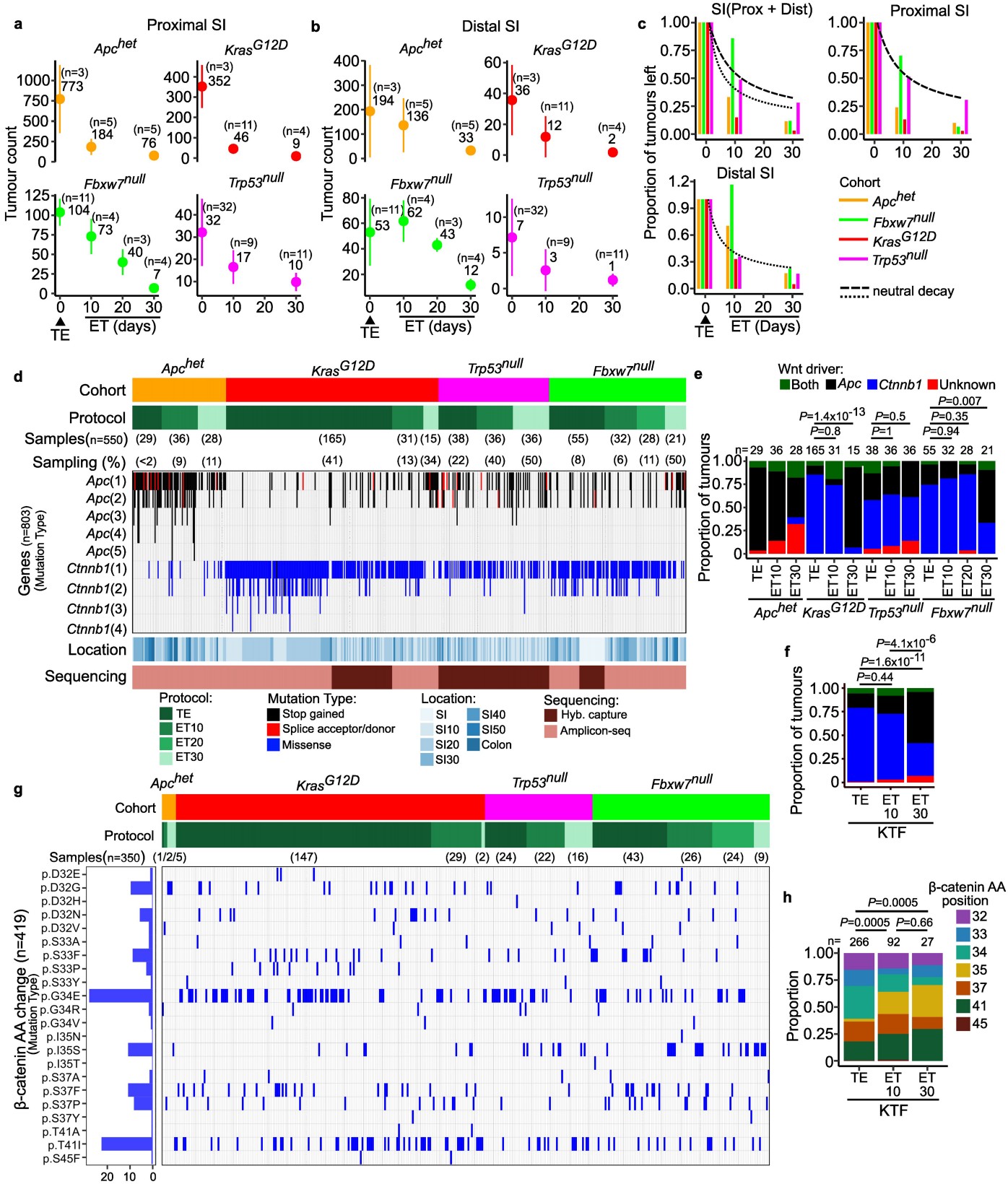

**Extended Data Fig. 8 | Tumour count and *Wnt* driver mutation analysis in rescue protocol. a**, **b**, Mean (SD) tumour count for priming and rescue protocols in proximal (a) and distal (b) small intestine. **c**, Proportion of tumours remaining in rescue protocol relative to priming protocol in proximal SI, distal SI, or both combined (SI) for each cohort. Neutral decay curves for each location shown. **d**, Oncoprint showing proportions of tumours in each cohort containing detected *Apc* or *Ctnnb1* mutations. **e**, **f**, Proportion of tumours containing detected *Apc* or *Ctnnb1* mutations with priming and rescue protocols for each cohort (e) and for combined *Kras*[G12D], *Trp53*[null], and *Fbxw7*[null] (KTF) cohorts (f). **g**, Oncoprint showing *Ctnnb1* exon 3 mutations in priming and rescue protocols. **h**, *Ctnnb1* exon 3 mutation proportion in priming and rescue protocols for KTF cohorts. *P*-values in panels (e, f) were derived using a two-sample test for equality of proportions with continuity correction (two-sided) of tumours driven by *Apc* mutations. *P*-values in panel (h) were derived using Fisher's exact test (two-sided).

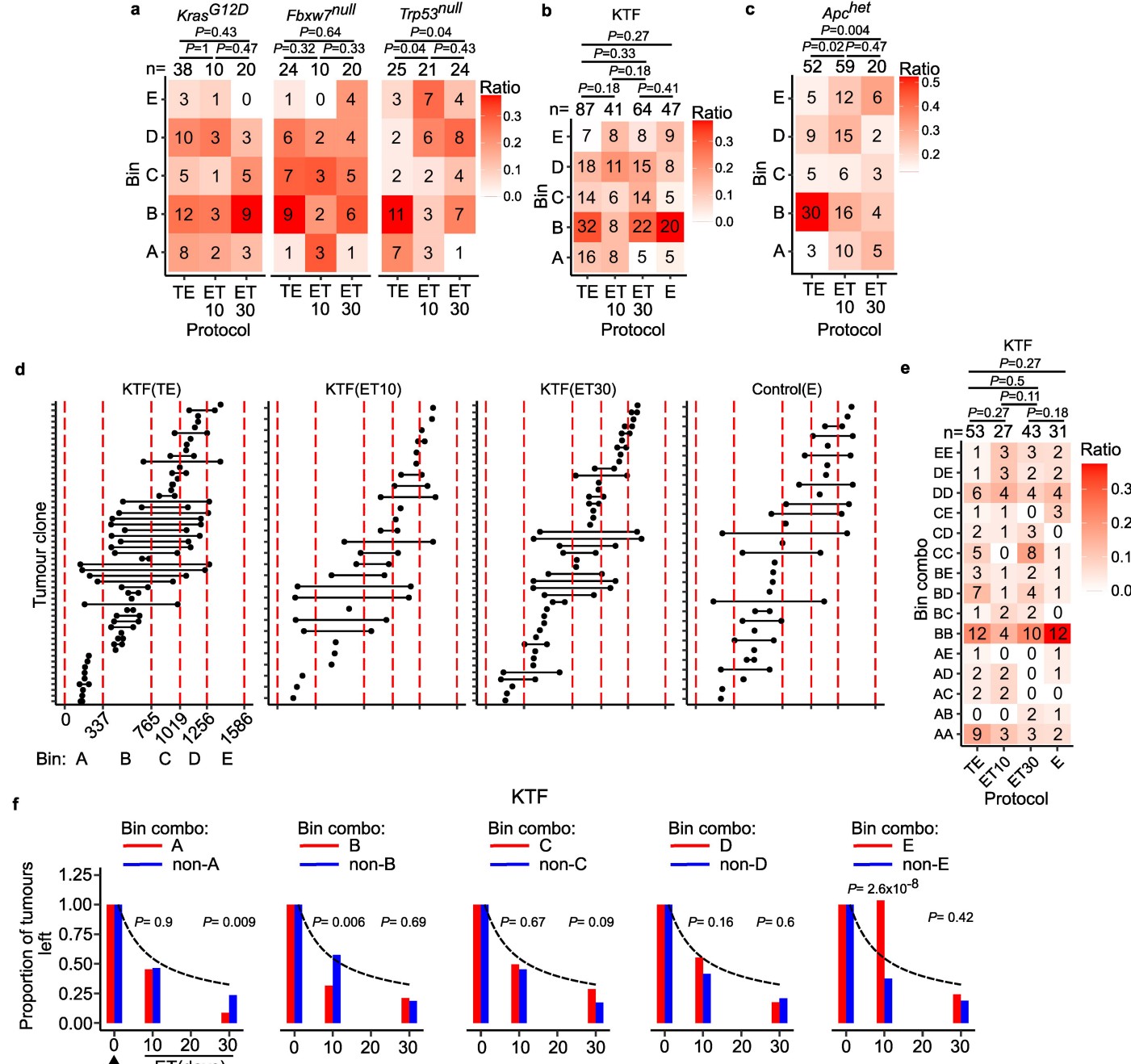

**Extended Data Fig. 9 | *Apc* mutation analysis in rescue protocol. a**, Heatmap with count data for *Apc* domain bins in relation to protocol for each cohort. **b**, Heatmap with count data for *Apc* domain bins in relation to protocol for combined *Kras*[G12D], *Trp53*[null], and *Fbxw7*[null] (KTF) cohorts. **c**, Heatmap with count data for *Apc* domain bins in relation to protocol for *Apc*[het] cohort. **d**, Plot showing truncation position in the *Apc* gene for each tumour or clone and linked N- and C-terminal truncation locations from cases where different mutations are paired in KTF and control cohorts and compared throughout different protocols. **e**, Heatmap with count data for each bin combination in

(d) for cohorts shown. Letters define N- and C-terminal bin combinations. **f**, Proportion of tumours remaining in rescue protocol with bin combinations containing bin A, B, C, D, or E compared to all but respective bin combination non-A, non-B, non-C, non-D, or non-E in relation to priming protocol for KTF cohort. Dashed line models neutral clonal decay in proximal SI. *P*-values in (a, b, c, e) were derived using Fisher's exact test (two-sided). *P*-values in panel (f) were derived for each rescue time-point using a chi-squared test (df=1) for each APC bin combination.

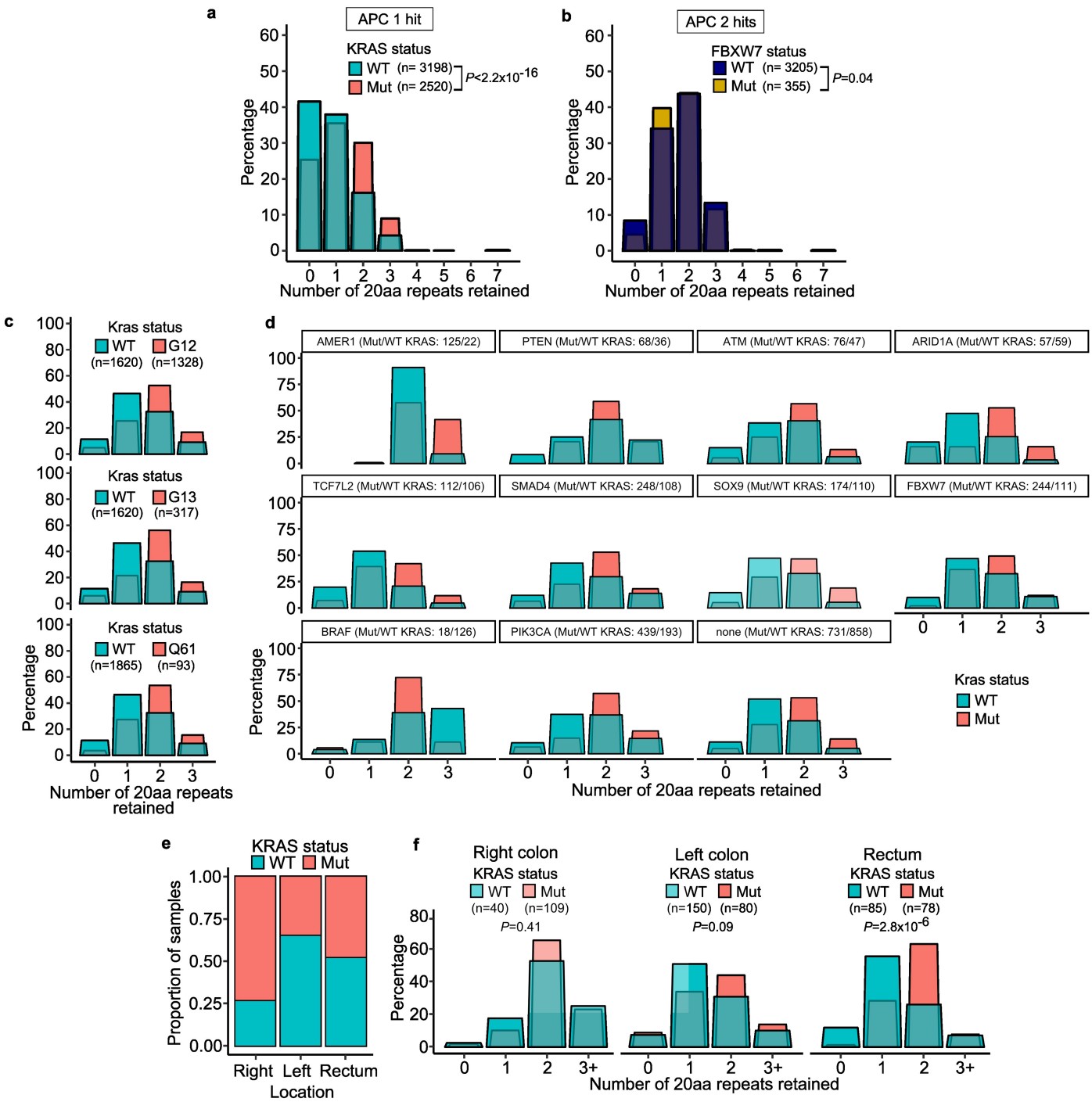

**Extended Data Fig. 10 | GENIE CRC cohort analysis. a**, Relationship between APC 20aa repeat retention and tumour sample percentage, stratified by *KRAS* status in samples with one *APC* hit. **b**, Relationship between APC 20aa repeat retention and tumour sample percentage, stratified by FBXW7 status in samples with two APC hits. **c**, Relationship between APC 20aa repeat retention and tumour sample percentage, stratified by modified *KRAS* driver mutation aa position (G12, G13, Q61). **d**, Relationship between APC 20aa repeat retention and tumour sample percentage in samples with additional driver mutations, stratified by *KRAS* status and driver mutation. **e**, Proportion of samples from GENIE cohort subset (DFCI and MSKCC) stratified by *KRAS* status and location in colon (right, left, rectum). **f**, Relationship between APC 20aa repeat retention and tumour sample percentage, stratified by *KRAS* status and location. *P*-values in (a, b, f) were derived from a Fisher's exact test (two-sided).

# Reporting Summary

## Statistics

For all statistical analyses, confirm that the following items are present in the figure legend, table legend, main text, or Methods section.

| n/a | Confirmed | |
|---|---|---|
| ☐ | ☒ | The exact sample size (*n*) for each experimental group/condition, given as a discrete number and unit of measurement |
| ☐ | ☒ | A statement on whether measurements were taken from distinct samples or whether the same sample was measured repeatedly |
| ☐ | ☒ | The statistical test(s) used AND whether they are one- or two-sided<br>*Only common tests should be described solely by name; describe more complex techniques in the Methods section.* |
| ☐ | ☒ | A description of all covariates tested |
| ☐ | ☒ | A description of any assumptions or corrections, such as tests of normality and adjustment for multiple comparisons |
| ☐ | ☒ | A full description of the statistical parameters including central tendency (e.g. means) or other basic estimates (e.g. regression coefficient) AND variation (e.g. standard deviation) or associated estimates of uncertainty (e.g. confidence intervals) |
| ☐ | ☒ | For null hypothesis testing, the test statistic (e.g. *F*, *t*, *r*) with confidence intervals, effect sizes, degrees of freedom and *P* value noted<br>*Give P values as exact values whenever suitable.* |
| ☒ | ☐ | For Bayesian analysis, information on the choice of priors and Markov chain Monte Carlo settings |
| ☒ | ☐ | For hierarchical and complex designs, identification of the appropriate level for tests and full reporting of outcomes |
| ☐ | ☒ | Estimates of effect sizes (e.g. Cohen's *d*, Pearson's *r*), indicating how they were calculated |

*Our web collection on statistics for biologists contains articles on many of the points above.*

## Software and code

Policy information about availability of computer code

Data collection
Microscopy and image analysis/scoring:
Leica SP5 PCS microscope with LAS X software (v2.8.0, Leica)
Leica Aperio AT2 slide scanner
Qupath (0.5.1)
Fiji/Image J (version 1.53t)

FACS:
BD FACS Aria SORP (BD Biosciences)

Amplicon design:
Standard BioTools D3 Assay Design software

Hybridization capture probe bait design:
SureDesign  (Agilent)

Sequencing:
Illumina HiSeq 4000
Illumina NovaSeq6000
Illumina NovaSeqX

Analysis scripts including Hybridisation capture sequencing variant calling, ENU mutation signature inference, and GENIE CRC cohort analysis are available in GitHub https://github.com/fcLourenco/ENU_mutagenesis_decay_driver_mutations.

Data analysis

Data and statistical analysis were performed in RStudio mostly with R version 4.3.2.
Data visualization was performed

DNA-seq variant calling:
biobambam2 (v2.0.79)
BWA-MEM (v0.7.17)
Pisces (v5.2.10.49)
Ensembl Variant Effect Predictor (Ensembl release 96)
Amplicon-seq pipeline (https://github.com/crukci-bioinformatics/ampliconseq)

Identification of driver genes:
dNdScv package (v0.1.0)

Copy Number analysis (sWGS):
BWA (v0.7.17)
QDNAseq (v1.30.0)
Rascal (v0.7.0)

INCISE cohort variant calling:
GIA HOLMES pipeline (V1.3.1)
Bcl2fastq conversion software (Illumina, V2.19.1.403, V2.20.0.422 on C++)
Burrows-Wheeler Alignment (V0.7.15)
deepSNV/Shearwater (V1.22/V1.1.0, V1.22.0.5/ V1.1.0)
Pindel (V0.2.5b8-ww1 )
CAVA(V1.2.2.ww1, V1.2.2.ww5)
vcf2maf (V1.6.22)
Ensembl VEP (V112)
BRASS (V5.3.3-ww10)
geneCN (V2.1)

For manuscripts utilizing custom algorithms or software that are central to the research but not yet described in published literature, software must be made available to editors and reviewers. We strongly encourage code deposition in a community repository (e.g. GitHub). See the Nature Portfolio guidelines for submitting code & software for further information.

# Data

Policy information about availability of data

All manuscripts must include a data availability statement. This statement should provide the following information, where applicable:
- Accession codes, unique identifiers, or web links for publicly available datasets
- A description of any restrictions on data availability
- For clinical datasets or third party data, please ensure that the statement adheres to our policy

DNA sequencing data from hybridisation capture sequencing is archived at the European Nucleotide Archive (ENA) under accession PRJEB26559. DNA sequencing data from amplicon sequencing and shallow whole genome sequencing (tumour copy number analysis) is archived in NCBI BioProject under accession PRJNA1327837. DNA sequencing data from INCISE polyp cohort is archived in EMBL-EBI ArrayExpress under accession E-MTAB-15619. The AACR GENIE colorectal cohort data was accessed through the registry's public data releases (https://https://genie.cbioportal.org/study/summary?id=genie_public;  version 15). Source data is provided with this manuscript.

# Research involving human participants, their data, or biological material

Policy information about studies with human participants or human data. See also policy information about sex, gender (identity/presentation), and sexual orientation and race, ethnicity and racism.

| | |
|---|---|
| Reporting on sex and gender | Patients recorded sex was defined by the entry in their electronic health record and their Community Health Index (CHI) identifier.  No sex or gender specific analysis was undertaken or reported in this manuscript. |
| Reporting on race, ethnicity, or other socially relevant groupings | No data on race, ethnicity or other socially relevant or vulnerable groups was collected or is reported in this manuscript. |
| Population characteristics | The patient cohort consists of 775 patients aged between 50 and 74 years. Patients who underwent polypectomy at bowel screening colonoscopy in NHS Greater Glasgow and Clyde between 2009 and 2016.  Patients with colorectal cancer, past history of colorectal cancer, diagnosed polyposis or colorectal cancer predisposition syndrome and inflammatory bowel disease were not included. |
| Recruitment | Data including demography, clinical characteristics, endoscopy report results and histopathological data were collected |

| Recruitment | retrospectively. Data were collected from electronic health records (Clinical Portal, Orion Health, Boston USA and Trakcare, Intersystem, Boston USA) , electronic endoscopy reporting software (Unisoft GI Reporting Software, v2.5, Unisoft Medical Systems, UK) and Telepath Laboratory Information Management System electronic pathology database and linked using the Community Health Index (CHI) unique identifier. |
|---|---|
| Ethics oversight | Ethical approval was obtained for the retrospective use of data (GSH/20/CO/002) and analysis of surplus diagnostic tissue without individual informed consent (22/WS/0020) following applications to Glasgow SafeHaven, NHS Greater Glasgow and Clyde and the West of Scotland Research Ethics Committee. |

Note that full information on the approval of the study protocol must also be provided in the manuscript.

# Field-specific reporting

Please select the one below that is the best fit for your research. If you are not sure, read the appropriate sections before making your selection.

☒ Life sciences ☐ Behavioural & social sciences ☐ Ecological, evolutionary & environmental sciences

For a reference copy of the document with all sections, see nature.com/documents/nr-reporting-summary-flat.pdf

# Life sciences study design

All studies must disclose on these points even when the disclosure is negative.

| Sample size | Formal sample size calculation and power analysis were not performed because no prior data or reliable effect size estimates were available for these novel transgenic mouse models combined with ENU mutagenesis. Instead, sample sizes were informed by preliminary pilot experiments while adhering to the 3Rs (Replacement, Reduction, Refinement) to minimize unnecessary animal use. |
|---|---|
| Data exclusions | No data was excluded from analysis with the exception of a small number of sequenced tumour samples that were found a posteriori to be non-diseased. |
| Replication | Each tumour was treated as a biological replicate given its unique genomic composition resulting from random ENU mutagenesis. Tumour burdens were reproducible and consistent across experimental mouse batches. Tissue sections from n = 14–173 tumours (depending on cohort) were stained and scored for invasive features and β-catenin status. For knockout efficiency, n = 6–16 tumours (depending on genetic background) were analysed by PCR or assessed by Trp53 RNA staining or Pten IHC. O6-ethyl-2-deoxyguanosine IHC was performed on multiple normal tissue sections from one mouse per experimental condition, consistent with findings we previously reported (Sadien et al., Nature 2024). BrdU, Lgr5-GFP, and confetti lineage-tracing staining were conducted on multiple intestinal regions in n<3 mice, with replication constrained by animal availability. |
| Randomization | Formal randomization was not performed, however, groups were partially randomized by co-housing control and experimental animals irrespective of genotype and treating them simultaneously. Cohorts included a balanced representation of both sexes. |
| Blinding | For animal welfare reasons, researchers were not blinded to mouse genotype during study and data collection. Blinding was performed for IHC section staining count. |

# Reporting for specific materials, systems and methods

We require information from authors about some types of materials, experimental systems and methods used in many studies. Here, indicate whether each material, system or method listed is relevant to your study. If you are not sure if a list item applies to your research, read the appropriate section before selecting a response.

## Materials & experimental systems

| n/a | Involved in the study |
|---|---|
| ☐ | ☒ Antibodies |
| ☒ | ☐ Eukaryotic cell lines |
| ☒ | ☐ Palaeontology and archaeology |
| ☐ | ☒ Animals and other organisms |
| ☒ | ☐ Clinical data |
| ☒ | ☐ Dual use research of concern |
| ☒ | ☐ Plants |

## Methods

| n/a | Involved in the study |
|---|---|
| ☒ | ☐ ChIP-seq |
| ☐ | ☒ Flow cytometry |
| ☒ | ☐ MRI-based neuroimaging |

## Antibodies

| Antibodies used | 1. CTNB1 antibody, mouse IgG1, clone 14/Beta-Catenin, cat#610154, BD Biosciences.<br>2. O6-ethyl-2'deoxyguanosine antibody, rat IgG2b, clone ER6, Cat# SQX-SQM001.1, Squarix Biotechnology |
|---|---|

3. GFP antibody (10μg/ml, chicken, ab13970, Abcam)
4.  PTEN antibody (1:300, rabbit, 9559, Cell Signalling)
5. BrdU antibody (2.6μg/ml, sheep, ab1893, Abcam)
6. CD326 (EpCAM) AlexaFluor 647 antibody (1:2,000, 118210, Biolegend)

Secondary antibodies:
7. rabbit anti-rat (1:250, A110-322A, Bethyl Laboratories)
8. rabbit anti-mouse (1:1500, ab125913, Abcam)
9. rabbit-anti-chicken (1:500, 303-005-003, Jackson Labs)
10. anti-sheep (1:500, 313-005-003, Jackson Labs).

**Validation**

Antibodies were validated by our histology core facility using negative controls. The staining patterns in our manuscript confirm previous publications:
1. Mientjes et al. Formation and persistence of O6-ethylguanine in genomic and transgene DNA in liver and brain of lacZ transgenic mice treated with N-ethyl-N-nitrosourea. Carcinogenesis (1996).
2. Tateishi K, Omata M, Tanaka K, Chiba T. The NEDD8 system is essential for cell cycle progression and morphogenetic pathway in mice. J Cell Biol. 2001; 155(4):571-579. (Clone-specific: Immunofluorescence, Immunohistochemistry)

# Animals and other research organisms

Policy information about studies involving animals; ARRIVE guidelines recommended for reporting animal research, and Sex and Gender in Research

**Laboratory animals**

Mus musculus. The majority of mouse cohorts are near-inbred C57Bl/6J  with outbred chromosomal regions corresponding to knocked-in genes.  All experiments used male and female mice of at least 6 and 8 weeks of age, respectively.
The following strains were used:
VillinCreERT2
Kras(lsl)G12D/wt (KrasG12D)
Pik3ca(lsl)H1047R/wt (Pik3caH1047R)
ApcΔ14/wt (Apchet)
Trp53Δ2-10/Δ2-10 (Trp53null)
PtenΔ5/Δ5 (Ptennull)
Fbxw7Δ5/Δ5 (Fbxw7null)
R26-Notch1ic/wt (N1-ICDhet)
Arid1aΔ8/Δ8 (Arid1anull)
R26-CAG-Brainbow2.1/Confetti (R26(lsl)Confetti/wt)
Lgr5-DTR-EGFP

**Wild animals**

No wild animals were used.

**Reporting on sex**

Male and female mice were used in all cohorts with different proportions. Sex based analysis for tumour burden and sequencing were not included and can be provided at request.

**Field-collected samples**

No field collected samples.

**Ethics oversight**

All animal experiments were performed in accordance with the guidelines of the UK Home Office under the authority of a project licence (PD5F099BE) approved by the Animal Welfare and Ethical Review Body at the CRUK Cambridge Institute, University of Cambridge.

Note that full information on the approval of the study protocol must also be provided in the manuscript.

# Plants

**Seed stocks**

*Report on the source of all seed stocks or other plant material used. If applicable, state the seed stock centre and catalogue number. If plant specimens were collected from the field, describe the collection location, date and sampling procedures.*

**Novel plant genotypes**

*Describe the methods by which all novel plant genotypes were produced. This includes those generated by transgenic approaches, gene editing, chemical/radiation-based mutagenesis and hybridization. For transgenic lines, describe the transformation method, the number of independent lines analyzed and the generation upon which experiments were performed. For gene-edited lines, describe the editor used, the endogenous sequence targeted for editing, the targeting guide RNA sequence (if applicable) and how the editor was applied.*

**Authentication**

*Describe any authentication procedures for each seed stock used or novel genotype generated. Describe any experiments used to assess the effect of a mutation and, where applicable, how potential secondary effects (e.g. second site T-DNA insertions, mosiacism, off-target gene editing) were examined.*

# Flow Cytometry

## Plots

Confirm that:

☒ The axis labels state the marker and fluorochrome used (e.g. CD4-FITC).

☒ The axis scales are clearly visible. Include numbers along axes only for bottom left plot of group (a 'group' is an analysis of identical markers).

☒ All plots are contour plots with outliers or pseudocolor plots.

☒ A numerical value for number of cells or percentage (with statistics) is provided.

## Methodology

Sample preparation

The first 10 cm of proximal small intestine from tamoxifen-induced mice was flushed in cold PBS, longitudinally opened, and cut into 0.5 cm segments. The segments were repeatedly passed through a 10 mL pipette using cold PBS until the solution ran clear, incubated in cold 5 mM EDTA/PBS for 30 min, washed, and resuspended in cold PBS. Crypt-enriched fractions were released by 5 × 5 s manual shaking cycles then pelleted and dissociated in 20 mL trypsin (1 × 0.05% EDTA, 25300054, Thermo Fisher Scientific) at 37 °C for 7 min with vigorous shaking every minute. After multiple washes in PBS/2% FBS, dissociated cells were resuspended in PBS/2% FBS before incubation with anti-mouse CD326 (EpCAM) AlexaFluor 647 antibody and addition of DAPI (10 µg/mL) to distinguish between live and dead cells.

Instrument

BD FACS Aria SORP (BD Biosciences)

Software

BD FACS DIVA 9.0.1 (BD Biosciences)

Cell population abundance

Epcam-AF647+ cells represented between 18-31% of parental population.

Gating strategy

Gating strategy shown in Supplementary Fig.6.

☒ Tick this box to confirm that a figure exemplifying the gating strategy is provided in the Supplementary Information.

