## [Peer Review File · Nature]

Decay of driver mutations shape the landscape of intestinal transformation

Corresponding Author: Dr Douglas Winton

Version 0:

Reviewer comments:

Referee #1

(Remarks to the Author)

Large-scale sequencing efforts documented the presence of cancer mutations being frequently present in otherwise normal looking epithelia, including gut. The widespread presence of somatic mosaicism is a topic that receives much interest, and the current manuscript by Lourenco et al. tries to address the question whether the presence of these mutant patches is of consequence for cancer initiation.

In the current manuscript, the authors generated a resource-intensive valuable dataset where various driver mutations are introduced in the intestinal epithelium of the mice (using Cre-LoxP). Next, the mutant yet normal looking epithelium receives a pulse of mutagenesis by ENU to score tumor formation (amount, phenotype, location). Many tumors are dissected and subjected to sequence analysis to identify the mutation that triggered transformation. Surprisingly, quite independent of the type of priming mutation, the secondary mutation that instigates tumor development is almost invariably APC loss or oncogenic B-catenin mutations. More specifically, the type of mutation in Bcat and APC seems to depend on the mutant background of the primed epithelia, suggestive of differential transformation capacity between point mutations that are context dependent. Lastly, the authors reverse the order of mutation. Due to the turn-over of the gut, most ENU-induced mutations are cleared and hence tumor incidents drop when the secondary mutation is induced after multiple rounds of epithelial self-renewal have occurred (20 to 30 days). However, for certain mutations, the drop is more than expected suggesting negative selection for the transformants.

The experiments described in the manuscript are of high quality. The manuscript is carefully written, but the topic that deals with loss of transformants (line 256 and further) is complex to understand due to the frequent use of probabilities, indirect assessments compared to theoretical outcomes, and wording (e.g. rescue?). This last part might be improved by more descriptive explanations.

Foremost, while I compliment the authors for the impressive amount of work, the outcome of all the work feels incremental rather than breakthrough. In addition, the work is almost entirely of descriptive nature, a clear take-home message remains elusive, and the paper provides no mechanistic understanding on key observations.

- Throughout the manuscript, the authors show that introducing WNT mutations by ENU in primed epithelia generate many more tumors than ENU-mediated WNT mutations in normal epithelium. The key question that remains unaddressed is whether the tumor cells-of-origin in mutant patches become intrinsically more susceptible for transformation, or whether the relative number of cells that became susceptible for transformation increased in absolute number. For instance, stem cells in normal crypts, versus the entire stem + progenitor cell compartment in mutant crypts. Mechanistic insights whether and how the cell-of-origins are rewired due to the primed mutation, or whether and how the population of potential cells-of-origins became enlarged, would improve the manuscript.

- Quantitative measurements of negative selection for transformed clones is of high interest. To assess negative bias, tumor numbers are compared to the amount of clones that are expected based on the 'decay of stem cell clones as they compete for niche occupancy of the crypt epithelium'. However, the authors make the assumption that stem cell competition in a highly mutagenized gut is identical to a normal epithelium. This needs to be demonstrated. Second, (stem) cell competition is likely altered once the priming mutation is introduced, affecting the probability by which the mutagenized cells can be

rescued.

- Tumors were macroscopically counted and dissected. Is there a possibility that certain subclasses of tumors have been missed due to alternative morphology and that a bias was introduced at the level of tumor identification/dissection? In addition, to get a comprehensive insight into the 'just right hypothesis', it is of interest to identify all WNT activating mutations that are possibly introduced, including those that are subjected to negative selection in vivo. Therefore, the authors may consider to generate organoid cultures of single intestinal cells isolated ~48hrs post ENU mutagenesis in the absence of WNT and Rspo. Sequencing analysis of APC and Bcat in WNT-independent surviving clonal organoids maybe of interest to compare with the in vivo datasets.

- Activating mutations of the WNT pathway, being APC loss or oncogenic B-catenin, are almost invariably the driver mutations for CRC development. However, there is no explanation why most of the identified tumors are driven by B-catenin mutations rather than APC loss as is mostly observed in patients.

- The observation that the type of priming mutation differentially affects transformation potential of different B-cat or APC mutation is of interest, but there is no experimental follow-up to elevate the observation into mechanistic understanding of cancer initiation.

- It is not investigated or discussed why WNT activating mutation other than APC and Bcat are not observed, in particular loss of Axin1. Alternatively, when field defects are created by oncogenic BRAF mutations, would RNF43 mutations be picked up?

- The pro-oncogenic fields that are studied in this manuscript involve many strong cancer mutations with clear effects within tumor progression. However, reports on the presence of these mutants in normal looking human gut epithelia are still scarce, but seem to pinpoint mostly to PIK3CA, ARID1A and FBXW7 as most clinically relevant. Can the authors address clinical relevance for the other studied alleles? (P53, KRAS, Notch).

- Since tumors were sequencing post PFA fixation, is there a change that fixation introduced mutational biases?

- The mutant clones that are introduced are uniformly present in the intestinal epithelium. As such, ENU mutagenesis is frankly applied to intestinal epithelia with 'germline' mutations rather than on sporadic local field defects. Can differences be excluded between cancer susceptibility in uniform mutant epithelia versus normal epithelium with mutant fields? (correct tumor multiplicity with fraction of mutant epithelium?)

Referee #2

(Remarks to the Author)

In this manuscript, Lourenço et al. investigated the impact of ENU-mediated mutagenesis on pre-existing fields of gut epithelial cells carrying various oncogenic mutations. ENU treatment was found to rapidly accelerate tumorigenesis in tamoxifen induced VilCreERT mice harboring mutations in Apc, Trp53, Fbxw7 and Pten alleles and gain-of-function mutations for KrasG12D, Pik3caH1047R and Notch1-ICD. To identify driver mutations, harvested tumors were subject to targeted hybridization capture, as well as amplicon or whole genome sequencing. As anticaptured mutations in Ctnnb1 or Apc were dominant. This analysis also revealed that priming events (i.e. introduction of fields of epithelial cells harboring known tumor promoting mutations) favored tumour development by Apc and Ctnnb1 mutation. In the case of Ctnnb1, most mutations affected exon 3 encoding the GSK-3 β and Casein Kinase-1 phosphorylation sites and specific mutations were found associated with specific exon 3 priming events.

Detailed analysis of the mutational signatures enriched in Ctnnb1 in comparison to all mutations suggested that exon 3 mutations of Ctnnb1 are a result of both mutational preferences presumably due to chemical constraints and positive selection. Furthermore, the transforming potential of specific exon 3 mutations also differed depending on the nature of the priming event. Similar analysis of the Apc mutational profile revealed that various priming events promoted mutations that led to the loss of the Beta-cat binding domain and retention of the dimerization domain and at least one Armadillo repeat. Finally, the authors examined the impact of post hoc induction of a priming events following chemical mutagenesis. These so-called rescue experiments resulted in a sharp decline in tumor incidence due to neutral drift or negative selection of transformants and therefore suggested that the order of mutations is a critical determinant of tumorigenesis.

General remarks: This is a thought-provoking study that attempts to address the significance of long held observations that normal self-renewing tissues can harbor an important number of cancer driver mutations. The first two figures provide limited conceptual novelty and primarily serve to describe the tumor model in great depth. The subsequent findings indicating that the type of priming events will impact the mutational preference in Apc and Ctnnb is novel but falls short of revealing mechanistic insight into why preferences are observed. The most important finding of the manuscript that many mutations are negatively selected is very interesting and calls into question previous assumptions that CRC proceeds via step wise mutations occurring independently and with neutral outcome. However, it's unclear how the experimental model of inducing fields of cancer drivers using a transgenic approach relates to the human context. Do permissive fields necessarily need to be hardwired through mutations or could there be epigenetic events that influence this process? How would permissive fields be created in humans if single mutations are rapidly lost and without consequence? Overall, although the manuscript puts forward important observations that must be taken into account when building a model of CRC tumorigenesis, this reviewer fails to see how the manuscript substantially changes our current view of this process. This point is further discussed below. In short, because of these limitations, I feel this manuscript would be best suited for a more specialized

journal. The following specific comments should be addressed:

The experimental model takes into account the mutational dynamics under homeostatic conditions and ignores a prime determinant of tumorigenesis being the inflammatory/regenerative state of the gut epithelium. In experimental models, it is well known that injury/inflammation can accelerate chemically-induced tumorigenesis. This begs the question as to whether certain mutations in cancer drivers like Apc that would normally be negatively selected in homeostatic contexts would also be in response to inflammation or injury. The experimental model described in this manuscript should take into account environmental factors as well.

In the intro the authors cite two distinct models of colon carcinogenesis (the classical stepwise model and the "big bang" theory, which implies multiple mutations occurring at the same time). Besides mentioning these models, it's not clear to this reviewer how the current findings relate to these models. Does the fact that single cancer driver mutations in Apc are typically lost from the tissue impose a requirement for multiple events to co-occur as is suggested in the big bang theory? Can the big bang model be validated in this experimental model?

Minor comment:

The representation of the data in Figures 5b, d, f, g as a line graph is confusing and suggests that all time points are from a single experiment. From my understanding of the figure, tumor counts are coming from different treatments i.e. TE and ET and not a continuous kinetic of a given treatment. In this case, it would be more appropriate to use a bar graph comparing TE vs ET treatments, as these are completely different treatments.

Referee #3

(Remarks to the Author)

This is a really interesting piece of work from an expert group in this field. The paper describes, in detail, using many novel descriptive tools, the phenotypic and genotypic consequences of key driver gene mutation ordering in the murine intestinal epithelium.

The key findings include

1. Different priming mutations bestow variable advantage to subsequent wnt disrupting mutations induced by chemical mutagenesis
2. In comparison with humans there is a clear proximal shift in the mutagenesis-induced Apc murine mutation cluster region
3. Timing of field induction either exacerbates or prevents impact of Wnt disrupting mutations arising from chemical mutagenesis
4. In the case of Fbxw7, mutation either protects against or exacerbates tumorigenesis depending on whether the mutation precedes or was subsequent to Apc mutation

All of these observations point to significant interaction and epistasis in acquisition of driver gene mutations in mouse models of CRC and are novel, interesting and important.

However the paper does not give significant functional insights as to how and why these important mechanistic observations arise.

Specific comments

1. The field effect of underlying priming driver mutations is assumed without being demonstrated, and this is reasonable given the use of villin Cre, which recombines effectively and in a non-chimeric fashion. However one way to check this would be to ensure that all primed tumors have the underlying Cre induced mutation, which may well have been done but isn't stated.
2. The different impact of priming driver mutations is fascinating but not expanded upon. In particular, priming p53 mutation gives rise to more highly mutated tumours but has lower tumour multiplicity than priming Kras. Given that the frequency of obligatory Apc/Ctnnb1 mutations is roughly the same as in tumors primed by other mutations, it is not clear whether these additional mutations are biologically relevant or simply accumulating passengers. Does the higher mutation burden but lower tumor multiplicity reflect negative selection through immune editing. Is there any evidence of differential immune infiltration into any of the different tumor types?
3. Line 129 - The text implies that these additional driver events were somatically acquired during tumor evolution, I assume these mutations were not present in any adjacent normal tissue?
4. There is clear evidence of a proximal shift in the ENU mutagenesis mutation cluster region (MCR) in the mouse Apc gene. This is consistent with the ENU derivation of the Min mouse (codon 850). Is this a consequence of ENU vulnerabilities in the proximal region of Apc or does it represent difference in murine/human biology? Are there any datasets of sporadic (non ENU induced) Apc mutations in mouse tumours that could be compared?
5. The data implicates a fascinating selection of different 'just right' wnt disrupting mutations dependent on the underlying priming mutation. I would be very interested in the authors thoughts as to how each priming mutation fine tunes the Wnt rheostat to have this effect. For example why does Kras priming switch wnt disrupting mutation selection to Ctnnb1 rather

than the predominance of Apc seen in the control cohort?

6. There is a slight issue with the TE and ET data being joined by line plots in Fig 5 as this implicates continuous datasets whereas the reality is that they are very different experiments. I appreciate that the TE comparison is required, but can the authors think of a modified way of demonstrating this?

6. The Kras rescue mutation essentially reverses the supercompetitor impact of Apc mutation, even imposing a significant degree of competitive advantage on non-Apc mutant cells. How does this have this profound effect? Does the Kras mutation induce stemness in non Lgr5 cells? could this be investigated?

7. Conversely Fbxw7 timing is variably protective or conducive to Wnt disrupting mutation dependent on mutation timing. Given that Fbxw7 could have very broad effect through altered ubiquitination can they speculate on a mechanism here - what does field Fbxw7 mutation do to stem cell dynamics?

Version 1:

Reviewer comments:

Referee #1

(Remarks to the Author)

All reviewers were unanimous that the work is of high quality. Moreover, all refs also indicated that the data regarding negative selection on mutant cells is of most interest. The new revised version includes new wording and additional experiments (mostly control and a comparison with human cancers).

Major criticism from the reviewers was the lack of mechanistic/functional insights, for example that explain how fitness barriers can be overcome, negative selection is imposed or how a priming mutation can shift the phenotypic outcome of a negative selected mutation in Apc 180degrees to a strong driver mutation.

The authors rebuttal that there is mechanistic insight in quantitative measurements. I agree with that statement and the value to chart and assess the different options (and likelihoods) for tumors to arise. That said, mechanistic understanding why the phenomena occur would have provided contextual meaning and tangibility, thereby elevating the manuscript beyond the current abstract and complex state that is difficult to digest and to understand.

Of note, the authors improved the embedding and context of their new work now a string of papers, including one from the main authors, documented how early intestinal lesions have a polyclonal origin when driven by Apc mutations. While the take home-message is still not easy to extract from the current manuscript, I got most excited about the text, statements and arguments in the accompanying intro of the rebuttal and wondered why that tone was not implemented throughout the manuscript.

All in all, the work is of high quality and the observed phenomena regarding the most likely path for a tumor to arise are of high interests. However, the abstract nature of the insights and difficulty to interpret, read and understand the text, makes this study in my honest opinion more suitable for a specialised journal.

Referee #2

(Remarks to the Author)

Overall, the authors have substantially improved the manuscript in the latest version. Most comments and questions were satisfactorily answered. Having said this the main criticism that emerged from the first round of revisions concerning mechanistic value of the manuscript remains a matter of debate. In the rebuttal the authors argue that mechanistic insight comes from quantifying negative biases affecting cancer drivers which in turn allows ranking of mutations based on their likelihood of occurring as primary drivers of tumour formation, without consideration of their biological function. Essentially as stated in the abstract: "Decay dynamics impose an effective order of mutation whereby negatively biased alleles are not bioavailable for subsequent mutation."

This reviewer would agree that this work provides an explanation for previous studies supporting a Big Bang type single clonal expansion driving CRC. Thus from this perspective, there is mechanistic value to this manuscript.

Where the manuscript is less convincing from a mechanistic point of view relates to its conclusions challenging the traditional, simplistic view that Wnt activating mutations are the primary event. Indeed, the manuscript argues the opposite, Wnt driver mutations are likely secondary to other hits that raise the ability of cells to survive and overcome the negative impact on cell fitness. Again, these are clearly important findings. But, beyond stating that "the order of mutation is proscribed that the strongest drivers are unlikely to occur first.". There are several specific points made throughout the manuscript that are left without explanation. They are all interesting findings worthy of further exploration but fall short in my view of providing real mechanistic insight.

Why do certain Apc mutations decay and what is the basis for the apparent cooperation between different priming mutations and Wnt driver mutations? Examples of these points include:

Line 219: Overall, the results suggest that different exon 3 mutations have different transformation potentials that are

dependent on the nature of the priming field.

Line 307: This indicates that the order of mutation dictates outcome with prior Fbxw7 mutation protecting against transformation by p.L35S and promoting it if mutated subsequently.

There is also some question marks regarding the interpretation of parts of the data. For instance, with regards to Fig 3d, the following is stated:

“Analysed by group, the control and Trp53null cohorts showed minimal deviation in O/E ratios while most primed groups displayed an under representation of A (excepting KrasG12D) and E domain mutations. This suggests that priming promotes transformation by Apc mutations that have retained both the dimerization and Apc protein domains (bin A) and at least one Armadillo (Arm) repeat (bin B) while having lost β -catenin-binding 20 amino acid (20AA) repeats (bin E).”

The data shows that 3 primed groups follow the trends mentioned above (i.e. Fbxw7, Pten and Apc) while 2 do not (trp53 and Kras). Thus, the under representation of bin A seems highly dependent on the type of priming event. Stating that most primed groups behave this way is an over interpretation. Secondly, it's not clear what the asterisks refer to in Fig 3d. Are the bars without asterisks not significant? For instance, the only priming events which show significant underrepresentation of bin A is Pten with a p value of 0.052. It's not clear whether the differences with Apc and Fbxw7 are significant. Also how is significance measured in this case? Are the authors comparing O/E ratios in primed conditions vs unprimed conditions? What are the control conditions in this experiment?

Another point lacking clarity is the following: “Comparison of the distribution of mutations in tumours arising in different groups revealed that control, and combined Trp53null and Trp53het /N1-ICDhet primed groups tended to have a greater distance between N- and C-terminal hits and an overall wider distribution compared to that observed with Pten null priming (Fig. 3e,f).”

Besides the fact that this observation remains without explanation, it's not clear what data shows this and whether this is significant. One can understand the text but the actual demonstration of these results is difficult to follow in Fig 3.

Similar comments can be made for Fig 4g

Additional comments:

As commented on by reviewer 1, in some passages the manuscript is quite challenging to read. To improve flow and comprehension additional explanations are required in certain passages. Examples include:

The analysis performed in Fig 4i, j and accompanying explanations in the text are not clear to this reviewer. It's not clear what conclusions can be drawn from this analysis.

The analysis of the human cohort begins with the following assumption: “ Reasoning that temporal first hits, occurring before transformative positive selection, would more closely correspond to signature predictions than subsequent events...”

Why would first hits more closely correspond to signature predictions than 2nd hits? This is not immediately obvious to this reviewer. Without a clear explanation of this it is difficult to follow the rest of the analysis in this section.

Referee #3

(Remarks to the Author)

The authors have responded positively to my comments with additional data where required. The new human data is of great interest and strengthens the findings

I was intrigued by 2 interesting observations in the discussion and would be interested in the authors thoughts in rebuttal form only -

1. "However, CTNNB1 mutations are found at higher frequencies in polyps of patients which better aligns with the predominantly adenomatous lesions described here" - Can the authors speculate on why CTNNB1 lesions fail to progress to established cancers when the wnt disrupting mutation would seem to have 'done the job' in initiating the lesion

2. "Further, recent studies have ascribed a “supercompetitor” behaviour to clones arising by biallelic inactivation of Apc whereby growth advantage is conferred, in part, by inhibition of self-renewal in neighbouring wildtype cells. Notably, such analyses focus on the growth properties of surviving clones and do not directly address their overall survival probability. Considered together, these observations suggest that mutations conferring the greatest potential for transformation have the highest bar to survival with priming acting rheostatically to lower the latter" - The supercompetitor growth advantage has been ascribed functionally to the secretion of Notum. Given that Notum is a responsive negative regulator of wnt, one might expect a proximal APC mutation to cause higher Notum expression - is there any evidence of quantitative association of Notum expression with different APC mutations - i.e. does a "just right" level of Notum expression explain the "just right" Wnt effect or is this a predominantly a cell-intrinsic phenomenon?.

Version 2:

Reviewer comments:

Referee #1

(Remarks to the Author)

I gave the new manuscript, that is now in its third version, a new thorough read. I was again struck by the enormous valuable resource of the in vivo collected data points. To start a project like this requires dedication and conviction as it cannot have been an easy task to collect. As such, the scale of the project reflects the uniqueness of the dataset which I am certain to study again and again over the years to come.

Along the suggestions of the reviewers, the authors have improved the manuscript text in clarity. That said, it remains very difficult to digest, that for a large part can be attributed to the abstract nature of the topic.

One of the original criticisms was lack of mechanistic insights to accompany the descriptive nature of the work, among others to provide contextual meaning to enhance interpretation of the data. Such mechanistic insights are now provided mostly in text to embed multiple noteworthy findings with interpretation and discussion.

All in all, I think that the current version is suitable for publication. The unique strength is its valuable dataset of in vivo work that is unmatched by any other initiative that I know. As mentioned earlier, I would not be surprised that this manuscript will be regularly studied as a reference work by scholars and colleagues to extract its wealth of data.

Minor comments:

In the 4th sentence of the summary, the authors state that early-stage CRCs comprise multiple clones, but in fact they should refer to premalignant lesions or adenomas.

Also, the added value of referencing the Big Bang model in the introduction is not clear and creates confusion, as it was originally proposed to describe the neutral growth expansion post-malignant transformation and is unlike supercompetitor behavior.

In the introduction, please change CTNNB1 oncoprotein into beta-catenin as in the rest of the manuscript. Perhaps to avoid confusion for non-experts, add the gene name behind in brackets.

Referee #2

(Remarks to the Author)

The authors have addressed my specific comments. I have no additional concerns.

Referee #3

(Remarks to the Author)

The authors have responded to my questions thoughtfully. No further queries from me

Authors: General response

We are grateful for the Reviewers' insights and critique. Before responding to their specific comments there are some general responses and also developments to report.

Although appreciative of the study the reviewers provide similar feedback that our observations though important are not adequately linked to a bigger picture in terms of the implications for cancer initiation. We acknowledge this deficiency and have made substantial changes to correct it. However, since submission of the manuscript there have been developments in our understanding of the early stages of transformation leading to CRC. A number of publications, including a precursor paper to the current one¹, have documented that a significant proportion of early intestinal lesions in both mouse and human have a polyclonal origin, in particular when driven by *Apc* truncating mutations¹⁻⁴. In contrast tumours arising on a background of *Kras* activation are mainly monoclonal. In many respects these recently described phenomena are counterintuitive with the accepted aetiology of CRC. These recent developments provide important context for the current study and are now referenced in Introduction (**lines 56-61**). *Apc* truncating mutations, commonly thought to be the earliest event leading to CRC, have largely been perceived as strong drivers because *Apc* loss autonomously promotes clonal growth while suppressing that of wildtype neighbours (acting as a super competitor)⁵⁻⁷; a behaviour consistent with BigBang single clonal expansions in CRC. The recent papers documenting polyclonality in human polyps provide no mechanistic insight or explanation for the phenomenon. Our observations using the model employed here established there was a recruitment process whereby very N-terminal truncations in *Apc* are recruited by adjacent lesions with more C-terminal truncations¹. This provided a clue that not all *Apc* mutations identified as drivers are equal with some strong drivers requiring cooperation from others. In sum, there is now uncertainty about the order of events that can initiate tumour growth. Our revised submission emphasises the following insights:

- Many of the most Wnt activating events that are the strongest drivers of tumour growth undergo active negative selection in a naïve epithelial tissue
- Priming with a range of driver events allows Wnt drivers to become not only positively selected but to initiate tumour growth
- Consequently, the order of mutation acquisition is in large part proscribed with the strongest Wnt drivers unlikely to be first.
- The mode of growth provided by priming favours a BigBang type single clonal expansion that arises because of the proscribed order of events.

These insights are made in the context of a mouse model and highlight where human studies may focus in the future. However, inspired by the reviewers' critique we have also identified features of available human CRC and polyp data that are consistent with both priming and negative biases and that we hope will provide further motivation for such studies.

We hope that the comments above demonstrate that our analyses are inherently mechanistic in terms of describing clonal biology if not the pathways that accompany

it. More formally, our lab focus has been to devise strategies that measure and compare changes in clonal competition between clonogenic cells within individual intestinal glands that are conferred by different somatic events. One challenge in deriving metrics that apply to mutations of genes that are inherently different in the pathways and processes they regulate is the ability to directly compare biases irrespective of how they are mediated (e.g. whether they promote proliferation or impact apoptosis and survival.) A second challenge arises from the acceptance in the stem cell field that self-renewal is not just a function of bone fide stem cells but is also a result of extensive plasticity of their descendants such that they can contribute to the stem cell pool and, in the gut, intra-gland competition. As reviewer 1 points out this effect may be compounded by mutations acting to promote or inhibit such plasticity. Essentially our approach is to integrate all these variables by determining the outcomes of clonal competition.

Therefore, our response is that an understanding of mechanism has to start with an understanding of how the outcomes of clonal competition are quantitatively dictated by the glandular architecture of the crypt to determine the bioavailability of cancer driver events for subsequent transformation.

Referees' comments:

Referee #1 (Remarks to the Author):

Large-scale sequencing efforts documented the presence of cancer mutations being frequently present in otherwise normal looking epithelia, including gut. The widespread presence of somatic mosaicism is a topic that receives much interest, and the current manuscript by Lourenco et al. tries to address the question whether the presence of these mutant patches is of consequence for cancer initiation. In the current manuscript, the authors generated a resource-intensive valuable dataset where various driver mutations are introduced in the intestinal epithelium of the mice (using Cre-LoxP). Next, the mutant yet normal looking epithelium receives a pulse of mutagenesis by ENU to score tumor formation (amount, phenotype, location). Many tumors are dissected and subjected to sequence analysis to identify the mutation that triggered transformation. Surprisingly, quite independent of the type of priming mutation, the secondary mutation that instigates tumor development is almost invariably APC loss or oncogenic B-catenin mutations. More specifically, the type of mutation in Bcat and APC seems to depend on the mutant background of the primed epithelia, suggestive of differential transformation capacity between point mutations that are context dependent. Lastly, the authors reverse the order of mutation. Due to the turn-over of the gut, most ENU-induced mutations are cleared and hence tumor incidents drop when the secondary mutation is induced after multiple rounds of epithelial self-renewal have occurred (20 to 30 days). However, for certain mutations, the drop is more than expected suggesting negative selection for the transformants.

The experiments described in the manuscript are of high quality. The manuscript is carefully written, but the topic that deals with loss of transformants (line 256 and further) is complex to understand due to the frequent use of probabilities, indirect

assessments compared to theoretical outcomes, and wording (e.g. rescue?). This last part might be improved by more descriptive explanations.

We thank the Reviewer for these comments. The text describing rescue has been amended to present actual numbers and not just probabilities and to expand on the terms used (lines 279-281; 332-334).

Foremost, while I compliment the authors for the impressive amount of work, the outcome of all the work feels incremental rather than breakthrough. In addition, the work is almost entirely of descriptive nature, a clear take-home message remains elusive, and the paper provides no mechanistic understanding on key observations.

We appreciate the recognition of the amount of work reported in the study and also the concerns. Space constraints made it difficult to capture the rationale (as outlined above) in the original submission when much of the background had not yet been published. Additional context and interpretation have been included in revision in both Introduction (lines 56-61; 66-69) and Discussion (lines 440-447; 463-471; 480-490).

As specified in our opening comments we believe there is mechanistic insight in quantifying negative biases affecting cancer drivers. This is because they, irrespective of how mediated, allow such effects to be ranked to indicate whether they are more or less likely to occur as primary drivers of tumour formation.

- Throughout the manuscript, the authors show that introducing WNT mutations by ENU in primed epithelia generate many more tumors than ENU-mediated WNT mutations in normal epithelium. The key question that remains unaddressed is whether the tumor cells-of-origin in mutant patches become intrinsically more susceptible for transformation, or whether the relative number of cells that became susceptible for transformation increased in absolute number. For instance, stem cells in normal crypts, versus the entire stem + progenitor cell compartment in mutant crypts. Mechanistic insights whether and how the cell-of-origins are rewired due to the primed mutation, or whether and how the population of potential cells-of-origins became enlarged, would improve the manuscript.

*We accept that there is an inherent interest in whether phenotypic stem cells are changed in mutant patches and have now performed a detailed profiling of these using *Lgr5* in situ hybridisation (RNAscope) for the priming fields providing the bulk of the data for the study (**Rebuttal Figure 1**). This shows that these are either unaltered in *Kras*^{G12D}, *Pten*^{null} and *Trp53*^{null} or increased in *Fbxw7*^{null} or *Apc*^{Het}.*

However, we would reference our introductory text about plasticity and emphasise that assessment of clonal competition is independent of the cells that are mediating it. The dynamics of the process are the same irrespective of the identity of cells marked by sporadic clonal level lineage tracing. For example, the clone dynamics following stem cell directed marking is identical to that obtained by marking progenitors that subsequently revert to a stem cell state⁹. This is because the dynamics are dictated by the cells competing for limited space within the crypt base. As the fold changes in tumour number with priming cannot be explained by the alterations in stem cell number the cells within such fields must be intrinsically more susceptible to transforming events mediated by ENU.

Rebuttal Figure 1 – *Lgr5* in situ hybridisation in small intestine crypts

Representative image and boxplots displaying *Lgr5* in situ hybridization (RNAscope) spot counts per crypt in relation to tamoxifen treatment for each cohort. Scale bar = 20 μ m. *P* values were calculated using a two-sample t-test with Welch's correction. All boxplots display the median and interquartile range (IQR; box bounds), with whiskers extending to most extreme points ($\leq 1.5 \times$ IQR) and all individual data points are shown.

- Quantitative measurements of negative selection for transformed clones is of high interest. To assess negative bias, tumor numbers are compared to the amount of clones that are expected based on the 'decay of stem cell clones as they compete for niche occupancy of the crypt epithelium'. However, the authors make the assumption that stem cell competition in a highly mutagenized gut is identical to a normal epithelium. This needs to be demonstrated.

We reference a previous lineage tracing study in mice with an MMR deficient background where the frequency of mutations arising spontaneously was increased to near 400-fold without impacting on clone dynamics⁹.

We also accept that a mutational pulse of ENU may have additional effects and have performed the requested analysis, now included as (Extended Data Fig. 12). The frequency and distribution of cell proliferation is identical in ENU treated and control animals after 10 days indicating repeat recovery at the earliest time of tamoxifen treatment after ENU. Also clone size distributions in ENU mutagenized intestines were assessed using the Confetti model at 21 days post induction of clones because this time point is optimal in capturing the wide distribution of clone sizes for comparison to control tissue. As shown, the distributions and average clone sizes are very similar indicating that clone dynamics and therefore inferred decay rates are not influenced by mutational burden.

Second, (stem) cell competition is likely altered once the priming mutation is introduced, affecting the probability by which the mutagenized cells can be rescued.

Yes, it is true that the rescue conditions change both the mutagenized cell and the background context in which it is competing. However, these are also the conditions under which priming is performed and in which mutagenized cells are transforming. Also, the informative decay period occurs prior to rescue of mutagenized cells when the context is the steady state.

- Tumors were macroscopically counted and dissected. Is there a possibility that

certain subclasses of tumors have been missed due to alternative morphology and that a bias was introduced at the level of tumor identification/dissection?

Tumour counts were performed on wholemount preparations but for all models tissue wholemounts were embedded in paraffin wax for sectioning and H&E assessment. This confirmed that all tumours were adenomatous and polypoid and, for example, no flat lesions were present (Extended Data Fig. 2d).

In addition, to get a comprehensive insight into the 'just right hypothesis', it is of interest to identify all WNT activating mutations that are possibly introduced, including those that are subjected to negative selection in vivo. Therefore, the authors may consider to generate organoid cultures of single intestinal cells isolated ~48hrs post ENU mutagenesis in the absence of WNT and Rspo. Sequencing analysis of APC and Bcat in WNT-independent surviving clonal organoids maybe of interest to compare with the in vivo datasets.

The reviewer proposes an interesting approach. Organoids were generated 48h post ENU as suggested and cultured in either minimally supplemented EN media (with EGF and Noggin but no R-spondin or Wnt (or the Wnt activating Chir99021)) or in ENRC media containing EGF, Noggin, R-spondin and Chir. Organoids in minimally supplemented media failed to grow while some of those in ENRC media grew (Rebuttal Figure 2). Around 7.5×10^5 cells were seeded (approximating to around 3,750 crypts). By comparison to the number in tumours induced with ENU and KrasG12D priming this might contain around 6 potential transformants that were not recovered in minimal media. The experiment was repeated with the same outcome when longer times between ENU treatment and culture were used to maximise recovery from any ENU induced toxicity and increase expression time for any ENU induced phenotype. Finally, cultures were established initially with ENRC supplementation and then transferred to the EN media. In all cases organoids never established or propagated without ENRC media.

It seems Wnt naïve conditions are insufficient to recover cells with Apc truncating, or b-catenin Exon 3 missense, mutations. However, sequencing of the organoids growing with ENRC supplementation identified 10 candidate driver mutations (8 from samples at 96h post-ENU and 2 from a single sample at 48h) only one of which (Apc Q12) was not observed in tumours indicating a good correspondence between drivers identified from organoids and in tumours.*

Rebuttal Figure 2 - Organoid culture from ENU treated mice

a, Experimental outline. **b**, Representative images of organoids cultured in different growth media at varying times post-ENU. **c**, OncoPrint showing driver mutations identified in *Apc* and *Ctnnb1* across samples. Each sample was sequenced twice as technical replicates, denoted by a common sample number with a letter suffix (e.g., Sample 1_A and 1_B). Mean read depth = 22,502. VAF threshold = 0.005.

- Activating mutations of the WNT pathway, being APC loss or oncogenic B-catenin, are almost invariably the driver mutations for CRC development. However, there is no explanation why most of the identified tumors are driven by b-catenin mutations rather than APC loss as is mostly observed in patients.

Whether drivers show positive or negative biases is context dependent, as shown for different primed fields. We accept this will include different species context and this may account for different dependencies on Apc and β -catenin mutation. That said, the mouse model used in the study is better aligned with early-stage disease and β -catenin mutations are more frequently observed in polyps than in CRCs. In a recent study analysing treatment naive precancerous lesions (polyps)⁴, CTNNB1 mutation frequency was 8.5%. In the HTAN polyp cohort (Cbio portal) frequency was 7%. In late-stage disease, CTNNB1 driver mutation frequencies are ~ 2.5% (Intogen database–COADREAD). APC frequencies were similar between early- and late-stage disease, underscoring differences in progression potential. The text has been amended to include these points (lines 422-424).

More generally, we believe the value of our study comes from defining negative selection as a process for many cancer driver events. The particular outcome for human colon favours APC mutations.

- The observation that the type of priming mutation differentially affects transformation potential of different β -cat or APC mutation is of interest, but there is no experimental

follow-up to elevate the observation into mechanistic understanding of cancer initiation.

We apologise for failing to connect observations to the implications for cancer initiation for the reasons stated above. We believe this insight comes at two levels: in terms of polyclonal origins Apc mutant clones that are recruited to participate in tumour formation are now identified as being under pronounced negative selection but are potent drivers in appropriate primed contexts; the order of mutation is proscribed that the strongest drivers are unlikely to occur first. Text has been amended to make this more explicit (lines 463-471; 480-490)

These are insights that come from the mouse model employed but we hope provide motivation for investigating the phenomenon in the human disease. Indeed, our approach of deconvoluting the contribution of mutational processes from selection by measuring the departure of observed mutations from those predicted by mutational signatures seemed a useful starting point for analysis of available human data. This signature has been described for the normal human colonic epithelium and is dominated by the clock type signatures associated with ongoing cell proliferation. We now present preliminary supporting data from such an analysis of human CRC and polyp data and identify novel features that are consistent with both the negative biases in APC N-terminal truncations and priming effects in cancers with both APC and KRAS mutations (Figure 5, Extended Data Fig.14, main text (lines 349-410; 440-447)).

- It is not investigated or discussed why WNT activating mutation other than APC and Bcat are not observed, in particular loss of Axin1. Alternatively, when field defects are created by oncogenic BRAF mutations, would RNF43 mutations be picked up?

Our study had a wide scope and was relatively unbiased in terms of the different fields tested and the use of a broad-spectrum mutagen with the potential to activate many potential transforming events. Despite this it was Apc and β -catenin mutations that predominated as transformants and inevitably our focus was on these two transforming events. There are undoubtedly many more contexts for priming that might favour different transformants such as those posited by the reviewer. Also, certain transforming events may be preferentially induced by other types of events than SNVs, for example chromosome rearrangements that result in PTPRK-RSPO3 fusion. Indeed, we hope that the current study provides motivation for using the approach for hypothesis testing of other disease relevant scenarios of the type highlighted by the reviewer. These points have now been included in discussion (lines 480-490)

- The pro-oncogenic fields that are studied in this manuscript involve many strong cancer mutations with clear effects within tumor progression. However, reports on the presence of these mutants in normal looking human gut epithelia are still scarce, but seem to pinpoint mostly to PIK3CA, ARID1A and FBXW7 as most clinically relevant. Can the authors address clinical relevance for the other studied alleles? (P53, KRAS, Notch).

The reviewer is correct that the catalogue of observed CRC drivers in normal gut epithelia is scarce and reports few in number. However, there is a substantial literature documenting the presence of Kras activating mutations^{10,11} and of Tp53 mutations¹².

(Notch mutations are relatively rare in CRC but its overexpression is often mediated by non-mutational mechanisms). In respect of ARID1A and PTEN we have, since submission of this manuscript, recently published on the positive biases associated with their loss¹³ and have updated referencing to include it.

The recent advances in determining mutational signatures for the renewing epithelia highlight the dominance of the clock type signatures associated with ongoing cell proliferation. The high somatic mutation rate must result in activation of many potential cancer driver events. What is unknown, excepting the examples above, is their subsequent fate and representation within the tissue and the consequences of their presence for subsequent transformation- the issue we are trying to address in the study.

- Since tumors were sequencing post PFA fixation, is there a change that fixation introduced mutational biases?

It is certainly possible, even likely, that at the level of individual DNA templates that fixation artefacts could be read as a variant. However, this is sporadic and adds noise at a level that does not compromise making calls of the tumour fraction of excised tissues as these are always at consistently high VAF.

- The mutant clones that are introduced are uniformly present in the intestinal epithelium. As such, ENU mutagenesis is frankly applied to intestinal epithelia with 'germline' mutations rather than on sporadic local field defects. Can differences be excluded between cancer susceptibility in uniform mutant epithelia versus normal epithelium with mutant fields? (correct tumour multiplicity with fraction of mutant epithelium?)

We agree that our model represents an exaggerated scenario. It was selected to provide certainty about the origin of tumours from within the field of interest, although unlike germline models the stromal components remain wildtype. The approach also allows direct comparison across priming events without concerns about the size of fields associated with different priming events, as harmonizing and quantifying mosaicism levels in different mouse models would be challenging. Conceptually, sporadic fields will be of different patch sizes depending on the degree of positive biases conferred by the priming mutation and edge effects, where primed patches meet adjacent wildtype epithelium, might show attenuated responses compared to those observed in crypts surrounded by other primed crypts. Consequently, we view our approach as a reasonable approximation of behaviours in the middle of sporadic expansions. The sporadic nature of the fields being modelled is now made explicit (lines 413-416)

*To qualitatively address the reviewer's concern, we adjusted the tamoxifen dose from 4 mg to 0.15 mg for the sequencing cohort. Despite this lower dose, mice still developed a large number of tumours while exhibiting increased survival (**Rebuttal Figure 3**).*

Rebuttal Figure 3 - Effect of tamoxifen induction dose in $Kras^{G12D}$ (Tam>ENU) cohort

a, Box plot of tumour counts per mouse in relation to Cohort. Mean \pm SD: Control (3.1 ± 2.7), $Kras^{G12D}$ 4 mg (920.7 ± 184.5), $Kras^{G12D}$ 0.15 mg (40.3 ± 37.5). N mice per cohort indicated in plot. Box plot displays the median and interquartile range (IQR; box bounds), with whiskers extending to most extreme points ($\leq 1.5 \times$ IQR) and all individual data points are shown. P value was calculated using ANOVA followed by Tukey's HSD for pairwise comparisons. **b**, Kaplan-Meier survival plot stratified by Cohort. Median post-ENU survival (days): 4 mg (37), 0.15 mg (101). P value was calculated using a log-rank test.

Referee #2 (Remarks to the Author):

In this manuscript, Lourenço et al. investigated the impact of ENU-mediated mutagenesis on pre-existing fields of gut epithelial cells carrying various oncogenic mutations. ENU treatment was found to rapidly accelerate tumorigenesis in tamoxifen induced *VilCreERT* mice harboring mutations in *Apc*, *Trp53*, *Fbxw7* and *Pten* alleles and gain-of-function mutations for $Kras^{G12D}$, *Pik3caH1047R* and *Notch1-ICD*. To identify driver mutations, harvested tumors were subject to targeted hybridization capture, as well as amplicon or whole genome sequencing. As anticaptured mutations in *Ctnnb1* or *Apc* were dominant. This analysis also revealed that priming events (i.e. introduction of fields of epithelial cells harboring known tumor promoting mutations) favored tumour development by *Apc* and *Ctnnb1* mutation. In the case of *Ctnnb1*, most mutations affected exon 3 encoding the GSK-3 β and Casein Kinase-1 phosphorylation sites and specific mutations were found associated with specific exon 3 priming events.

Detailed analysis of the mutational signatures enriched in *Ctnnb1* in comparison to all mutations suggested that exon 3 mutations of *Ctnnb1* are a result of both mutational preferences presumably due to chemical constraints and positive selection. Furthermore, the transforming potential of specific exon 3 mutations also differed depending on the nature of the priming event. Similar analysis of the *Apc* mutational profile revealed that various priming events promoted mutations that led to the loss of the Beta-cat binding domain and retention of the dimerization domain and at least one Armadillo repeat.

Finally, the authors examined the impact of post hoc induction of a priming events following chemical mutagenesis. These so-called rescue experiments resulted in a sharp decline in tumor incidence due to neutral drift or negative selection of

transformants and therefore suggested that the order of mutations is a critical determinant of tumorigenesis.

General remarks: This is a thought-provoking study that attempts to address the significance of long held observations that normal self-renewing tissues can harbor an important number of cancer driver mutations. The first two figures provide limited conceptual novelty and primarily serve to describe the tumor model in great depth. The subsequent findings indicating that the type of priming events will impact the mutational preference in Apc and Ctnnb is novel but falls short of revealing mechanistic insight into why preferences are observed. The most important finding of the manuscript that many mutations are negatively selected is very interesting and calls into question previous assumptions that CRC proceeds via step wise mutations occurring independently and with neutral outcome. However, it's unclear how the experimental model of inducing fields of cancer drivers using a transgenic approach relates to the human context.

*We thank the reviewer for these thoughtful comments. It is clear that our original submission fell short in linking our observations to their implications for development of colorectal cancers, as discussed above, and that we have attempted to address in revision. In response to the point about the first two figures we have merged these into a single main figure (**Figure 1**) with the remaining panels added into Extended Data Figures (**Extended Data Figs. 2 and 4**).*

Do permissive fields necessarily need to be hardwired through mutations or could there be epigenetic events that influence this process?

*Our choice of a model, in which fields are created by Cre-recombination and SNVs by a mutagen, was essentially pragmatic allowing these to be directly correlated and for transforming events to be easily identified and tracked. We entirely accept that other stable sources of somatic variation, including epigenetic changes and inflammation, could provide permissive fields to alter the selection of transforming events leading to neoplastic disease. We have included an explicit statement to this effect in Discussion (**lines 480-490**).*

How would permissive fields be created in humans if single mutations are rapidly lost and without consequence?

The implication would seem to be that single driver mutations can be neutral or positively selected, as previously assumed, or negatively selected as we now show. Our selection of candidate drivers to test as potential priming fields was based in part on our own observations that KRAS activating mutations are positively selected in the human epithelium¹⁰. Also, as mentioned, above we have recently published that heterozygous loss of PTEN and ARID1A similarly show positive selection¹³.

Overall, although the manuscript puts forward important observations that must be taken into account when building a model of CRC tumorigenesis, this reviewer fails to see how the manuscript substantially changes our current view of this process. This point is further discussed below. In short, because of these limitations, I feel this

manuscript would be best suited for a more specialized journal. The following specific comments should be addressed:

We accept the criticism as discussed in our general comments, above. To reiterate we believe that the insights come from a revised view of the series of steps leading to CRC. These are:

- Many of the most Wnt activating events that are the strongest drivers of tumour growth undergo active negative selection in a naïve epithelial tissue*
- Priming with a range of driver events allows Wnt drivers to become not only positively selected but to initiate tumour growth*
- Consequently, the order of mutation acquisition is in large part proscribed with the strongest Wnt drivers unlikely to be first.*
- The mode of growth provided by priming favours a BigBang type single clonal expansion that arises because of the proscribed order of events.*

These insights are now clearly highlighted in the revised manuscript in both Introduction (lines 56-61; 66-69) and Discussion (lines 440-447; 463-471; 480-490).

The experimental model takes into account the mutational dynamics under homeostatic conditions and ignores a prime determinant of tumorigenesis being the inflammatory/regenerative state of the gut epithelium. In experimental models, it is well known that injury/inflammation can accelerate chemically-induced tumorigenesis. This begs the question as to whether certain mutations in cancer drivers like Apc that would normally be negatively selected in homeostatic contexts would also be in response to inflammation or injury. The experimental model described in this manuscript should take into account environmental factors as well.

We accept that other contexts than homeostasis are relevant and would change the selection preferences for many drivers. However, the bulk of CRCs arise sporadically in the absence of pronounced inflammation, and this motivated our choice of model. It is notable that APC mutations occur at low frequency (20%) and are a late event in CRCs arising in patients with ulcerative colitis suggesting that there may be no switch from negative to positive bias for initiation of CRCs with this aetiology. Clearly this is of interest but would ask that investigating it be viewed as beyond the scope of the current study.

In the intro the authors cite two distinct models of colon carcinogenesis (the classical stepwise model and the “big bang” theory, which implies multiple mutations occurring at the same time). Besides mentioning these models, it’s not clear to this reviewer how the current findings relate to these models. Does the fact that single cancer driver mutations in Apc are typically lost from the tissue impose a requirement for multiple events to co-occur as is suggested in the big bang theory? Can the big bang model be validated in this experimental model?

The identification from multi-region sampling of CRCs that multiple cancer driver events are truncal mutations arising as a single expansion from a last common ancestor (the big bang theory) leaves unaddressed both the nature of that ancestor

and the route by which it has acquired multiple mutations. The current study indicates that positively selected driver events change the selection landscape for drivers that would otherwise be negatively selected.

Moreover, our observation that the most negatively biased Apc mutations are those with the greatest transforming potential explains why there is a single tumour-forming expansion. The order of events is proscribed because of the requirement for negatively biased Apc mutations to co-occur with other drivers.

*As regards validating the Big Bang interpretation we can point to our previously submitted study, now published¹, on clonality using confetti lineage tracing that allowed the number of driver events detected by sequencing to be aligned with the clonal status of individual tumours. Extending that approach in the current study to tumours arising with diverse priming events shows that priming predominantly promotes monoclonal tumours (**Figure 1i**) - a prerequisite of the Big Bang theory, a point made explicit in the revised manuscript (**lines 480-490**).*

Of note, our previous study documented the exception that priming with heterozygous loss of Apc promoted a polyclonal origin in a significant proportion of tumours. Clones containing more N-terminal Apc truncations were recruited by being adjacent to clones with more C-terminal truncations. Taken together this demonstrates that Wnt drivers negatively selected in homeostasis require priming either in cis or in trans.

Minor comment:

The representation of the data in Figures 5b, d, f, g as a line graph is confusing and suggests that all time points are from a single experiment. From my understanding of the figure, tumor counts are coming from different treatments i.e. TE and ET and not a continuous kinetic of a given treatment. In this case, it would be more appropriate to use a bar graph comparing TE vs ET treatments, as these are completely different treatments.

*We accept this criticism also raised by Reviewer 3 and have amended the plots in the revised manuscript (**Figure 4, Extended Data Figs. 11 and 13**).*

Referee #3 (Remarks to the Author):

This is a really interesting piece of work from an expert group in this field. The paper describes, in detail, using many novel descriptive tools, the phenotypic and genotypic consequences of key driver gene mutation ordering in the murine intestinal epithelium.

The key findings include

1. Different priming mutations bestow variable advantage to subsequent wnt disrupting mutations induced by chemical mutagenesis
2. In comparison with humans there is a clear proximal shift in the mutagenesis-induced Apc murine mutation cluster region
3. Timing of field induction either exacerbates or prevents impact of Wnt disrupting mutations arising from chemical mutagenesis

4. In the case of Fbxw7, mutation either protects against or exacerbates tumorigenesis depending on whether the mutation precedes or was subsequent to Apc mutation

All of these observations point to significant interaction and epistasis in acquisition of driver gene mutations in mouse models of CRC and are novel, interesting and important.

However the paper does not give significant functional insights as to how and why these important mechanistic observations arise.

We accept the criticism as discussed in our general comments, above. To reiterate we believe that the insights come from a revised view of the series of steps leading to CRC are:

- Many of the most Wnt activating events that are the strongest drivers of tumour growth undergo active negative selection in a naïve epithelial tissue*
- Priming with a range of driver events allows Wnt drivers to become not only positively selected but to initiate tumour growth*
- Consequently, the order of mutation acquisition is in large part proscribed with the strongest Wnt drivers unlikely to be first.*
- The mode of growth provided by priming favours a BigBang type single clonal expansion that arises because of the proscribed order of events.*

These insights are now clearly highlighted in the revised manuscript in both Introduction (lines 56-61; 66-69) and Discussion (lines 440-447; 463-471; 480-490).

Specific comments

1. The field effect of underlying priming driver mutations is assumed without being demonstrated, and this is reasonable given the use of villin Cre, which recombines effectively and in a non-chimeric fashion. However one way to check this would be to ensure that all primed tumors have the underlying Cre induced mutation, which may well have been done but isn't stated.

We have addressed this concern for many of the primed groups, namely $Kras^{G12D}$, $Fbxw7^{null}$, $Pten^{null}$ and $Trp53^{null}$ (Extended Data Fig. 3).

For priming with $Kras^{G12D}$ and Fbxw7 loss we have confirmed the presence of the recombined alleles in DNA extracted from excised tumours as requested. This also showed the presence of unrecombined DNA that is expected due to the presence of normal stromal cells in the tumour mass. To confirm that normal cells must be non-epithelial, $Kras^{G12D}$ and Fbxw7 mice intercrossed with VillinCreERT were treated with 4mg of tamoxifen and the DNA from purified epithelial (Epcam+) cells analysed for recombination status. This confirmed that no unrecombined allele was detectable in the epithelium.

For Trp53 recombination following tamoxifen treatment and in the tumours after ENU was investigated by in situ hybridisation. This revealed that around 90% crypts and

the epithelial component of all of six tumours analysed were negative for Trp53 expression.

For PTEN immunohistochemistry revealed no immunoreactivity in either epithelium or the tumour cells.

2. The different impact of priming driver mutations is fascinating but not expanded upon. In particular, priming p53 mutation gives rise to more highly mutated tumours but has lower tumour multiplicity than priming Kras. Given that the frequency of obligatory Apc/Ctnnb1 mutations is roughly the same as in tumors primed by other mutations, it is not clear whether these additional mutations are biologically relevant or simply accumulating passengers. Does the higher mutation burden but lower tumor multiplicity reflect negative selection through immune editing. Is there any evidence of differential immune infiltration into any of the different tumor types?

Indeed, the study has a broad scope and has revealed a diversity of outcomes and epistatic responses. It is the collective behaviour that we focussed on because we felt the impact of individual observations was secondary to the pronounced negative biases associated with potential transforming events.

*However, to answer the Reviewer's question on Trp53 mediated priming we performed dN/dS analysis on the data from this group. (**Rebuttal Figure 4a**). The analysis implicated no additional genes as being positively selected other than Apc and Ctnnb1 suggesting that the additional mutational burden is not due to a requirement for an additional driver. Moreover, if the increased tumour burden was important for Trp53 driven tumours, that would also be the case in the rescue experiment whereas they are in fact similar to the controls.*

*Directly visualising lymphocyte using CD3 immunodetection indicated relatively few infiltrating lymphocytes in Trp53 primed tumours. To more broadly compare the immune status of tumours arising from different primed backgrounds immune signatures were deconvoluted from a bulk RNAseq analysis. This suggested equivalent lymphoid infiltration in Trp53 mutant and other groups (notably an exception is Fbxw7 primed tumours which show a trend for increased lymphoid infiltration) (**Rebuttal Figure 4b-d**).*

Rebuttal Figure 4 – Genes under selection and T-cell profiling in *Trp53^{null}* cohort

a, dN/dS ratio and corresponding q-values for missense and nonsense mutations in the *Trp53^{null}* (Tam>ENU) cohort. Genes with statistically significant ratios are labelled. **b**, Representative Cd3 IHC images of adjacent normal and tumour tissue from the proximal small intestine of *Trp53^{null}* (Tam>ENU) mice, with higher magnification insets. Scale bars (left to right): 50, 20, 100, 20 μ m. **c**, Boxplot of Cd3+ cell density (cells/mm²) in relation to region. **d**, Mean normalized counts from whole-tumour RNA-seq experiment stratified by cohort and Cd3d-, Cd3e-, Cd3g-positive T cell population; n tumours indicated in legend. *P* value was calculated using ANOVA followed by Tukey's HSD for pairwise comparisons. Statistically significant pairwise comparisons shown. Asterisks denote statistical significance: (*:*P*<0.05; **:*P*<0.01; ***:*P*<0.001).

3. Line 129 - The text implies that these additional driver events were somatically acquired during tumor evolution, I assume these mutations were not present in any adjacent normal tissue?

Comparison of the variant allele frequencies with clonal Wnt-drivers Apc and Ctnnb1 shows comparable range and values (Rebuttal Figure 5) indicating they both reflect tumour representation in the sample. Consequently, these additional driver events are most likely generated in the tumour cell of origin by ENU.

Rebuttal Figure 5 – Variant allele frequency for driver genes

Variant allele frequency in relation to driver type. Other drivers: *Trp53*, *Kras*, *Pik3ca*, *Smad4*, *Fbxw7*, *Tcf7l2*, *Amer1*, *Braf*, *Sox9*, *Pten*. Wnt drivers: *Apc*, *Ctnnb1*. P value was calculated using a two-sample *t*-test with Welch's correction.

4. There is clear evidence of a proximal shift in the ENU mutagenesis mutation cluster region (MCR) in the mouse *Apc* gene. This is consistent with the ENU derivation of the Min mouse (codon 850). Is this a consequence of ENU vulnerabilities in the proximal region of *Apc* or does it represent difference in murine/human biology? Are there any datasets of sporadic (non ENU induced) *Apc* mutations in mouse tumours that could be compared?

The proximal shift is highlighted by consideration of low Observed/Expected values where the expectation is based on the ENU mutational signature. This shows the MCR is well populated with trinucleotide contexts that can be converted to truncating mutations by ENU (Extended Data Fig. 9a-c, Extended Data Fig. 10a). This indicates that the shift does arise from differences in murine versus human biology.

*However, seeking to address the Reviewer's query revealed very few studies that have performed *Apc* mutational profiling of sporadic mouse tumours and no datasets are available. We had not fully appreciated this. It seems that we are reporting a novel observation. We thank the reviewer for bringing this to our attention.*

5. The data implicates a fascinating selection of different 'just right' wnt disrupting mutations dependent on the underlying priming mutation. I would be very interested in the authors thoughts as to how each priming mutation fine tunes the Wnt rheostat to have this effect. For example why does *Kras* priming switch wnt disrupting mutation selection to *Ctnnb1* rather than the predominance of *Apc* seen in the control cohort?

*We thank the reviewer for the raising the rheostat concept. The rheostat likely operates within a range that has to be set low enough to be pro-survival and high enough at the upper end to allow transformation. With priming by both *Kras*^{G12D} activation and *Fbxw7* loss the rheostat increases the survival threshold for *Ctnnb1* mutation. This is supported by control mice having a low tumour burden where *Apc* mutations dominate over *Ctnnb1* mutations. The comparison of transformation efficiencies of *Ctnnb1* and *Apc* mutations at the upper end of the rheostat is more problematic because the latter requires two hits and the former only one. However, that also means that *Ctnnb1* mutations are more switch-like and *Apc*, due to the broad distribution of potential*

truncating mutations in both alleles, more like a true rheostat operating over a wide dynamic range.

Space constraints limit full inclusion of this topic in the submitted manuscript, but we have included the rheostat concept in Discussion (Lines 458-461).

6. There is a slight issue with the TE and ET data being joined by line plots in Fig 5 as this implicates continuous datasets whereas the reality is that they are very different experiments. I appreciate that the TE comparison is required, but can the authors think of a modified way of demonstrating this?

We acknowledge this was not optimal as also indicated by Reviewer 2 and the figures have now been modified to show tumours frequencies as bar plots (Figure 4, Extended Data Figs. 11 and 13).

6. The Kras rescue mutation essentially reverses the supercompetitor impact of Apc mutation, even imposing a significant degree of competitive advantage on non-Apc mutant cells. How does this have this profound effect? Does the Kras mutation induce stemness in non Lgr5 cells? could this be investigated?

To clarify, prior to rescue the Apc transformants show negative bias in their decay. Therefore, it is under homeostatic conditions that wild type cells have a competitive advantage over some Apc transformants with N-terminal truncations. The likely conclusion is that extreme supercompetitor behaviour is not 'just right' and the level of Wnt perturbation, despite neighbourhood suppression, is too high.

7. Conversely Fbxw7 timing is variably protective or conducive to Wnt disrupting mutation dependent on mutation timing. Given that Fbxw7 could have very broad effect through altered ubiquitination can they speculate on a mechanism here - what does field Fbxw7 mutation do to stem cell dynamics?

We originally co-submitted with another paper that dealt in depth with the different outcomes of ordering APC and FBXW7 in human organoids and felt that our observation of epistasis in the mouse model would best serve as orthogonal support of that study. In brief, the conclusion of the study, currently available as a pre-print was that FBXW7 loss preceding loss of APC established an epigenetic context whereby APC loss failed to significantly perturb gene expression compared to that expected, and observed, when the order is reversed¹⁴.

References

1. Sadien, I. D. *et al.* Polyclonality overcomes fitness barriers in Apc-driven tumorigenesis. *Nature* **634**, 1196–1203 (2024).
2. Islam, M. *et al.* Temporal recording of mammalian development and precancer. *Nature* **634**, 1187–1195 (2024).
3. Gaynor, L. *et al.* Crypt density and recruited enhancers underlie intestinal tumour initiation. *Nature* (2025) doi:10.1038/s41586-024-08573-9.

4. Lu, Z. *et al.* Polyclonal-to-monoclonal transition in colorectal precancerous evolution. *Nature* **636**, 233–240 (2024).
5. Flanagan, D. J. *et al.* NOTUM from Apc-mutant cells biases clonal competition to initiate cancer. *Nature* **594**, (2021).
6. van Neerven, S. M. *et al.* Apc-mutant cells act as supercompetitors in intestinal tumour initiation. *Nature* **594**, (2021).
7. Yum, M. K. *et al.* Tracing oncogene-driven remodelling of the intestinal stem cell niche. *Nature* **594**, (2021).
8. Tomic, G. *et al.* Phospho-regulation of ATOH1 Is Required for Plasticity of Secretory Progenitors and Tissue Regeneration. *Cell Stem Cell* **23**, 436–443.e7 (2018).
9. Christopher, J. *et al.* Quantifying Microsatellite Mutation Rates from Intestinal Stem Cell Dynamics in Msh2-Deficient Murine Epithelium. *Genetics* **212**, 655–665 (2019).
10. Nicholson, A. M. *et al.* Fixation and Spread of Somatic Mutations in Adult Human Colonic Epithelium. *Cell Stem Cell* **22**, (2018).
11. Parsons, B. L. *et al.* ACB-PCR Quantification of K- RAS Codon 12 GAT and GTT Mutant Fraction in Colon Tumor and Non-Tumor Tissue. *Cancer Invest* **28**, 364–375 (2010).
12. Matas, J. *et al.* Colorectal Cancer Is Associated with the Presence of Cancer Driver Mutations in Normal Colon. *Cancer Res* **82**, (2022).
13. Skoufou-Papoutsaki, N. *et al.* Haploinsufficient phenotypes promote selection of PTEN and ARID1A-deficient clones in human colon. *EMBO Rep* (2025) doi:10.1038/s44319-025-00373-0.
14. Chan, D. K. H. *et al.* Mutational order and epistasis determine the consequences of FBXW7 mutations during colorectal cancer evolution. Preprint at <https://doi.org/10.1101/2023.08.25.554836> (2023).

Referees' comments:

We gratefully acknowledge the reviewers' appreciation that the revised manuscript was much improved and also accept their feedback that more is required in respect of clarity and mechanistic insight. Our response has been shaped by their view, reinforced by the editor that in 'closing the gap' these are to a significant degree linked. Following the other editorial requirement, that the manuscript is more strictly configured to Nature formatting, has also shaped our response. The outcome is a more concise account with new Summary, Introduction and Discussion (**lines 22-34, 36-54 and 354-412**), with additional explanatory text in the Results, and changes in emphasis that focus on the core findings.

Referee #1 (Remarks to the Author):

All reviewers were unanimous that the work is of high quality. Moreover, all refs also indicated that the data regarding negative selection on mutant cells is of most interest. The new revised version includes new wording and additional experiments (mostly control and a comparison with human cancers).

Major criticism from the reviewers was the lack of mechanistic/functional insights, for example that explain how fitness barriers can be overcome, negative selection is imposed or how a priming mutation can shift the phenotypic outcome of a negative selected mutation in Apc 180degrees to a strong driver mutation.

The authors rebuttal that there is mechanistic insight in quantitative measurements. I agree with that statement and the value to chart and assess the different options (and likelihoods) for tumors to arise. That said, mechanistic understanding why the phenomena occur would have provided contextual meaning and tangibility, thereby elevating the manuscript beyond the current abstract and complex state that is difficult to digest and to understand.

The Referee here links the importance of mechanism and clarity in conveying the narrative, feedback that is re-stated more formally below. Indeed, the Referee's caveat in accepting that there is "mechanistic insight in quantitative measurements" arises in large part in respect to the lack of clarity, and accords with editorial prompting to make the narrative more accessible to a less specialist audience. In revision we have focussed on combining concision with inclusion of additional explanatory text to place findings in biological context.

Of note, the authors improved the embedding and context of their new work now a string of papers, including one from the main authors, documented how early intestinal lesions have a polyclonal origin when driven by Apc mutations. While the take home-message is still not easy to extract from the current manuscript, I got most excited about the text, statements and arguments in the accompanying intro of the rebuttal and wondered why that tone was not implemented throughout the manuscript.

Thank you. This comment provided a strong steer for us. The introduction has been extensively rewritten to comprise the more conceptual (and shorter) text previously included in the rebuttal document (lines 36-54).

All in all, the work is of high quality and the observed phenomena regarding the most likely path for a tumor to arise are of high interests. However, the abstract nature of the insights and difficulty to interpret, read and understand the text, makes this study in my honest opinion more suitable for a specialised journal.

We hope the revisions incorporated in response to feedback changes the Referee's view.

Referee #2 (Remarks to the Author):

Overall, the authors have substantially improved the manuscript in the latest version. Most comments and questions were satisfactorily answered. Having said this the main criticism that emerged from the first round of revisions concerning mechanistic value of the manuscript remains a matter of debate. In the rebuttal the authors argue that mechanistic insight comes from quantifying negative biases affecting cancer drivers which in turn allows ranking of mutations based on their likelihood of occurring as primary drivers of tumour formation, without consideration of their biological function. Essentially as stated in the abstract: "Decay dynamics impose an effective order of mutation whereby negatively biased alleles are not bioavailable for subsequent mutation."

This reviewer would agree that this work provides an explanation for previous studies supporting a Big Bang type single clonal expansion driving CRC. Thus from this perspective, there is mechanistic value to this manuscript.

Where the manuscript is less convincing from a mechanistic point of view relates to its conclusions challenging the traditional, simplistic view that Wnt activating mutations are the primary event. Indeed, the manuscript argues the opposite, Wnt driver mutations are likely secondary to other hits that raise the ability of cells to survive and overcome the negative impact on cell fitness. Again, these are clearly important findings. But, beyond stating that "the order of mutation is proscribed that the strongest drivers are unlikely to occur first.". There are several specific points made throughout the manuscript that are left without explanation. They are all interesting findings worthy of further exploration but fall short in my view of providing real mechanistic insight.

Why do certain Apc mutations decay and what is the basis for the apparent cooperation between different priming mutations and Wnt driver mutations? Examples of these points include:

Thank you. This comment provides the key example of where we have failed to pull together the broader implications of negative biases within a more accessible narrative. The Referee accurately describes both our challenge to the traditional

simplistic view and our safe, but unambitious conclusion, about the order of mutations. In fact, our interpretation that “the strongest drivers are unlikely to occur first” also allows for the reciprocal interpretation that weaker drivers can occur first. Consequently, our observations reveal something unappreciated about the tumour suppressive nature of *Apc*, namely that it is the most frequently mutated CRC driver event because it can occur as a primary or secondary event, and because the spectrum of mutations that allow it to transform are collectively able to exploit many contexts. We have included this mechanistic point in revised Discussion (**Lines 406-412**).

Line 219: Overall, the results suggest that different exon 3 mutations have different transformation potentials that are dependent on the nature of the priming field.

As stated, the observation highlights both the context dependency in the epithelium being permissive for transformation by *Ctnnb1* and that different exon 3 mutations have different transformation potentials depending on those contexts. Both of these aspects remain under-investigated hindering an immediate mechanistic interpretation. However, in light of our novel data from the GENIE CRC cohort showing increased retention of the second 20aa repeat and CID domain in APC in tumours harbouring both *APC* and *KRAS* mutations, we now reference literature in the Discussion (**lines 393-400**) that explores less appreciated direct interactions between KRAS and WNT pathway. For example, exon 3 mutations in β -Catenin can stabilize KRAS by interfering with GSK3 β phosphorylation sites and β -TRCP-mediated degradation mechanisms^{1,2}. It is therefore conceivable that different exon 3 mutations exert distinct effects on this regulatory interaction. Supporting this, our data demonstrate an exclusive co-occurrence of the β -Catenin p.S33P mutation with *Kras*^{G12D} priming (**lines 166–170**), suggesting functional specificity among β -catenin exon 3 variants.

Line 307: This indicates that the order of mutation dictates outcome with prior Fbxw7 mutation protecting against transformation by p.I35S and promoting it if mutated subsequently.

To reiterate our previous response to Referee 3 on this point, we originally co-submitted with another paper that dealt in depth with the different outcomes of ordering APC and FBXW7 in human organoids and felt that our observation of epistasis in the mouse model would best serve as orthogonal support of that study. In brief, the conclusion of the study, currently available as a pre-print was that **FBXW7 loss preceding loss of APC established an epigenetic context whereby APC loss failed to significantly perturb gene expression compared to that expected, and observed, when the order is reversed**³. We have now added additional detail from that report to give more context to the finding (**lines 295-298**).

There is also some question marks regarding the interpretation of parts of the data. For instance, with regards to Fig 3d, the following is stated:

“Analysed by group, the control and Trp53null cohorts showed minimal deviation in O/E ratios while most primed groups displayed an under representation of A (excepting

Kras^{G12D}) and E domain mutations. This suggests that priming promotes transformation by Apc mutations that have retained both the dimerization and Apc protein domains (bin A) and at least one Armadillo (Arm) repeat (bin B) while having lost β -catenin-binding 20 amino acid (20AA) repeats (bin E).”

The data shows that 3 primed groups follow the trends mentioned above (i.e. Fbxw7, Pten and Apc) while 2 do not (trp53 and Kras). Thus, the under representation of bin A seems highly dependent on the type of priming event. Stating that most primed groups behave this way is an over interpretation.

Before addressing the referee’s comment, we would like to note a correction to our data analysis. We identified an error in the coding script that affected the calculation of observed proportions for Apc bins in Fig. 3c,d and Extended Data Fig. 9c. Specifically, the script mistakenly used the column containing the sum of ENU-signature frequencies for each observed bin mutation, rather than the correct column that summed raw mutation counts per bin. As both measures scale with mutation count, the resulting plots appeared qualitatively correct and retained expected trends across total and individual cohorts. As shown below (**Rebuttal Figure 1**), the overall trends remain consistent, with minor changes to the O/E Chi-square test p-values.

We have now replaced the affected plots and updated the relevant manuscript text, also considering the referee’s comments. Specifically, we have adopted a stricter approach in our interpretation, focusing the analysis on statistically significant O/E ratios within the *Pten*^{null} and *Apc*^{het} cohort bins. In contrast to the control cohort, *Trp53*^{null} and *Kras*^{G12D} cohorts exhibit O/E ratios that more closely align with expectations, suggesting weaker selection pressures in these contexts (**lines 191-199**).

These more conservative revisions ensure that our interpretations are better aligned with the strength of the underlying data.

Fig.3c,d Old Version

Fig.3c,d New Version

E.D.Fig.9c Old Version

E.D.Fig.9c New Version

Rebuttal Figure 1 – Side-by-side comparison of old (left) and new (right) version of figure 3c, d and extended data figure 9c.

Secondly, it's not clear what the asterisks refer to in Fig 3d. Are the bars without asterisks not significant? For instance, the only priming events which show significant underrepresentation of bin A is *Pten* with a p value of 0.052. It's not clear whether the differences with *Apc* and *Fbxw7* are significant. Also how is significance measured in this case? Are the authors comparing O/E ratios in primed conditions vs unprimed conditions?

The asterisks in Fig. 3d indicate statistically significant differences between observed and expected values for each bin, assessed using a Chi-square test. This analysis is performed within each condition independently, not as a comparison between primed and unprimed cohorts. Only statistically significant results are marked with asterisks,

and borderline significance is indicated by the corresponding p -value. For bins where neither criterion is met, no annotation is shown. We have clarified this labelling system in the respective figure legends.

What are the control conditions in this experiment?

Control refers to the unprimed wildtype + ENU cohort.

Another point lacking clarity is the following: “Comparison of the distribution of mutations in tumours arising in different groups revealed that control, and combined Trp53null and Trp53het /N1-ICDhet primed groups tended to have a greater distance between N- and C-terminal hits and an overall wider distribution compared to that observed with Pten null priming (Fig. 3e,f).”

Besides the fact that this observation remains without explanation, it's not clear what data shows this and whether this is significant. One can understand the text but the actual demonstration of these results is difficult to follow in Fig 3.

Similar comments can be made for Fig 4g

We have revised the relevant text to more clearly reference the quantified data presented in Fig. 3f and Fig 4g. Specifically, we now highlight the statistically significant differences in bin combination distributions observed in the *Pten*^{null} cohort, as determined by Fisher's exact test showing a marked depletion of BB combinations and combinations involving bin E, consistent with findings in Fig. 3d, and a corresponding enrichment of BC and BD combinations (**lines 210-218**).

In respect to Fig 4g, the relevant section of the text has been revised to more clearly emphasize the statistically significant differences emerging in the combined KTF cohort, rather than in individual cohorts. Importantly, we have now provided mechanistic interpretations of these differences, which was previously lacking, to improve both clarity and interpretability (**lines 248-256**).

Additional comments:

As commented on by reviewer 1, in some passages the manuscript is quite challenging to read. To improve flow and comprehension additional explanations are required in certain passages. Examples include:

The analysis performed in Fig 4i, j and accompanying explanations in the text are not clear to this reviewer. It's not clear what conclusions can be drawn from this analysis.

Thank you. Figures 4i, j modelled a virtual *Apc* two-hit scenario across a theoretical cohort of 1,000 tumours. The analysis was designed to illustrate that mutation decay plays a more dominant role than mutational signature in shaping the observed mutation landscape. However, the original analysis clearly did not convey the manuscript's central conclusion that the most strongly transforming mutations are also subject to the strongest negative selection bias. The revised version instead plots the

actual number of tumours observed in the priming protocol against the *Apc* bin-specific decay rates inferred from Figure 4h, which we believe provides a clearer and more direct demonstration of this principle (**Rebuttal Figure 2; lines 271-273**).

Rebuttal Figure 2 – Side-by-side comparison of old figure 4i,j (left), replaced by new figure 4i (right). In new Fig.4i, estimated number of tumours in *Apc*^{het} (TE) cohort containing each *Apc* bin is plotted against the corresponding decay at 30 days post-ENU. Decay was calculated from panel 4h (1 - Prop. Tumours left).

The analysis of the human cohort begins with the following assumption: “ Reasoning that temporal first hits, occurring before transformative positive selection, would more closely correspond to signature predictions than subsequent events...”

Why would first hits more closely correspond to signature predictions than 2nd hits? This is not immediately obvious to this reviewer. Without a clear explanation of this it is difficult to follow the rest of the analysis in this section.

Our rationale was that, in the early stages of tumorigenesis, mutations are more likely to reflect the underlying mutational processes active in the normal tissue. However, it is also the case that a consequence of “the traditional, simplistic view that Wnt activating mutations are the primary event” is that as a tumour suppressor gene APC requires loss-of-function mutations in both alleles to mediate oncogenic transformation. According to this paradigm, both first and second hits would closely correspond to those predicted from normal mutational signatures. Our analyses demonstrated that in contrast, second hits are often subject to stronger selective pressures and may be biased in order to achieve a “just-right” optimum by selecting less prevalent mutational outcomes. This important point is apparent by considering APC truncating mutations without the need to order them and is powerful in

demonstrating active selection in the outcome of both APC hits. Therefore, in responding to the referee's specific comment and the more general request for clarity, we have revised the analysis to adopt this simpler approach. The revised strategy enables robust detection of mutation bin biases, including the strong under-representation of bin B, which mirrors the pattern observed in mouse (**Rebuttal Figure 3; lines 306-318**).

Rebuttal Figure 3 – Comparison of old figure 5a-d (top), replaced by new figure 5a-c (bottom). New figure legends: **a**, Overview of GENIE CRC samples by APC and KRAS mutation status. **b**, Expected frequency for each potential APC stop assuming an SNV, superimposed with observed APC mutations (two-hit tumours with truncations arising from SNVs; $n = 952$) from GENIE cohort in APC N-terminal half (aa 1–1,588). Frequencies were normalised against highest expected (multiple sites) and observed (aa: 1,450). Black dashed lines demarcate domain bins. Green dashed lines separate 20aa repeat domain retention boundaries. **c**, Observed/expected ratios in relation to APC domain bin.

Referee #3 (Remarks to the Author):

The authors have responded positively to my comments with additional data where required. The new human data is of great interest and strengthens the findings

I was intrigued by 2 interesting observations in the discussion and would be interested in the authors thoughts in rebuttal form only -

1. "However, CTNNB1 mutations are found at higher frequencies in polyps of patients which better aligns with the predominantly adenomatous lesions described here" - Can the authors speculate on why CTNNB1 lesions fail to progress to established cancers

when the wnt disrupting mutation would seem to have 'done the job' in initiating the lesion

We thank the reviewer for their comments and interest. Responding in a more discursive mode: There is plenty of evidence showing that CTNNB1 mutations are enriched in MSI-H CRCs, mainly in Lynch Syndrome patients⁴⁻⁷. This suggests that CTNNB1 mutations are selected and able to drive CRC in the MMRd context, mainly MLH1-deficient⁶, suggesting a need for extra events for “just-right” conditions.

Most CTNNB1-driven carcinomas seem to have a high tumour mutation burden (cBioportal) and most carcinomas display homozygous or hemizygous mutations in CTNNB1⁷, underscoring the need for extra WNT activation for progression. Of note, CTNNB1 copy number status in adenomas remains uninvestigated, limiting our ability to comment on earlier lesions.

Moving to non-mutational reasons for the reduced probability of progression, there are reports of WNT-driven and especially CTNNB1-mutant tumours being immune-excluded⁸⁻¹¹. This could explain why CTNNB1 mutations are enriched in MSI-H CRC, as these tumours are more immune infiltrated due to their increased antigenic burden; CTNNB1 mutations would lower infiltration. Some clinical reports support this¹². However, this does not explain why MSS colonic adenomas with CTNNB1 mutations have less potential to progress unless too-cold might be not “just-right” as some immune populations are tumour-promoting (neutrophils, macrophages, etc.). More observations of immune profiles in CTNNB1-mutant adenomas would be informative in this area.

2. "Further, recent studies have ascribed a “supercompetitor” behaviour to clones arising by biallelic inactivation of Apc whereby growth advantage is conferred, in part, by inhibition of self-renewal in neighbouring wildtype cells. Notably, such analyses focus on the growth properties of surviving clones and do not directly address their overall survival probability. Considered together, these observations suggest that mutations conferring the greatest potential for transformation have the highest bar to survival with priming acting rheostatically to lower the latter" - The supercompetitor growth advantage has been ascribed functionally to the secretion of Notum. Given that Notum is a responsive negative regulator of wnt, one might expect a proximal APC mutation to cause higher Notum expression - is there any evidence of quantitative association of Notum expression with different APC mutations - i.e. does a "just right" level of Notum expression explain the "just right" Wnt effect or is this a predominantly a cell-intrinsic phenomenon?.

Supporting this idea, our previous report on polyclonality included limited organoid CRISPR data showing that truncations perturbing the Armadillo domains (bin B) resulted in higher levels of NOTUM expression than more N or C-terminal truncations. From our observations on the loss of such mutations, this would be consistent with the highest level of NOTUM expression being “too high” and that suppression of neighbours is not sufficient to compensate for detrimental cell-intrinsic processes, as the reviewer proposes.

The original interpretation of the supercompetitor phenotype was that Wnt antagonists produced from Apc mutant cells were acting primarily to suppress wildtype neighbours, but perhaps an equally valid interpretation is that those cells receive a direct benefit by lowering the survival end of the rheostat enough for them to survive. A plausible hypothesis would be that a "just-right" level of Notum expression does explain the "just right" Wnt effect because it reflects both autocrine and paracrine effects?

References

1. Jeong, W.-J. *et al.* Ras Stabilization Through Aberrant Activation of Wnt/ β -Catenin Signaling Promotes Intestinal Tumorigenesis. *Sci Signal* **5**, (2012).
2. Lee, S. *et al.* β -Catenin-RAS interaction serves as a molecular switch for RAS degradation via GSK3 β . *EMBO Rep* **19**, (2018).
3. Chan, D. K. H. *et al.* Mutational order and epistasis determine the consequences of FBXW7 mutations during colorectal cancer evolution. Preprint at <https://doi.org/10.1101/2023.08.25.554836> (2023).
4. Ahadova, A., von Knebel Doeberitz, M., Bläker, H. & Kloor, M. CTNNB1-mutant colorectal carcinomas with immediate invasive growth: a model of interval cancers in Lynch syndrome. *Fam Cancer* **15**, (2016).
5. Ahadova, A. *et al.* Three molecular pathways model colorectal carcinogenesis in Lynch syndrome. *Int J Cancer* **143**, (2018).
6. Engel, C. *et al.* Associations of Pathogenic Variants in MLH1, MSH2, and MSH6 With Risk of Colorectal Adenomas and Tumors and With Somatic Mutations in Patients With Lynch Syndrome. *Gastroenterology* **158**, (2020).
7. Arnold, A. *et al.* The majority of β -catenin mutations in colorectal cancer is homozygous. *BMC Cancer* **20**, (2020).
8. Luke, J. J., Bao, R., Sweis, R. F., Spranger, S. & Gajewski, T. F. WNT/b-catenin pathway activation correlates with immune exclusion across human cancers. *Clinical Cancer Research* **25**, (2019).
9. de Galarreta, M. R. *et al.* β -catenin activation promotes immune escape and resistance to anti-PD-1 therapy in hepatocellular carcinoma. *Cancer Discov* **9**, (2019).
10. Liu, S., Ding, G., Zhou, Z. & Feng, C. β -Catenin-driven adrenocortical carcinoma is characterized with immune exclusion. *Onco Targets Ther* **11**, (2018).
11. Pinyol, R., Sia, D. & Llovet, J. M. Immune exclusion-WNT/CTNNB1 class predicts resistance to immunotherapies in HCC. *Clinical Cancer Research* **25**, (2019).
12. Élez, E. *et al.* A Comprehensive Biomarker Analysis of Microsatellite Unstable/Mismatch Repair Deficient Colorectal Cancer Cohort Treated with Immunotherapy. *Int J Mol Sci* **24**, 118 (2022).